# Testing with Non-identically Distributed Samples

**Shivam Garg**                                    *shivam.garg55@gmail.com*
*Microsoft Research*

**Chirag Pabbaraju**                               *cpabbara@stanford.edu*
*Stanford University*

**Kirankumar Shiragur**                            *kshiragur@microsoft.com*
*Microsoft Research*

**Gregory Valiant**                                *gvaliant@stanford.edu*
*Stanford University*

**Reviewed on OpenReview:** *https://openreview.net/forum?id=FUzvztzBlW*

## Abstract

We examine the extent to which sublinear-sample property testing and estimation apply to settings where samples are independently but not identically distributed. Specifically, we consider the following distributional property testing framework: Suppose there is a set of distributions over a discrete support of size $k$, $\mathbf{p}_1, \mathbf{p}_2, \ldots, \mathbf{p}_T$, and we obtain $c$ independent draws from each distribution. Suppose the goal is to learn or test a property of the average distribution, $\mathbf{p}_{\text{avg}}$. This setup models a number of important practical settings where the individual distributions correspond to heterogeneous entities — either individuals, chronologically distinct time periods, spatially separated data sources, etc. From a learning standpoint, even with $c = 1$ samples from each distribution, $\Theta(k/\varepsilon^2)$ samples are necessary and sufficient to learn $\mathbf{p}_{\text{avg}}$ to within error $\varepsilon$ in $\ell_1$ distance. To test uniformity or identity — distinguishing the case that $\mathbf{p}_{\text{avg}}$ is equal to some reference distribution, versus has $\ell_1$ distance at least $\varepsilon$ from the reference distribution, we show that a linear number of samples in $k$ is necessary given $c = 1$ samples from each distribution. In contrast, for $c \geq 2$, we recover the usual sublinear sample testing guarantees of the i.i.d. setting: we show that $O(\sqrt{k}/\varepsilon^2 + 1/\varepsilon^4)$ total samples are sufficient, matching the optimal sample complexity in the i.i.d. case in the regime where $\varepsilon \geq k^{-1/4}$. Additionally, we show that in the $c = 2$ case, there is a constant $\rho > 0$ such that even in the linear regime with $\rho k$ samples, no tester that considers the multiset of samples (ignoring which samples were drawn from the same $\mathbf{p}_i$) can perform uniformity testing. We further extend our techniques to the problem of testing "closeness" of two distributions: given $c = 3$ independent draws from each of $\mathbf{p}_1, \mathbf{p}_2, \ldots, \mathbf{p}_T$ and $\mathbf{q}_1, \mathbf{q}_2, \ldots, \mathbf{q}_T$, one can distinguish the case that $\mathbf{p}_{\text{avg}} = \mathbf{q}_{\text{avg}}$ versus having $\ell_1$ distance at least $\varepsilon$ using $O(k^{2/3}/\varepsilon^{8/3})$ total samples, where $k$ is an upper bound on the support size, matching the optimal sample complexity of the i.i.d. setting up to the $\varepsilon$-dependence.

## 1 Introduction

The problem of estimating statistics of an unknown distribution, or determining whether the distribution in question possesses a property of interest has been studied by the Statistics community for well over a century. Interest in these problems from the TCS community was sparked by the seminal paper of Goldreich (Goldreich & Ron, 2000), demonstrating the surprising result that such tasks might be accomplished given a sample size that scales *sublinearly* with the support of the underlying distribution. Over the subsequent 20+ years, the lay of the land of sublinear sample property testing and estimation has largely been worked out: for a number of natural properties of distributions (or pairs of distributions), testing or estimation can be

accomplished with a sample size that is sublinear in the support size of the distributions in question, and in most settings, the optimal sample complexities are now known to constant or subconstant factors.

More recently, these questions have been revisited with an eye towards relaxing the assumption that samples are drawn i.i.d. from the distribution in question. The majority of such departures focused on relaxing the independence assumption; this includes work from the robust statistics perspective where some portion of samples may be adversarially corrupted, as well as other models of dependent sampling.

In this work, we consider the implications of relaxing the more innocuous-seeming assumption of identical samples. Given samples drawn independently from a collection of distributions, $\mathbf{p}_1, \ldots, \mathbf{p}_T$, there are a number of different questions one can ask, including estimating properties of how the various distributions are related. Here, however, we focus on the practically motivated question of testing or estimating properties of the *average* distribution $\mathbf{p}_{\mathrm{avg}} = \frac{1}{T} \sum \mathbf{p}_i$. To what extent is sublinear sample testing or estimation still possible in this setting, and do existing estimators suffice? Below, we describe several concrete practical settings that motivate these questions.

**Example 1.** ***The "Federated" Setting:*** *Consider the setting where there are a number of users/ individuals, each with a corresponding distribution (over online purchases, language, biometrics, etc.) and the goal is to test whether the average distribution is close to some reference. There might be significant heterogeneity between users, and the number of samples (i.e., amount of data) collected from each user might be very modest, due to privacy, bandwidth, or power considerations. Analogous settings arise when compiling training datasets for LLMs, as there is significant heterogeneity between documents, websites, or clusters of websites, and lightweight tools for testing properties of the average/aggregate distribution are useful.*

**Example 2.** ***Chronological Heterogeneity:*** *Consider a setting where samples are collected chronologically, and the underlying distribution varies over time. Consider a wearable technology that periodically samples biometric statistics, or language spoken. The goal may be to test whether the average distribution is close to some reference (with the goal of providing an early warning of illness in the case of biometrics, or for early detection of dementia in the case of sampling words spoken). Ideally, this test can be successfully accomplished with far less data than would be required to* learn. *There is significant heterogeneity across time but samples collected close together can be regarded as being sampled from the same distribution, $\mathbf{p}_i$.*

**Example 3.** ***Spatially Segregated Sampling:*** *Consider a setting where the goal is to sample species (of animals, bacteria, etc.) and test whether the overall distribution is close to a reference distribution. In many such settings, the distribution of species has significant spatial variation — the distribution of soil bacteria in a grassy spot is quite different from the distribution in dense forest, similarly, the distribution of wildlife observed by trail-cameras in national parks varies according to the location of the cameras. Again, the goal is to accomplish this statistical test with less data than would be required to learn the underlying average distribution.*

In some of these motivating settings, one may assume some structure in how the various distributions relate to each other— for example in chronological settings, for most indices $i$, $\mathbf{p}_i$ is similar to $\mathbf{p}_{i-1}$. Nevertheless, it is still worth understanding whether such assumptions *must* be leveraged. Indeed, for *learning* the average distribution $\mathbf{p}_{\mathrm{avg}}$ to within total variation distance $\varepsilon$, the sample complexity is the same as in the i.i.d. case, $O(k/\varepsilon^2)$ where $k$ is the support size, *without any assumptions* on the $\mathbf{p}_i$s. At the highest level, our results demonstrate that as soon as one draws at least $c = 2$ samples from each distribution, this non-identically distributed setting recovers much of the power of sublinear-sample property testing. The caveat is that one must design new algorithms for this setting that are cognizant of which distribution each sample was drawn from; as we show, testers that return a function of the multiset of examples do not suffice.

## 1.1 Summary of Results

First, we establish that in the case that we draw $c = 1$ samples from each distribution $\mathbf{p}_1, \ldots, \mathbf{p}_T$, one *can* learn the average distribution with the same sample complexity as in the i.i.d. setting. Despite this, with $c = 1$ sample from each distribution, sublinear-sample uniformity or identity testing is impossible:

**Claim 1.1** (Learning the distribution, proof in Appendix A)**.** *Given access to $T$ distributions, $\mathbf{p}_1, \ldots, \mathbf{p}_T$, each supported on a common domain of size $\leq k$, for any $\varepsilon > 0$, given $c = 1$ samples drawn from each*

$\boldsymbol{p}_i$, *one can output a distribution* $\tilde{\boldsymbol{p}}_{\mathrm{avg}}$ *such that with probability at least* $2/3$, $\mathrm{d}_{\mathrm{TV}}(\boldsymbol{p}_{\mathrm{avg}}, \tilde{\boldsymbol{p}}_{\mathrm{avg}}) \leq \varepsilon$, *provided* $T = O(k/\varepsilon^2)$. *Furthermore,* $\Omega(k/\varepsilon^2)$ *samples are necessary, even if* $\boldsymbol{p}_1 = \ldots = \boldsymbol{p}_T = \boldsymbol{p}_{avg}$ *for such a guarantee.*

**Claim 1.2** (Impossibility result with $c = 1$, proof in Appendix B)**.** *Given access to* $T \leq k/2$ *distributions,* $\boldsymbol{p}_1, \ldots, \boldsymbol{p}_T$, *and* $c = 1$ *sample drawn from each distribution, no tester can distinguish the case that the average distribution,* $\boldsymbol{p}_{\mathrm{avg}}$, *is the uniform distribution of support* $k$, *versus has total variation distance at least* $1/4$ *from* $\mathrm{Unif}(k)$, *with probability of success greater than* $2/3$.

The above impossibility of sublinear-sample testing in the case of $c = 1$ sample from each distribution follows from the following simple construction, which also provides intuition behind the positive results for $c \geq 2$ : Consider the instance where $T = k/2$, and each $\mathbf{p}_i$ is a point mass on a distinct element, versus the instance where each $\mathbf{p}_i$ is a uniform distribution over two distinct domain elements that are disjoint from the supports of $\mathbf{p}_j$ for $j \neq i$. In both instances, given $c = 1$ sample from each distribution, one will observe $T = k/2$ distinct elements each observed exactly once. Despite this, in the second instance, $\mathbf{p}_{\mathrm{avg}}$ is uniform over support $k$, whereas in the former the average distribution has total variation distance $1/4$ from $\mathrm{Unif}(k)$.

Our first main result is that with just $c = 2$ samples from each distribution, we recover the full strength of sublinear sample uniformity (and identity) testing. The key intuition is that with two samples from each distribution, we can count both the collision statistics within distributions, as well as the collisions across distributions, and build an unbiased estimator for $\|\mathbf{p}_{\mathrm{avg}}\|_2^2$. We then leverage techniques from Diakonikolas et al. (2016) and adapt them to this setting of non-identical samples to bound the variance of our estimator to get the following sublinear sample complexity for uniformity testing.

**Theorem 1.3** (Uniformity testing, proof in Section 3)**.** *There is an absolute constant,* $\alpha$, *such that given access to* $T$ *distributions,* $\boldsymbol{p}_1, \ldots, \boldsymbol{p}_T$, *each supported on a common domain of size* $\leq k$, *for any* $\varepsilon > 0$, *provided* $T \geq \alpha(\sqrt{k}/\varepsilon^2 + 1/\varepsilon^4)$ *and given* $c = 2$ *samples drawn from each* $\boldsymbol{p}_i$, *there exists a testing algorithm that succeeds with probability* $2/3$ *and distinguishes whether,*

$$\boldsymbol{p}_{\mathrm{avg}} = \mathrm{Unif}(k) \quad versus \quad \mathrm{d}_{\mathrm{TV}}(\boldsymbol{p}_{\mathrm{avg}}, \mathrm{Unif}(k)) \geq \varepsilon \ .$$

Note that when $\varepsilon \geq k^{-1/4}$, our sample complexity for uniformity testing simplifies to $O(\sqrt{k}/\varepsilon^2)$, which matches the optimal sample complexity for testing uniformity in the i.i.d. setting. Since the setting of non-identical samples is a generalization of the i.i.d. setting, the lower bound of $\Omega(\sqrt{k}/\varepsilon^2)$ for testing uniformity in the i.i.d. setting also holds in our setting; hence we get optimal sample complexity in the regime where $\varepsilon \geq k^{-1/4}$.

Our results also hold for identity testing, through standard reduction techniques. In particular, we can adapt the reduction from identity testing to uniformity testing of Goldreich (2016) to our setting with non-identical distributions, yielding the following result on the reduction from an instance of identity testing to an instance of uniformity testing.

**Lemma 1.4** (Identity to uniformity testing, proof in Section 4)**.** *Given an arbitrary reference distribution* $\boldsymbol{q}$ *supported on* $\leq k$ *elements, there exists a pair* $(\Phi_{\boldsymbol{q}}, \Psi_{\boldsymbol{q}})$ *that maps distributions and samples over* $[k]$ *to distributions and samples over* $[4k]$, *and satisfies the following,*

- *For a sequence of distributions* $\boldsymbol{q}_1, \ldots, \boldsymbol{q}_T$ *such that* $\mathrm{avg}(\boldsymbol{q}_1, \ldots, \boldsymbol{q}_T) = \boldsymbol{q}$, *the map* $\Phi_{\boldsymbol{q}}$ *satisfies,*

$$\mathrm{avg}(\Phi_{\boldsymbol{q}}(\boldsymbol{q}_1), \ldots, \Phi_{\boldsymbol{q}}(\boldsymbol{q}_T)) = \mathrm{Unif}(4k).$$

- *For two sequences of distributions* $\boldsymbol{p}_1^1, \ldots, \boldsymbol{p}_T^1$ *and* $\boldsymbol{p}_1^2, \ldots, \boldsymbol{p}_T^2$, *the map* $\Phi_{\boldsymbol{q}}$ *satisfies,*

$$\mathrm{d}_{\mathrm{TV}}(\mathrm{avg}(\Phi_{\boldsymbol{q}}(\boldsymbol{p}_1^1), \ldots, \Phi_{\boldsymbol{q}}(\boldsymbol{p}_T^1)), \mathrm{avg}(\Phi_{\boldsymbol{q}}(\boldsymbol{p}_1^2), \ldots, \Phi_{\boldsymbol{q}}(\boldsymbol{p}_T^2))) \geq \frac{1}{4} \cdot \mathrm{d}_{\mathrm{TV}}(\mathrm{avg}(\boldsymbol{p}^1, \ldots, \boldsymbol{p}_T^1), \mathrm{avg}(\boldsymbol{p}^2, \ldots, \boldsymbol{p}_T^2)) \ .$$

The above result combined with the result for uniformity testing, immediately yields the following result for identity testing. Our bounds for identity testing are optimal in the worst case for large values of $\varepsilon \geq k^{-1/4}$.

**Corollary 1.5** (Identity testing)**.** *There is an absolute constant, $\alpha$, such that given a reference distribution $\boldsymbol{q}$ supported on $\leq k$ elements and $c = 2$ samples from each of $T$ distributions, $\boldsymbol{p}_1, \ldots, \boldsymbol{p}_T$, there exists a testing algorithm that succeeds with probability $2/3$ and distinguishes whether,*

$$\boldsymbol{p}_{\mathrm{avg}} = \boldsymbol{q} \quad versus \quad \mathrm{d}_{\mathrm{TV}}(\boldsymbol{p}_{\mathrm{avg}}, \boldsymbol{q}) \geq \varepsilon \ ,$$

*provided $T \geq \alpha(\sqrt{k}/\varepsilon^2 + 1/\varepsilon^4)$.*

We note that our sublinear sample testing algorithms leverage the knowledge of which distribution each sample was drawn from. As the following theorem asserts, this is necessary — there is a constant $\rho > 0$ such that even in the linear regime with $\rho k$ samples, there is no testing algorithm that simply returns a function of the multiset of samples, throwing away the information about which samples were drawn from the same distributions.

**Theorem 1.6** (Lower bound for "pooled" estimators, proof in Section 5)**.** *There is an absolute constant, $\rho > 0$, such that given access to distributions $\boldsymbol{p}_1, \ldots, \boldsymbol{p}_{\rho k}$ supported on a domain of size $k$, given $c = 2$ samples from each of $\boldsymbol{p}_1, \ldots, \boldsymbol{p}_{\rho k}$, no tester that returns a function of the multiset of samples (i.e., ignoring the information about which samples were drawn from the same distributions) can distinguish the case*

$$\boldsymbol{p}_{\mathrm{avg}} = \mathrm{Unif}(k) \quad versus \quad \mathrm{d}_{\mathrm{TV}}(\boldsymbol{p}_{\mathrm{avg}}, \mathrm{Unif}(k)) \geq 0.01$$

*with success probability at least $2/3$.*

Our positive results, and the intuition behind the uniformity testing algorithm also naturally extend to testing properties of pairs of distributions. In particular, we consider the "closeness" testing problem in this non-i.i.d. setting: given two sequences of distributions supported on a set of size $k$, $\mathbf{p}_1, \ldots, \mathbf{p}_T$ and $\mathbf{q}_1, \ldots, \mathbf{q}_T$ and $c$ i.i.d. draws from each of the distributions, how many total samples are required to distinguish between $\mathbf{p}_{\mathrm{avg}} = \mathbf{q}_{\mathrm{avg}}$ versus $\mathrm{d}_{\mathrm{TV}}(\mathbf{p}_{\mathrm{avg}}, \mathbf{q}_{\mathrm{avg}}) > \varepsilon$? In this setting, we show that, given $c = 3$ samples from each distribution, we recover the optimal sample complexity of $O(k^{2/3})$ from the i.i.d. setting, up to the $\varepsilon$ dependence (Batu et al., 2013; Chan et al., 2014). While our proof leverages $c = 3$ samples to simplify some of the analysis, we strongly believe that an identical result holds for $c = 2$ examples from each distribution.

**Theorem 1.7** (Closeness testing, proof in Appendix G)**.** *There is an absolute constant, $\alpha$, such that for two sequences of distributions, supported on a common domain of size $k$, $\boldsymbol{p}_1, \ldots, \boldsymbol{p}_T$, and $\boldsymbol{q}_1, \ldots, \boldsymbol{q}_T$, given $c = 3$ samples from each of the distributions, there exists a testing algorithm that succeeds with probability $2/3$ and distinguishes whether*

$$\boldsymbol{p}_{\mathrm{avg}} = \boldsymbol{q}_{\mathrm{avg}} \ versus \ \mathrm{d}_{\mathrm{TV}}(\boldsymbol{p}_{\mathrm{avg}}, \boldsymbol{q}_{\mathrm{avg}}) \geq \varepsilon,$$

*provided $T \geq \alpha(k^{2/3}/\varepsilon^{8/3})$.*

Finally, we note that the interesting phase shift between $c = 1$ and $c > 1$ holds only in the non-Poisson setting. Results in the Poisson setting remain the same as that of the i.i.d. setting. In particular, if instead of $c$ samples, we are given $Poi(c)$ samples from each distribution, then even in the case of $c = 1$, we recover all testing results known in the i.i.d. setting. This follows from the following basic fact: receiving $Poi(c)$ samples from every distribution is equivalent in distribution to receiving $Poi(cT)$ samples from the average distribution. A more formal version of this result is stated in Appendix C for identity testing, but such a result also holds for testing other properties. To see how this bypasses the obstruction discussed in Claim 1.2, note that with Poi(1) samples per distribution, in the point-mass case, each distribution yields repeated draws of its single element, whereas when the distributions have support on two distinct elements, within-group collisions start to occur.

Proving our upper bounds mostly involves adapting the techniques from the i.i.d. case to our setting. Establishing the lower bound for pooled samples is more complex, involving a new construction that might be of independent interest. Nonetheless, more than the techniques, our main contribution is the problem definition and conceptual takeaways. Commonly, when collecting samples, heterogeneity invariably arises, particularly when these samples are gathered over various time points, geographical locations, or from different users, as highlighted in the examples above. Thus, it is natural to ask whether sublinear sample testers are robust to such departures from the i.i.d. setting. We demonstrate that while existing testers are not

directly applicable (as evidenced by our lower bound for pooled samples), they can be effectively adapted, provided one collects multiple samples per distribution and leverages the source information of samples. For example, for identity testing, a standard estimator involves counting collisions, treating all collisions similarly. In contrast, our modified estimator uses a weighted collision count, with weights varying based on whether collisions occur within samples from the same distribution or not.

## 1.2 Future Directions

There are many possible directions for future work. From a technical standpoint, a concrete goal is to match the i.i.d. $\varepsilon$-dependence for uniformity and closeness. For uniformity, the gap comes from variance-reducing terms in the i.i.d. analysis that are not available when distributions differ (see Remark 5). For closeness, the current $\varepsilon^{-8/3}$ dependence comes from the classical collision analysis (Batu et al., 2013); matching the i.i.d. $\varepsilon^{-4/3}$ will likely require a new analysis beyond collisions (e.g., adapting the counts-based statistic of Chan et al. (2014) to the non-i.i.d. setting). It is not clear to us whether that optimal $\varepsilon$-dependence is achievable here.

Another direction is improving the number of samples per distribution. For closeness, we currently use $c = 3$ to keep the collision-based $\ell_2$ estimate independent of the heavy/light estimation; with a more careful reuse of the same samples and explicit dependency tracking, $c = 2$ may suffice. For uniformity, we show that $c = 1$ is impossible in the worst case; it is natural to ask for structural conditions under which $c = 1$ becomes feasible. If all $p_i$ equal $p_{\mathrm{avg}}$, the i.i.d. guarantees apply; more generally, if each $p_i$ is sufficiently close to $p_{\mathrm{avg}}$, we expect i.i.d. rates to persist with a small additional failure probability. As a simple check, suppose each $p_i$ is within $\delta/T$ TV distance of $p_{\mathrm{avg}}$. Then the joint distribution of the $T$ samples is within TV distance $\leq \delta$ of $T$ i.i.d. samples from $p_{\mathrm{avg}}$. Hence any optimal i.i.d. tester applies with an extra failure probability $\delta$. It would be interesting to study whether larger deviations beyond this naive bound can be tolerated.

Finally, beyond the specific problems addressed in this work, there are many obvious directions, including mapping the sample complexities for other natural property testing and estimation problems within the framework of non-identically distributed samples. Studying properties of the average distribution is a natural starting point, though other summary statistics of a collection of distributions are also worth considering. For example, in chronological settings, one might be interested in how certain properties vary over time, or predicting properties of future distributions. More broadly, probing the robustness of sublinear sample testing from other perspectives may yield further surprises with practical implications.

## 2 Related Work

Interest from the theoretical computer science community on distribution testing, given i.i.d. sample access, was sparked by the problem of distinguishing whether an unknown probability distribution was the uniform distribution over support $k$, versus has total variation distance at least $\varepsilon$ from this uniform distribution. The early work of Goldreich & Ron (2000) proposed a collision-based estimator, demonstrating that testing this property could be accomplished given significantly fewer samples than would be required to *learn* the distribution in question. Since then, a line of work eventually pinned down the exact sample complexity of $\Theta(\sqrt{k}/\varepsilon^2)$ — optimal to constant factors including the dependence on the probability of the tester failing (Paninski, 2008; Valiant & Valiant, 2017; Diakonikolas et al., 2014; 2016; 2017). For a comprehensive exposition of this line of work, we refer the reader to Chapters 2 and 3 in Canonne (2022).

A closely related problem to uniformity testing is testing *identity*, where the task is to determine if an unknown probability distribution, $\mathbf{p}$, is equal to versus far from a fixed reference distribution, $\mathbf{q}$, given i.i.d sample access (Batu et al., 2001). While "instance-optimal" estimators — estimators that are optimal for *every* $\mathbf{q}$ — are known (Valiant & Valiant, 2017), among reference distributions supported on $\leq k$ elements, the uniform distribution is the most difficult to test. Beyond this, building off Diakonikolas & Kane (2016), Goldreich (2016) gave a reduction from identity testing to uniformity testing, showing that any uniformity testing algorithm can be leveraged in a black-box fashion for identity testing. We leverage both the high-level design of the optimal uniformity testing algorithms, and this reduction.

For the problem of "closeness" testing, the goal is still to distinguish $\mathbf{p} = \mathbf{q}$ from the case that these two distributions have distance at least $\varepsilon$, though now $\mathbf{q}$ is not a fixed reference distribution, and instead we are given samples from both distributions. In the i.i.d. sample setting, for distributions of support size $k$, a sample complexity of $O(k^{2/3} \log k / \epsilon^{8/3})$ was established in Batu et al. (2001; 2013), with the optimal sample complexity of $\Theta(\max(k^{2/3}/\epsilon^{4/3}, k^{1/2}/\epsilon^2))$ later pinned down in Chan et al. (2014).

Beyond uniformity, identity, and closeness testing, there is an enormous body of work on testing or estimating other distribution properties or properties of pairs of distributions. Broadly speaking, the optimal sample complexities for these tasks are now pinned down to constant factors, at least in the setting where one obtains i.i.d. samples

**Beyond i.i.d. samples:**    The questions of learning, property testing and estimation have also been considered in various settings that deviate from the idealized setting of samples drawn i.i.d. from a fixed distribution. These include works from the perspective of robust statistics, where some portion of the samples may be adversarially corrupted (see e.g., Diakonikolas et al. (2019); Diakonikolas & Kane (2019); Charikar et al. (2017), as well as the broad area of time-series analysis where observations are drawn from time-varying distributions (e.g., Chernoff & Zacks (1964); Kolar et al. (2010); Lampert (2015)).

Most similar to the present paper are works that explore learning and testing questions in various settings where there are a number of data sources, each providing a batch of samples. In Qiao & Valiant (2018) and the followup papers Jain & Orlitsky (2020); Chen et al. (2020), it is assumed that an unknown $1 - \varepsilon$ fraction of data sources correspond to identical distributions, each providing a batch of $c$ i.i.d. samples, and no assumptions are made about the samples in the remaining batches. The main results here are that as $c$ increases, the batches can be leveraged for learning: specifically, the distribution of the $1 - \varepsilon$ fraction can be learned to accuracy $O(\varepsilon/\sqrt{c})$. The papers Tian et al. (2017); Vinayak et al. (2019) explore the setting where batches of size $c$ are drawn i.i.d. from a collection of heterogeneous distributions, and the goal is to learn the multiset of distributions. The punchline here is that the multiset of distributions can be learned given asymptotically fewer draws from each distribution than would be necessary to learn them in isolation. While the focus in these works is on the setting where the distributions have support size 2 (i.e. the setting corresponds to flipping coins with heterogeneous probabilities), the results apply more broadly. The paper of Levi et al. (2013) considered the setting of a collection of $T$ distributions over large support size, and explored the question of testing whether all distributions in the collection are the same, versus have significant variation. They considered two sampling models — a "query" model where one can adaptively choose which distribution to draw a sample from, and a weaker model where each sample is drawn from a distribution selected uniformly at random from the $T$ distributions. We note that in this latter model, testing properties of the average distribution trivially reduces to the i.i.d. setting. Later works by Diakonikolas & Kane (2016); Aliakbarpour et al. (2016) improve the results obtained by Levi et al. (2013). Finally, the paper by Aliakbarpour & Silwal (2019) considers the setting with a collection of distributions, and given one sample (in expectation) from each distribution, the task is to distinguish whether all the distributions are equal to a reference distribution, or all distributions are far from it. In contrast, our focus is on distinguishing whether the average distribution is equal to or far from a reference distribution. They also consider the case where we only have sample access to the reference distribution. General non-vacuous results are not possible in their setting given a single sample from each distribution. To bypass this, they impose a structural condition on the collection of distributions. We also face a similar issue and show that it can be bypassed in our setting by drawing multiple samples from each distribution.

## 3  Uniformity testing from non-identical samples

In this section, we prove our sample complexity results for uniformity testing in the setting of non-identical samples. Our tester for uniformity in the setting of non-identical samples is based on collisions and constructs an unbiased estimator for $\|\mathbf{p}_{\text{avg}}\|_2^2$. As $\|\mathbf{p}_{\text{avg}}\|_2^2$ equals $\frac{1}{k}$ when $\mathbf{p}_{\text{avg}}$ is uniform and is larger than $(1+\varepsilon^2)\frac{1}{k}$ when $\mathbf{p}_{\text{avg}}$ is $\varepsilon$-far from uniform, we use this separation to solve the testing problem. The main part of the proof constitutes bounding the variance of the estimator that we construct in order to show that it concentrates around its mean. We adapt the proof from Diakonikolas et al. (2016) to the setting of non-identical samples to bound the variance of our estimator.

To test uniformity, we build an unbiased estimator for $\|\mathbf{p}_{\text{avg}}\|_2^2$. In the following, we provide a description of this estimator. For each $t \in [T]$, let $(X_t(1), X_t(2))$ denote the 2 i.i.d. samples drawn from the distribution $\mathbf{p}_t$. For each $t \in [T]$, we define,

$$Y_t = \mathbb{1}[X_t(1) = X_t(2)] \text{ and } Y_{st} = \mathbb{1}[X_t(1) = X_s(1)] .$$

Note that $\mathbb{E}[Y_t] = \|\mathbf{p}_t\|_2^2$ and $\mathbb{E}[Y_{st}] = \langle \mathbf{p}_t, \mathbf{p}_s \rangle$ for all $s, t \in [T]$. Using these random variables, we define an estimator $Z$ as follows,

$$Z = \frac{1}{T^2}\Big( \sum_{t \in [T]} Y_t + 2\sum_{s<t} Y_{st} \Big). \tag{1}$$

$Z$ is an unbiased estimator of $\|\mathbf{p}_{\text{avg}}\|_2^2$. In the following we formally state this result and bound the variance of the estimator.

**Lemma 4.** *The estimator $Z$ defined in Equation* (1) *satisfies the following:*

$$\mathbb{E}[Z] = \|\boldsymbol{p}_{\text{avg}}\|_2^2 \text{ and } \operatorname{Var}[Z] \leq \frac{4}{T^2}\Big\|\boldsymbol{p}_{\text{avg}}\Big\|_2^2 + \frac{48}{T}\Big\|\boldsymbol{p}_{\text{avg}}\Big\|_3^3.$$

*Proof of Lemma 4.* The expectation of the random variable $Z$ is given by

$$\mathbb{E}[Z] = \frac{1}{T^2}\Big( \sum_{t \in [T]} \|\mathbf{p}_t\|_2^2 + 2\sum_{s<t}\langle \mathbf{p}_t, \mathbf{p}_s \rangle \Big) = \|\mathbf{p}_{\text{avg}}\|_2^2 .$$

Therefore $Z$ is an unbiased estimator for $\|\mathbf{p}_{\text{avg}}\|_2^2$. In the remainder of the proof, we bound the variance of this estimator $Z$.

$$\operatorname{Var}[Z] = \mathbb{E}\Big[\Big(Z - \mathbb{E}[Z]\Big)^2\Big] = \frac{1}{T^4}\mathbb{E}\Big[\Big( \sum_{t \in [T]}(Y_t - \mathbb{E}[Y_t]) + 2\sum_{s<t}(Y_{st} - \mathbb{E}[Y_{st}]) \Big)^2\Big].$$

Define

$$\hat{Y}_t = Y_t - \mathbb{E}[Y_t] \text{ and } \hat{Y}_{st} = Y_{st} - \mathbb{E}[Y_{st}],$$

and note that $\mathbb{E}[\hat{Y}_t] = 0$ and $\mathbb{E}[\hat{Y}_{st}] = 0$. Rewriting the variance of $Z$ in terms of these new random variables we get the following,

$$\operatorname{Var}[Z] = \frac{1}{T^4}\mathbb{E}\Big[\Big( \sum_{t \in [T]}\hat{Y}_t + 2\sum_{s<t}\hat{Y}_{st} \Big)^2\Big] \leq \frac{2}{T^4}\mathbb{E}\Big[\Big( \sum_{t \in [T]}\hat{Y}_t \Big)^2 + 4\Big(\sum_{s<t}\hat{Y}_{st}\Big)^2\Big]$$

$$= \frac{2}{T^4}\Big(\mathbb{E}\Big[\Big( \sum_{t \in [T]}\hat{Y}_t \Big)^2\Big] + 4 \cdot \mathbb{E}\Big[\Big(\sum_{s<t}\hat{Y}_{st}\Big)^2\Big]\Big) \tag{2}$$

In the second inequality, we used $(a + b)^2 \leq 2(a^2 + b^2)$. In the final inequality, we used the linearity of expectation. To bound the variance of $Z$, we bound the terms in the final expression separately. We start with the first term.

$$\mathbb{E}\Big[\Big( \sum_{t \in [T]}\hat{Y}_t \Big)^2\Big] = \mathbb{E}\Big[ \sum_{t \in [T]}\hat{Y}_t^2 + 2\sum_{s<t}\hat{Y}_t\hat{Y}_s \Big] = \sum_{t \in [T]}\mathbb{E}[\hat{Y}_t^2] + 2\sum_{s<t}\mathbb{E}[\hat{Y}_t]\mathbb{E}[\hat{Y}_s] = \sum_{t \in [T]}\mathbb{E}[\hat{Y}_t^2]$$

$$= \sum_{t \in [T]}(\mathbb{E}[Y_t^2] - \mathbb{E}[Y_t]^2) \leq \sum_{t \in [T]}\mathbb{E}[Y_t^2] = \sum_{t \in [T]}\mathbb{E}[Y_t] = \sum_{t \in [T]}\|\mathbf{p}_t\|_2^2. \tag{3}$$

In the first equality, we expanded the expression. In the second equality, we used the fact that the random variables $Y_s, Y_t$ are independent for all $s, t \in [T]$ and $s \neq t$. In the third equality, we used $\mathbb{E}[\hat{Y}_t] = 0$ for all $t \in [T]$. The fourth and fifth inequalities are immediate. The sixth equality follows because $Y_t$ is an indicator

random variable. In the seventh inequality, we substituted the value of $\mathbb{E}[Y_t] = \|\mathbf{p}_t\|_2^2$. Similar to the above, we now bound the second term of Equation (2).

$$
\begin{aligned}
\mathbb{E}\Big[\Big(\sum_{s<t}\hat{Y}_{st}\Big)^2\Big] &= \mathbb{E}\Big[\Big(\sum_{s<t}Y_{st}\Big)^2\Big] - \Big(\mathbb{E}\Big[\sum_{s<t}Y_{st}\Big]\Big)^2 \\
&= \underbrace{\sum_{s<t}\mathbb{E}\big[Y_{st}^2\big]}_{\binom{T}{2}\text{ terms}} + \underbrace{\sum_{\substack{s<t\\a<b\\|\{s,t,a,b\}|=3}}\mathbb{E}[Y_{st}Y_{ab}]}_{6\binom{T}{3}\text{ terms}} + \underbrace{\sum_{\substack{s<t\\a<b\\|\{s,t,a,b\}|=4}}\mathbb{E}[Y_{st}Y_{ab}]}_{6\binom{T}{4}\text{ terms}} - \underbrace{\Big(\mathbb{E}\Big[\sum_{s<t}Y_{st}\Big]\Big)^2}_{\binom{T}{2}^2\text{ terms}}
\end{aligned} \tag{4}
$$

The first term above is equal to $\sum_{s<t}\langle\mathbf{p}_t,\mathbf{p}_s\rangle$ — we can club this together with the summands in the summation in the last term that are of the form $\mathbb{E}[Y_{st}]^2$. For the third term, observe that if the indices $s,t,a,b$ are all distinct, the random variables $Y_{st}$ and $Y_{ab}$ are independent, and hence $\mathbb{E}[Y_{st}Y_{ab}] = \mathbb{E}[Y_{st}]\mathbb{E}[Y_{ab}]$. But all these summands are also included in the summation in the last term, and hence will get canceled out. Finally, we are left with the second term. Observe that for any 3 fixed distinct values, there are 6 possible orderings of the indices $s < t$ and $a < b$ such that the multiset $\{s,t,a,b\}$ has exactly these distinct values. Further, for each of these orderings, the random variable $Y_{st}Y_{ab}$ is non-zero if and only if the first sample at the three distinct indices is the same. Thus, we have that,

$$
\sum_{\substack{s<t\\a<b\\|\{s,t,a,b\}|=3}}\mathbb{E}[Y_{st}Y_{ab}] = 6\sum_{a<b<c}\sum_{\ell=1}^{k}p_a(\ell)p_b(\ell)p_c(\ell) := 6\sum_{a<b<c}\langle\mathbf{p}_a,\mathbf{p}_b,\mathbf{p}_c\rangle. \tag{5}
$$

Putting the above together with the remaining terms corresponding to $|\{s,t,a,b\}| = 3$ in the last summation, we get

$$
\begin{aligned}
\mathbb{E}\Big[\Big(\sum_{s<t}\hat{Y}_{st}\Big)^2\Big] &= \sum_{s<t}\langle\mathbf{p}_t,\mathbf{p}_s\rangle(1 - \langle\mathbf{p}_t,\mathbf{p}_s\rangle) + 6\sum_{a<b<c}\langle\mathbf{p}_a,\mathbf{p}_b,\mathbf{p}_c\rangle \\
&\quad - 2\sum_{a<b<c}[\langle\mathbf{p}_a,\mathbf{p}_b\rangle\langle\mathbf{p}_a,\mathbf{p}_c\rangle + \langle\mathbf{p}_a,\mathbf{p}_b\rangle\langle\mathbf{p}_b,\mathbf{p}_c\rangle + \langle\mathbf{p}_a,\mathbf{p}_c\rangle\langle\mathbf{p}_b,\mathbf{p}_c\rangle] \tag{6} \\
&\leq \sum_{s<t}\langle\mathbf{p}_t,\mathbf{p}_s\rangle + 6\sum_{a<b<c}\langle\mathbf{p}_a,\mathbf{p}_b,\mathbf{p}_c\rangle \tag{7} \\
&\leq \sum_{s<t}\langle\mathbf{p}_t,\mathbf{p}_s\rangle + 6\Big\|\sum_{t\in[T]}\mathbf{p}_t\Big\|_3^3.
\end{aligned}
$$

Substituting the above inequality, along with Equation (3) into Equation (2), we get the following bound for the variance of $Z$,

$$
\begin{aligned}
\mathrm{Var}[Z] &\leq \frac{2}{T^4}\Big(\sum_{t\in[T]}\|\mathbf{p}_t\|_2^2 + 4\sum_{s<t}\langle\mathbf{p}_t,\mathbf{p}_s\rangle + 24\Big\|\sum_{t\in[T]}\mathbf{p}_t\Big\|_3^3\Big) \\
&\leq \frac{4}{T^4}\Big\|\sum_{t\in[T]}\mathbf{p}_t\Big\|_2^2 + \frac{48}{T^4}\Big\|\sum_{t\in[T]}\mathbf{p}_t\Big\|_3^3 = \frac{4}{T^2}\Big\|\mathbf{P}_{\mathrm{avg}}\Big\|_2^2 + \frac{48}{T}\Big\|\mathbf{P}_{\mathrm{avg}}\Big\|_3^3. \tag{8}
\end{aligned}
$$

$\square$

We can now use the variance bound from the above lemma along with Chebyshev's inequality to prove Theorem 1.3,

*Proof of Theorem 1.3.* We divide the proof into two cases.

**Uniform case:** In this case, we are promised that the $\mathbf{p}_{\mathrm{avg}} = \mathbf{u}_k$. Therefore, it is immediate that $\mathbb{E}[Z] = \|\mathbf{p}_{\mathrm{avg}}\|_2^2 = \frac{1}{k}$ and $\|\mathbf{p}_{\mathrm{avg}}\|_3^3 = \frac{1}{k^2}$, which further gives us that,

$$\Pr\left[Z \geq \left(1 + \frac{\varepsilon^2}{3}\right)\frac{1}{k}\right] = \Pr\left[Z \geq \left(1 + \frac{\varepsilon^2}{3}\right)\mathbb{E}[Z]\right] \leq \frac{9}{\varepsilon^4 \mathbb{E}[Z]^2} \cdot \mathrm{Var}[Z] = \frac{9k^2}{\varepsilon^4} \cdot \mathrm{Var}[Z] ,$$
$$\leq \frac{9k^2}{\varepsilon^4}\left(\frac{4}{T^2 k} + \frac{48}{Tk^2}\right) = \frac{36k}{\varepsilon^4 T^2} + \frac{432}{T\varepsilon^4} .$$

In the first inequality, we used that $\mathbb{E}[Z] = 1/k$. In the second inequality, we used the Chebyshev's inequality. In the third and fourth inequality, we substituted the values of $\mathbb{E}[Z]$ and $\mathrm{Var}[Z]$ respectively; to upper bound the variance we used bounds from Equation (8). In the final inequality, we simplified the expression. Note that when $T \geq \frac{2600}{\varepsilon^4}$, the second term above is smaller than $1/6$, and for $T \geq \frac{216\sqrt{k}}{\varepsilon^2}$, the first term is smaller than $1/6$. Therefore, we have that, for $T = \max\left(\frac{2600}{\varepsilon^4}, \frac{216\sqrt{k}}{\varepsilon^2}\right) = O\left(\max\left(\frac{\sqrt{k}}{\varepsilon^2}, \frac{1}{\varepsilon^4}\right)\right)$,

$$\Pr\left[Z \geq \left(1 + \frac{\varepsilon^2}{3}\right)\frac{1}{k}\right] \leq 1/3 .$$

Therefore, $Z$ is smaller than $(1 + \varepsilon^2/3)\frac{1}{k}$ with probability at least $2/3$ in the uniform case, that is, when $\mathbf{p}_{\mathrm{avg}} = \mathbf{u}_k$.

**Far from uniform case:** In this case, we are promised that, $\|\mathbf{p}_{\mathrm{avg}} - \mathbf{u}_k\|_1 \geq \varepsilon$, which further implies $\mathbb{E}[Z] = \|\mathbf{p}_{\mathrm{avg}}\|_2^2 = \frac{1+\alpha^2}{k} \geq \frac{1}{k}(1 + \varepsilon^2)$, where $\alpha^2 \overset{\mathrm{def}}{=} k\|\mathbf{p}_{\mathrm{avg}} - \mathbf{u}_k\|_2^2 \geq \varepsilon^2$. In the following, we upper bound the probability that $Z$ takes a value smaller than $(1 + \varepsilon^2/3)\frac{1}{k}$.

$$\Pr\left[Z \leq \left(1 + \frac{\varepsilon^2}{3}\right)\frac{1}{k}\right] = \Pr\left[Z \leq \left(\frac{1 + \frac{\varepsilon^2}{3}}{1 + \alpha^2}\right)\left(\frac{1 + \alpha^2}{k}\right)\right] = \Pr\left[Z \leq \left(\frac{1 + \frac{\varepsilon^2}{3}}{1 + \alpha^2}\right)\mathbb{E}[Z]\right]$$
$$= \Pr\left[Z \leq \left(1 - \frac{\alpha^2 - \frac{\varepsilon^2}{3}}{1 + \alpha^2}\right)\mathbb{E}[Z]\right] \leq \Pr\left[Z \leq \left(1 - \frac{\alpha^2 - \frac{\alpha^2}{3}}{1 + \alpha^2}\right)\mathbb{E}[Z]\right]$$
$$= \Pr\left[Z \leq \left(1 - \frac{2\alpha^2}{3(1 + \alpha^2)}\right)\mathbb{E}[Z]\right] \leq \frac{9(1 + \alpha^2)^2}{4\alpha^4 \mathbb{E}[Z]^2} \cdot \mathrm{Var}[Z]$$
$$= \frac{9(1 + \alpha^2)^2}{4\alpha^4 \|\mathbf{p}_{\mathrm{avg}}\|_2^4} \cdot \mathrm{Var}[Z] \leq \frac{9(1 + \alpha^2)^2}{T^2 \alpha^4 \|\mathbf{p}_{\mathrm{avg}}\|_2^2} + \frac{108(1 + \alpha^2)^2}{T\alpha^4} \cdot \frac{\left\|\mathbf{p}_{\mathrm{avg}}\right\|_3^3}{\|\mathbf{p}_{\mathrm{avg}}\|_2^4}$$
$$= \frac{9k(1 + \alpha^2)}{T^2 \alpha^4} + \frac{108k^2}{T\alpha^4} \cdot \left\|\mathbf{p}_{\mathrm{avg}}\right\|_3^3. \tag{9}$$

In the first equality, we multiplied and divided by the term $(1 + \alpha^2)$. In the second equality, we used the definition of $\alpha$. In the third equality, we used $\frac{1 + \frac{\varepsilon^2}{3}}{1 + \alpha^2} = 1 - \frac{\alpha^2 - \frac{\varepsilon^2}{3}}{1 + \alpha^2}$. In the fourth inequality, we used $\alpha^2 \geq \varepsilon^2$ and in the fifth equality, we simplified the expression. In the sixth inequality, we used Chebyshev's inequality. In the seventh and eighth inequality, we used $\mathbb{E}[Z] = \|\mathbf{p}_{\mathrm{avg}}\|_2^2$ and substituted the bound on the variance of $Z$ from Equation (8). In the final expression, we used the definition of $\alpha$. To upper bound the above probability term, we next bound each of the terms in the final expression separately. The first term above is bounded by,

$$\frac{9k(1 + \alpha^2)}{T^2 \alpha^4} \leq \frac{9k(1 + \varepsilon^2)}{T^2 \varepsilon^4} \leq \frac{18k}{T^2 \varepsilon^4} . \tag{10}$$

In the first inequality, we used the fact that for $x > 0$, the function $\frac{1+x}{x^2}$ is decreasing in $x$ and also $\alpha^2 \geq \varepsilon^2$. The final inequality follows because $\varepsilon \leq 1$. To upper bound the second term in Equation (9), we first note that,

$$\|\mathbf{p}_{\mathrm{avg}}\|_3^3 = \|\mathbf{p}_{\mathrm{avg}} - \mathbf{u}_k + \mathbf{u}_k\|_3^3 = \sum_{i=1}^k ((\mathbf{p}_{\mathrm{avg}}(i) - \mathbf{u}_k(i)) + \mathbf{u}_k(i))^3$$

$$= \sum_{i=1}^{k} \Big[ (\mathbf{p}_{\mathrm{avg}}(i) - \mathbf{u}_k(i))^3 + \mathbf{u}_k(i)^3 + 3(\mathbf{p}_{\mathrm{avg}}(i) - \mathbf{u}_k(i))\mathbf{u}_k(i)\mathbf{p}_{\mathrm{avg}}(i) \Big]$$

$$= \|\mathbf{p}_{\mathrm{avg}} - \mathbf{u}_k\|_3^3 + \frac{1}{k^2} + \frac{3}{k}\Big( \sum_{i=1}^{k} \mathbf{p}_{\mathrm{avg}}(i)^2 - \frac{1}{k} \Big) = \|\mathbf{p}_{\mathrm{avg}} - \mathbf{u}_k\|_3^3 + \frac{1}{k^2} + \frac{3}{k}\|\mathbf{p}_{\mathrm{avg}} - \mathbf{u}_k\|_2^2$$

$$\leq \|\mathbf{p}_{\mathrm{avg}} - \mathbf{u}_k\|_2^3 + \frac{3}{k}\|\mathbf{p}_{\mathrm{avg}} - \mathbf{u}_k\|_2^2 + \frac{1}{k^2} = \frac{\alpha^3}{k^{3/2}} + \frac{3\alpha^2}{k^2} + \frac{1}{k^2}.$$

The first five equalities follow from simple algebraic manipulation. The sixth inequality holds because $\|\cdot\|_3 \leq \|\cdot\|_2$. In the final equality, we used the definition of $\alpha$. Thus, the second term is bounded as,

$$\frac{108k^2}{T\alpha^4} \cdot \Big\|\mathbf{p}_{\mathrm{avg}}\Big\|_3^3 \leq \frac{108k^2}{T\alpha^4} \cdot \Big( \frac{\alpha^3}{k^{3/2}} + \frac{3\alpha^2}{k^2} + \frac{1}{k^2} \Big) = \frac{108\sqrt{k}}{T\alpha} + \frac{324}{T\alpha^2} + \frac{108}{T\alpha^4}$$

$$\leq \frac{108\sqrt{k}}{T\varepsilon} + \frac{324}{T\varepsilon^2} + \frac{108}{T\varepsilon^4}, \tag{11}$$

In the final inequality, we used $\alpha \geq \varepsilon$. Substituting Equation (10) and Equation (11) in Equation (9), we get the following bound on the probability,

$$\Pr\Big[ Z \leq \Big(1 + \frac{\varepsilon^2}{3}\Big)\frac{1}{k} \Big] \leq \frac{18k}{T^2\varepsilon^4} + \frac{108\sqrt{k}}{T\varepsilon} + \frac{324}{T\varepsilon^2} + \frac{108}{T\varepsilon^4} \leq \frac{18k}{T^2\varepsilon^4} + \frac{432\sqrt{k}}{T\varepsilon^2} + \frac{108}{T\varepsilon^4}.$$

Note that for $T \geq \frac{2600\sqrt{k}}{\varepsilon^2}$, the sum of the first two terms is smaller than $1/6$, and for $T \geq \frac{648}{\varepsilon^4}$, the last term is smaller than $1/6$. Thus, we have that for $T = O\Big( \max\Big( \frac{\sqrt{k}}{\varepsilon^2}, \frac{1}{\varepsilon^4} \Big) \Big)$,

$$\Pr\Big[ Z \leq \Big(1 + \frac{\varepsilon^2}{3}\Big)\frac{1}{k} \Big] \leq 1/3.$$

Therefore, $Z$ is larger than $(1 + \varepsilon^2/3)\frac{1}{k}$ with probability at least $2/3$ in the far from uniform case, that is when $\|\mathbf{p}_{\mathrm{avg}} - \mathbf{u}_k\|_1 \geq \varepsilon$. In conclusion, for $T = O\Big( \frac{\sqrt{k}}{\varepsilon^2} + \frac{1}{\varepsilon^4} \Big)$, we can test uniformity, that is we can separate $\mathbf{p}_{\mathrm{avg}} = \mathbf{u}_k$ from $\|\mathbf{p}_{\mathrm{avg}} - \mathbf{u}_k\|_1 \geq \varepsilon$, for any $\varepsilon > 0$. We conclude the proof.

$\square$

**Remark 5** (Optimal sample complexity)**.** *We can compare the bound on the variance in Equation (8) to the similar bound in the tight analysis (e.g., Equation (2.9), Theorem 2.1 in Canonne (2022)) of the standard i.i.d setting. It would appear that we are missing a term of the form $-\frac{1}{T}\|\boldsymbol{p}_{\mathrm{avg}}\|_2^4$ which helps reduce the variance. In fact, if we had a term of this form, we would be able to shed the $O(1/\varepsilon^4)$ factor to obtain the optimal sample complexity of $O(\sqrt{k}/\varepsilon^2)$ even in the non-i.i.d setting. The main hurdle in the analysis above for getting this crucial variance-reducing term can be seen in Equation (6). In the i.i.d setting with one distribution $\boldsymbol{p}$, the terms of the form $\langle \boldsymbol{p}_a, \boldsymbol{p}_b\rangle\langle \boldsymbol{p}_a, \boldsymbol{p}_c\rangle$ are all the same and equal $\|\boldsymbol{p}\|_2^4$ (which is at least $\frac{1}{k^2}$). But in our setting, these terms can even be 0 (if $\boldsymbol{p}_a, \boldsymbol{p}_b, \boldsymbol{p}_c$ have disjoint supports), and hence, we cannot conveniently lower bound the contribution of these terms in reducing the variance. That is why, for lack of a better bound, we had to drop all these $\Big( 6\binom{T}{3}$ many!$\Big)$ terms in going from Equation (6) to Equation (7), which ultimately results in the stated sample complexity.*

## 4 Identity testing from non-identical samples

We now prove our result which reduces the problem of identity testing to uniformity testing. Such a reduction for the i.i.d. setting is already known (Goldreich, 2016) (see also (Canonne, 2022, Section 2.2.3)), and we adapt the same reduction for the setting of non-identical samples. In particular, we use the same reduction functions from that of the i.i.d case and show that they work even in the case of a sequence of changing distributions. A more formal statement of this reduction, along with its proof for a sequence of distributions is provided below.

**Lemma 1.4.** *Given an arbitrary reference distribution $\boldsymbol{q}$ supported on $\leq k$ elements, there exists a pair $(\Phi_{\boldsymbol{q}}, \Psi_{\boldsymbol{q}})$ that maps distributions and samples over $[k]$ to distributions and samples over $[4k]$, and satisfies the following,*

- *For a sequence of distributions $\boldsymbol{q}_1, \ldots, \boldsymbol{q}_T$ such that $\mathrm{avg}(\boldsymbol{q}_1, \ldots, \boldsymbol{q}_T) = \boldsymbol{q}$, the map $\Phi_{\boldsymbol{q}}$ satisfies,*

$$\mathrm{avg}(\Phi_{\boldsymbol{q}}(\boldsymbol{q}_1), \ldots, \Phi_{\boldsymbol{q}}(\boldsymbol{q}_T)) = \mathrm{Unif}(4k) .$$

- *For two sequences of distributions $\boldsymbol{p}_1^1, \ldots, \boldsymbol{p}_T^1$ and $\boldsymbol{p}_1^2, \ldots, \boldsymbol{p}_T^2$, the map $\Phi_{\boldsymbol{q}}$ satisfies,*

$$\mathrm{d_{TV}}(\mathrm{avg}(\Phi_{\boldsymbol{q}}(\boldsymbol{p}_1^1), \ldots, \Phi_{\boldsymbol{q}}(\boldsymbol{p}_T^1)), \mathrm{avg}(\Phi_{\boldsymbol{q}}(\boldsymbol{p}_1^2), \ldots, \Phi_{\boldsymbol{q}}(\boldsymbol{p}_T^2))) \geq \frac{1}{4} \cdot \mathrm{d_{TV}}(\mathrm{avg}(\boldsymbol{p}^1, \ldots, \boldsymbol{p}_T^1), \mathrm{avg}(\boldsymbol{p}^2, \ldots, \boldsymbol{p}_T^2)) .$$

*Proof.* We divide the definition of our maps $(\Phi_{\mathbf{q}}, \Psi_{\mathbf{p}})$ into three parts:

$$\Phi_{\mathbf{q}} \overset{\mathrm{def}}{=} \Phi_{\mathbf{q}}^1 \circ \Phi_{\mathbf{q}}^2 \circ \Phi_{\mathbf{q}}^3 .$$

$$\Psi_{\mathbf{q}} \overset{\mathrm{def}}{=} \Psi_{\mathbf{q}}^1 \circ \Psi_{\mathbf{q}}^2 \circ \Psi_{\mathbf{q}}^3 .$$

We first define the map $\Phi_{\mathbf{q}}$, which we divide into three parts $\Phi_{\mathbf{q}}^3, \Phi_{\mathbf{q}}^2$ and $\Phi_{\mathbf{q}}^1$. The map $\Phi_{\mathbf{q}}^3 : \Delta_k \to \Delta_k$ is defined as follows,

$$\Phi_{\mathbf{q}}^3(\mathbf{p}) = \frac{1}{2}\mathbf{p} + \frac{1}{2}\mathrm{Unif}(k) .$$

Let $k' = 4k$ and $r = \Phi_{\mathbf{q}}^3(\mathbf{q}) = \frac{1}{2}\mathbf{q} + \frac{1}{2}\mathrm{Unif}(k)$. The map $\Phi_{\mathbf{q}}^2 : \Delta_k \to \Delta_{k+1}$ is defined as follows,

$$\Phi_{\mathbf{q}}^2(\mathbf{p})(i) = \begin{cases} \frac{\lfloor k'r(i) \rfloor}{k'r(i)} \cdot \mathbf{p}(i) & \text{for all } i \in [k] \\ 1 - \sum_{j \in [k]} \frac{\lfloor k'r(j) \rfloor}{k'r(j)} \cdot \mathbf{p}(j) & \text{for } i = k+1 . \end{cases} \tag{12}$$

Let $s = \Phi_{\mathbf{q}}^2(r) = \Phi_{\mathbf{q}}^2(\Phi_{\mathbf{q}}^3(\mathbf{q}))$ and define $k_i = k' \cdot s(i)$ for $i \in [k+1]$. Let $S_1, \ldots, S_{k+1}$ be any partition of $[k']$ such that $|S_i| = k_i$ for all $i \in [k+1]$. Then the map $\Phi_{\mathbf{q}}^1 : \Delta_{k+1} \to \Delta_{4k}$ is defined as follows,

$$\Phi_{\mathbf{q}}^1(\mathbf{p})(i) = \frac{1}{k'} \sum_{j \in [k+1]} \frac{\mathbf{p}(j)}{s(j)} \mathbf{1}(i \in S_j) .$$

Using these definitions, we now prove the two claims stated in the theorem.

**Claim 1:** For any $t \in [T]$, let,

$$x_t = \Phi_{\mathbf{q}}^3(\mathbf{q}_t), \ y_t = \Phi_{\mathbf{q}}^2(x_t) \text{ and } z_t = \Phi_{\mathbf{q}}^1(y_t) .$$

In the following, we show that,

$$\mathrm{avg}(\Phi_{\mathbf{q}}(\mathbf{q}^1), \ldots, \Phi_{\mathbf{q}}(\mathbf{q}_T))(i) = \mathrm{avg}(z_1, \ldots, z_t) = \mathbf{u}_{4k} .$$

We prove the above statement entry-wise, and we consider $i$th entry of the average distribution. We divide the proof into two cases. Let $i$ be such that $i \in S_j$ for some $j \in [k]$, then note that,

$$\mathrm{avg}(\Phi_{\mathbf{q}}(\mathbf{q}^1), \ldots, \Phi_{\mathbf{q}}(\mathbf{q}_T))(i) = \frac{1}{T} \sum_{t \in [T]} z_t(i) = \frac{1}{T} \sum_{t \in [T]} \frac{1}{k'} \sum_{j' \in [k+1]} \frac{y_t(j')}{s(j')} \mathbf{1}(i \in S_{j'}),$$

$$= \frac{1}{Tk'} \sum_{t \in [T]} \frac{y_t(j)}{s(j)} = \frac{1}{Tk'} \sum_{t \in [T]} \frac{\frac{\lfloor k'r(j) \rfloor}{k'r(j)} \cdot x_t(j)}{s(j)},$$

$$= \frac{1}{Tk'} \sum_{t \in [T]} \frac{\frac{\lfloor k'r(j) \rfloor}{k'r(j)} \cdot x_t(j)}{\frac{\lfloor k'r(j) \rfloor}{k'}} = \frac{1}{Tk'} \sum_{t \in [T]} \frac{x_t(j)}{r(j)},$$

$$= \frac{1}{Tk'} \sum_{t \in [T]} \frac{\frac{1}{2}\mathbf{q}_t(j) + \frac{1}{2}\text{Unif}(k)(j)}{r(j)}$$

$$= \frac{1}{Tk'} T \cdot \frac{\frac{1}{2}\mathbf{q}(j) + \frac{1}{2}\text{Unif}(k)(j)}{r(j)}$$

$$= \frac{1}{k'}$$

In the second equality, we substituted the value $z_t = \Phi_{\mathbf{q}}^1(y_t)$. In the third equality, we used the assumption that $i \in S_j$. In the fourth and fifth inequality, we substituted the values of $y_t$ and $s$ respectively. We simplified the expression in the sixth equality and substituted the value of $x_t$ in the seventh equality. In the eighth equality, we used the definition of $\mathbf{q} = \text{avg}(\mathbf{q}_1, \dots, \mathbf{q}_T)$. In the final equality, we used the definition of $r$.

Let $i \in S_{k+1}$, then note that,

$$\text{avg}(\Phi_{\mathbf{q}}(\mathbf{q}^1), \dots, \Phi_{\mathbf{q}}(\mathbf{q}_T))(i) = \frac{1}{T} \sum_{t \in [T]} z_t(i) = \frac{1}{T} \sum_{t \in [T]} \frac{1}{k'} \sum_{j' \in [k+1]} \frac{y_t(j')}{s(j')} \mathbf{1}(i \in S_{j'}),$$

$$= \frac{1}{Tk'} \sum_{t \in [T]} \frac{y_t(k+1)}{s(k+1)} = \frac{1}{Tk'} \sum_{t \in [T]} \frac{1 - \sum_{j \in [k]} \frac{\lfloor k' r(j) \rfloor}{k' r(j)} \cdot x_t(j)}{s(k+1)},$$

$$= \frac{1}{Tk'} \frac{1}{s(k+1)} \sum_{t \in [T]} \left( 1 - \sum_{j \in [k]} \frac{\lfloor k' r(j) \rfloor}{k' r(j)} \cdot x_t(j) \right),$$

$$= \frac{1}{Tk'} \frac{1}{s(k+1)} \cdot T \cdot \left( 1 - \sum_{j \in [k]} \frac{\lfloor k' r(j) \rfloor}{k' r(j)} \cdot \sum_{t \in [T]} \frac{x_t(j)}{T} \right),$$

$$= \frac{1}{k'} \frac{1}{s(k+1)} \left( 1 - \sum_{j \in [k]} \frac{\lfloor k' r(j) \rfloor}{k' r(j)} \cdot \left( \frac{1}{2}\mathbf{q}(j) + \frac{1}{2}\text{Unif}(k)(j) \right) \right),$$

$$= \frac{1}{k'} \frac{1}{s(k+1)} \left( 1 - \sum_{j \in [k]} \frac{\lfloor k' r(j) \rfloor}{k'} \right) = \frac{1}{k'} \frac{1}{s(k+1)} \cdot s(k+1)$$

$$= \frac{1}{k'}.$$

In the second equality, we substituted the value $z_t = \Phi_{\mathbf{q}}^1(y_t)$. In the third equality, we used the assumption that $i \in S_{k+1}$. In the fourth equality, we substituted the values of $y_t$ and simplified the expression in the fifth and sixth equality. In the seventh equality, substituted the value of $x_t$ and used the definition of $\mathbf{q} = \text{avg}(\mathbf{q}_1, \dots, \mathbf{q}_T)$. In the eighth and ninth equality, we used the definition of $r$ and $s$ respectively.

Combining the above two derivations, we have that, for all $i \in [k+1]$,

$$\text{avg}(\Phi_{\mathbf{q}}(\mathbf{q}^1), \dots, \Phi_{\mathbf{q}}(\mathbf{q}_T))(i) = \frac{1}{k'} = \frac{1}{4k}.$$

Therefore,

$$\text{avg}(\Phi_{\mathbf{q}}(\mathbf{q}^1), \dots, \Phi_{\mathbf{q}}(\mathbf{q}_T)) = \mathbf{u}_{4k}.$$

**Claim 2:** For any $t \in [T]$, let,

$$x_t^a = \Phi_{\mathbf{q}}^3(\mathbf{p}_t^a), y_t^a = \Phi_{\mathbf{q}}^2(x_t^a) \text{ and } z_t^a = \Phi_{\mathbf{q}}^1(y_t^a) \text{ for } a \in \{1, 2\}.$$

In the remainder of the proof, we show that

$$d_{\text{TV}}(\text{avg}(z_1^1, \dots, z_T^1), \text{avg}(z_1^2, \dots, z_T^2)) = d_{\text{TV}}(\text{avg}(\Phi(\mathbf{p}_1^1), \dots, \Phi(\mathbf{p}_T^1)), \text{avg}(\Phi(\mathbf{p}_1^2), \dots, \Phi(\mathbf{p}_T^1)))$$

$$\geq \frac{1}{4} d_{\mathrm{TV}}(\mathrm{avg}(\mathbf{p}_1^1, \ldots, \mathbf{p}_T^1), \mathrm{avg}(\mathbf{p}_1^2, \ldots, \mathbf{p}_T^2)) \ .$$

It is immediate that,

$$d_{\mathrm{TV}}(\mathrm{avg}(z_1^1, \ldots, z_T^1), \mathrm{avg}(z_1^2, \ldots, z_T^2)) = d_{\mathrm{TV}}(\mathrm{avg}(y_1^1, \ldots, y_T^1), \mathrm{avg}(y_1^2, \ldots, y_T^2)) \ .$$

Furthermore, note that for $i \in [k]$,

$$\mathrm{avg}(y_1^a, \ldots, y_T^a)(i) = \frac{1}{T} \sum_t \frac{\lfloor k' r(i) \rfloor}{k' r(i)} \cdot x_t^a(i) = \frac{\lfloor k' r(i) \rfloor}{k' r(i)} \cdot (\frac{1}{2} \mathbf{p}_{\mathrm{avg}}^a(i) + \frac{1}{2} \mathrm{Unif}(k)(i))$$

and,

$$\mathrm{avg}(y_1^a, \ldots, y_T^a)(k+1) = \frac{1}{T} \sum_t \left( 1 - \sum_{j \in [k]} \frac{\lfloor k' r(j) \rfloor}{k' r(j)} \cdot x_t^a(j) \right)$$

$$= 1 - \sum_{j \in [k]} \frac{\lfloor k' r(j) \rfloor}{k' r(j)} \cdot (\frac{1}{2} \mathbf{p}_{\mathrm{avg}}^a(j) + \frac{1}{2} \mathrm{Unif}(k)(j)) \ .$$

In the following we bound $d_{\mathrm{TV}}(\mathrm{avg}(y_1^1, \ldots, y_T^1), \mathrm{avg}(y_1^2, \ldots, y_T^2))$, which in turn bounds the quantity $d_{\mathrm{TV}}(\mathrm{avg}(z_1^1, \ldots, z_T^1), \mathrm{avg}(z_1^2, \ldots, z_T^2))$.

$$d_{\mathrm{TV}}(\mathrm{avg}(y_1^1, \ldots, y_T^1), \mathrm{avg}(y_1^2, \ldots, y_T^2))$$

$$= \sum_{i \in [k]} \left| \frac{\lfloor k' r(i) \rfloor}{k' r(i)} \cdot (\frac{1}{2} \mathbf{p}_{\mathrm{avg}}^1(i) + \frac{1}{2} \mathrm{Unif}(k)(i)) - \frac{\lfloor k' r(i) \rfloor}{k' r(i)} \cdot (\frac{1}{2} \mathbf{p}_{\mathrm{avg}}^2(i) + \frac{1}{2} \mathrm{Unif}(k)(i)) \right|,$$

$$+ \left| 1 - \sum_{j \in [k]} \frac{\lfloor k' r(j) \rfloor}{k' r(j)} \cdot (\frac{1}{2} \mathbf{p}_{\mathrm{avg}}^1(j) + \frac{1}{2} \mathrm{Unif}(k)(j)) - \right.$$

$$\left. \left( 1 - \sum_{j \in [k]} \frac{\lfloor k' r(j) \rfloor}{k' r(j)} \cdot (\frac{1}{2} \mathbf{p}_{\mathrm{avg}}^2(j) + \frac{1}{2} \mathrm{Unif}(k)(j)) \right) \right|,$$

$$= \sum_{i \in [k]} \frac{\lfloor k' r(i) \rfloor}{k' r(i)} \frac{1}{2} \left| \mathbf{p}_{\mathrm{avg}}^1(i) - \mathbf{p}_{\mathrm{avg}}^2(i) \right| + \sum_{j \in [k]} \frac{\lfloor k' r(j) \rfloor}{k' r(j)} \frac{1}{2} \left| \mathbf{p}_{\mathrm{avg}}^1(j) - \mathbf{p}_{\mathrm{avg}}^2(j) \right|,$$

$$= \sum_{i \in [k]} \frac{\lfloor k' r(i) \rfloor}{k' r(i)} \left| \mathbf{p}_{\mathrm{avg}}^1(i) - \mathbf{p}_{\mathrm{avg}}^2(i) \right| \geq \frac{1}{2} \sum_{i \in [k]} \left| \mathbf{p}_{\mathrm{avg}}^1(i) - \mathbf{p}_{\mathrm{avg}}^2(i) \right| \ .$$

In the first equality, we substituted the values of $\mathrm{avg}(y_1^a, \ldots, y_T^a)$ for $a \in \{1, 2\}$ that were computed earlier. In the second and third equality, we simplified the expression. The last inequality follows because $r(i) \geq \frac{1}{2k}$, $k' r(i) \geq 2$ and $\lfloor k' r(i) \rfloor \geq k' r(i) - 1$. Putting it all together, we have that,

$$d_{\mathrm{TV}}(\mathrm{avg}(\Phi(\mathbf{p}_1^1), \ldots, \Phi(\mathbf{p}_T^1)), \mathrm{avg}(\Phi(\mathbf{p}_1^2), \ldots, \Phi(\mathbf{p}_T^1))$$

$$\geq \frac{1}{4} d_{\mathrm{TV}}(\mathrm{avg}(\mathbf{p}_1^1, \ldots, \mathbf{p}_T^1), \mathrm{avg}(\mathbf{p}_1^2, \ldots, \mathbf{p}_T^2)) \ ,$$

and we conclude the proof for this case.

**Sampling:** In the remainder of the proof, we state the maps $\Psi^3, \Psi^2, \Psi^1$ which help us generate the samples from the mapped distributions. The definitions of these sampling maps are pretty straightforward and we state them below to conclude the proof.

$\Psi_{\mathbf{q}}^3$ : Given $i \in [k]$, return $i$ with probability $1/2$ and a uniformly random element of $[k]$ otherwise.

$\Psi_{\mathbf{q}}^2$ : Given $i \in [k]$, return $i$ with probability $\frac{\lfloor k' r(i) \rfloor}{k' r(i)}$ and $k+1$ otherwise.

$\Psi_{\mathbf{q}}^1$ : Given $i \in [k+1]$, return a uniformly random element from $S_i$. $\qquad \square$

# 5    Lower bound for pooling-based estimators

In this section, we prove the indistinguishability result for pooling-based estimators given in Theorem 1.6. First, we observe that uniformity of the average distribution is a symmetric property, and thus, labels of the samples do not matter for a testing algorithm. Thus, without loss of generality, we can assume that the testing algorithm takes as input the pooled multiset of samples and operates only on the *fingerprint* of the pooled multiset (refer to Appendix E for a formal discussion on this). Here, the fingerprint is the count of elements that occur exactly zero times, once, twice etc. In what follows, we will construct two sequences of distributions: one whose average is the uniform distribution and another whose average is far from uniform. However, we show that the total variation distance between the distributions of their corresponding fingerprints is small, implying impossibility of uniformity testing with pooled samples. Below, we describe the key proof steps:

**Step 1:**   We construct two sequences of distributions **a** and **b** (Definition 8) such that avg(**a**) is uniform and avg(**b**) is far from uniform (Claim 5.1). Moreover, these sequences are chosen in such a way that when we draw 2 samples from each distribution in the sequence, no element can be observed more than 4 times in the pooled multiset. The building blocks needed for defining these sequences of distributions are introduced in Lemma 6.

**Step 2:**   Next, we show that when we draw 2 samples from each distribution in the sequences, the fingerprints of the pooled multisets have the same mean (Claim 5.2), and similar covariances (Claim 5.3). A minor detail here is that we show this for the collision counts instead of fingerprints, but this is not an issue, as one can use an invertible linear transform to relate the two (Appendix D).

**Step 3:**   We want to eventually show that the distributions of fingerprints are close, and hence cannot be distinguished. For this, we will first show that the distributions of these fingerprints are close, in total variation distance, to the multivariate Gaussian distributions $G_{\mathbf{a}}$ and $G_{\mathbf{b}}$ of corresponding mean and covariance, that have been discretized by rounding all probability mass to the nearest point in the integer lattice (Lemma 11). We show this via a CLT (Lemma 10). One slight complication in proving this CLT is that in our setting, the fingerprints no longer correspond to a generalized multinomial distribution, and hence we cannot directly leverage multivariate CLTs for such distributions (e.g., Valiant & Valiant (2011)). Instead, we will show that the distribution of fingerprints is represented as a sum of independent samples, supported on the all-zero vector, the basis vectors, and the vector $[2, 0, 0, \ldots]$. Our proof essentially "splits" each distribution into the component supported on the zero vector and the basis vectors, and a component supported on the zero vector and the $[2, 0, 0, \ldots]$ vector. The portion supported on the basis vectors *is* a generalized multinomial, and hence is close to the corresponding discretized Gaussian by the CLT in Valiant & Valiant (2011), and the remaining portion is simply a binomial, scaled by a factor of 2. We then argue that the convolution of these distributions is close to a discretized Gaussian.

**Step 4:**   Having shown that the distributions of fingerprints are close to corresponding discretized multivariate Gaussian distributions, we are left with the tractable task of showing that the TV distance between these two (discretized) Gaussians is small. This is shown in Lemma 12 and relies on certain technical results proved in Lemma 16 and Lemma 17. Here, we use the fact that the two collision-count distributions have the same mean and similar covariances as discussed in Step 2 above. Finally, a triangle inequality yields the desired result (Lemma 13).

We now proceed towards the proof details of the above steps. We start by defining some distributions which are the building blocks for our hard instance. We will tile up these distributions over disjoint supports to form the overall hard instance

**Lemma 6** (Building block). *There exist distributions $\boldsymbol{p}_1, \boldsymbol{p}_2, \boldsymbol{q}_1, \boldsymbol{q}_2, \boldsymbol{r}_1, \boldsymbol{r}_2, \boldsymbol{s}_1, \boldsymbol{s}_2$, each supported on m elements, satisfying the following:*

*(1)* $\mathrm{avg}(\boldsymbol{p}_1, \boldsymbol{p}_2) = \mathrm{avg}(\boldsymbol{r}_1, \boldsymbol{r}_2) = \mathrm{Unif}(m).$

(2) $\mathrm{d_{TV}}(\mathrm{avg}(\boldsymbol{p}_1, \boldsymbol{p}_2), \mathrm{avg}(\boldsymbol{q}_1, \boldsymbol{q}_2)) \geq \frac{1}{16\sqrt{2}}$ and
$\mathrm{d_{TV}}(\mathrm{avg}(\boldsymbol{r}_1, \boldsymbol{r}_2), \mathrm{avg}(\boldsymbol{s}_1, \boldsymbol{s}_2)) \geq \frac{1}{16\sqrt{2}}$ .

(3) *Let $FP(\boldsymbol{p}) = [FP(\boldsymbol{p})_2, FP(\boldsymbol{p})_3, FP(\boldsymbol{p})_4]$ be the random variable representing the tuple of counts of elements observed exactly 2 times, 3 times and 4 times respectively, when we draw $c = 2$ samples i.i.d each from $\boldsymbol{p}_1$ and $\boldsymbol{p}_2$ (i.e. we obtain a total 4 samples). Note that $FP(\boldsymbol{p})$ is supported on $[0, 0, 0], [1, 0, 0], [0, 1, 0], [0, 0, 1], [2, 0, 0]$. Similarly, define $FP(\boldsymbol{q}), FP(\boldsymbol{r}), FP(\boldsymbol{s})$. Furthermore, let $C_{\boldsymbol{p}} = [c_2^{\boldsymbol{p}}, c_3^{\boldsymbol{p}}, c_4^{\boldsymbol{p}}]$ be the 3-dimensional vector of 2-way, 3-way and 4-way collisions observed when we draw $c = 2$ samples i.i.d each from $\boldsymbol{p}_1$ and $\boldsymbol{p}_2$[1]. Similarly define $C_{\boldsymbol{q}}, C_{\boldsymbol{r}}, C_{\boldsymbol{s}}$. There exists an $\alpha \in (0, 1)$ such that,*

$$\mathbb{E}[\alpha C_{\boldsymbol{p}} + (1 - \alpha)C_{\boldsymbol{r}}] = \mathbb{E}[\alpha C_{\boldsymbol{q}} + (1 - \alpha)C_{\boldsymbol{s}}].$$
$$\mathbb{E}[\alpha FP(\boldsymbol{p}) + (1 - \alpha)FP(\boldsymbol{r})]$$
$$= \mathbb{E}[\alpha FP(\boldsymbol{q}) + (1 - \alpha)FP(\boldsymbol{s})] .$$

*Proof.* First, we describe the distributions $\mathbf{p}_1, \mathbf{p}_2, \mathbf{r}_1, \mathbf{r}_2$. Let $\mathbf{p}_1 = \mathbf{p}_2 \in \Delta_m$ be the uniform distribution on $m$ elements. Let $\mathbf{r}_1 \in \Delta_m$ be such that it assigns mass $\frac{2}{m}$ each on the first $\frac{m}{2}$ elements of $[m]$. Let $\mathbf{r}_2 \in \Delta_m$ be such that it assigns mass $\frac{2}{m}$ each on the last $\frac{m}{2}$ elements of $[m]$. This is illustrated more clearly in Figure 1a. It is clear that $\mathrm{avg}(\mathbf{p}_1, \mathbf{p}_2) = \mathrm{avg}(\mathbf{r}_1, \mathbf{r}_2) = \mathrm{Unif}(m)$, and hence part (1) of the lemma holds.

Next, we describe the distributions $\mathbf{q}_1, \mathbf{q}_2, \mathbf{s}_1, \mathbf{s}_2$. Set $\varepsilon = \frac{1}{\sqrt{2}}$. Let $\mathbf{q}_1 \in \Delta_m$ be such that $\mathbf{q}_1$ has a mass $\frac{1+\varepsilon}{m}$ each on the first $\frac{m}{2}$ elements, and a mass $\frac{1-\varepsilon}{m}$ each on the remaining $\frac{m}{2}$ elements. Next, consider $\mathbf{q}_2 \in \Delta_m$ defined as follows: Of the first $\frac{m}{2}$ elements, $\mathbf{q}_2$ has mass $\frac{1-\varepsilon}{m}$ each on the first $\frac{7}{8}$th fraction, and a mass $\frac{1+\varepsilon}{m}$ each on the remaining $\frac{1}{8}$th fraction. Similarly, of the latter $\frac{m}{2}$ elements, $\mathbf{q}_2$ has a mass $\frac{1+\varepsilon}{m}$ each on the first $\frac{7}{8}$th fraction, and a mass $\frac{1-\varepsilon}{m}$ each on the remaining $\frac{1}{8}$th fraction. Further, we set $\mathbf{s}_1 = \mathbf{q}_1$ and $\mathbf{s}_2 = \mathbf{q}_2$. This is illustrated more clearly in Figure 1b.

We can readily see that $\|\mathrm{avg}(\mathbf{p}_1, \mathbf{p}_2) - \mathrm{avg}(\mathbf{q}_1, \mathbf{q}_2)\|_1 = \|\mathrm{avg}(\mathbf{r}_1, \mathbf{r}_2) - \mathrm{avg}(\mathbf{s}_1, \mathbf{s}_2)\|_1 = \frac{\varepsilon}{8}$. Recalling that $\varepsilon = \frac{1}{\sqrt{2}}$, part (2) of the lemma holds.

Now, we prove the last part. Fix $\beta = \frac{7}{8}$. Let $c_2^{\mathbf{q}}, c_3^{\mathbf{q}}, c_4^{\mathbf{q}}$ denote the number of 2-way, 3-way and 4-way collisions respectively when we draw 2 samples from $\mathbf{q}_1$ and 2 samples from $\mathbf{q}_2$.

$$\mathbb{E}[c_2^{\mathbf{q}}] = 2\left[\frac{m}{2} \cdot \left(\frac{1+\varepsilon}{m}\right)^2 + \frac{m}{2} \cdot \left(\frac{1-\varepsilon}{m}\right)^2\right]$$
$$+ 4\left[\beta m \cdot \left(\frac{1+\varepsilon}{m}\right)\left(\frac{1-\varepsilon}{m}\right) + \frac{(1-\beta)m}{2} \cdot \left(\left(\frac{1+\varepsilon}{m}\right)^2 + \left(\frac{1-\varepsilon}{m}\right)^2\right)\right] = \frac{11}{2m}.$$
$$\mathbb{E}[c_3^{\mathbf{q}}] = 4\left[\frac{(1-\beta)m}{2} \cdot \left(\left(\frac{1+\varepsilon}{m}\right)^3 + \left(\frac{1-\varepsilon}{m}\right)^3\right)\right.$$
$$\left. + \frac{\beta m}{2}\left(\left(\frac{1+\varepsilon}{m}\right)^2\left(\frac{1-\varepsilon}{m}\right) + \left(\frac{1+\varepsilon}{m}\right)\left(\frac{1-\varepsilon}{m}\right)^2\right)\right] = \frac{3}{m^2}.$$
$$\mathbb{E}[c_4^{\mathbf{q}}] = \frac{(1-\beta)m}{2}\left[\left(\frac{1+\varepsilon}{m}\right)^4 + \left(\frac{1-\varepsilon}{m}\right)^4\right] + \beta m\left[\left(\frac{1+\varepsilon}{m}\right)^2\left(\frac{1-\varepsilon}{m}\right)^2\right] = \frac{3}{4m^3}.$$

Since $\mathbf{q}_1 = \mathbf{s}_1$ and $\mathbf{q}_2 = \mathbf{s}_2$, for any $\alpha \in (0, 1)$,

$$\mathbb{E}\begin{bmatrix} c_2^{\mathbf{q}} \\ c_3^{\mathbf{q}} \\ c_4^{\mathbf{q}} \end{bmatrix} = \mathbb{E}\begin{bmatrix} c_2^{\mathbf{s}} \\ c_3^{\mathbf{s}} \\ c_4^{\mathbf{s}} \end{bmatrix} = \mathbb{E}\begin{bmatrix} \alpha c_2^{\mathbf{q}} + (1-\alpha)c_2^{\mathbf{s}} \\ \alpha c_3^{\mathbf{q}} + (1-\alpha)c_3^{\mathbf{s}} \\ \alpha c_4^{\mathbf{q}} + (1-\alpha)c_4^{\mathbf{s}} \end{bmatrix} = \begin{bmatrix} 11/2m \\ 3/m^2 \\ 3/4m^3 \end{bmatrix} . \tag{13}$$

---

[1]$m$-way collisions also formally defined in Appendix D.

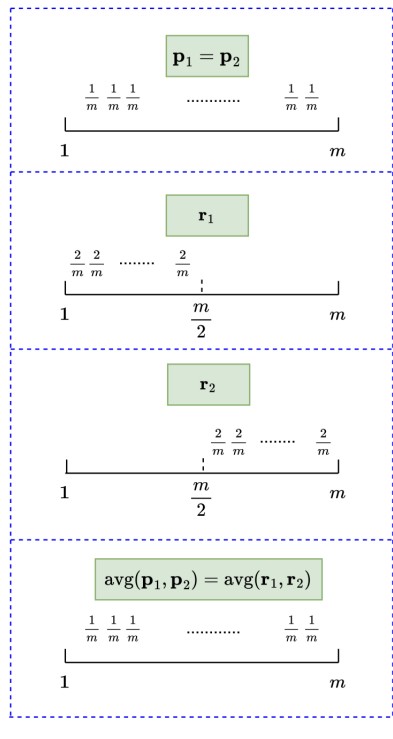 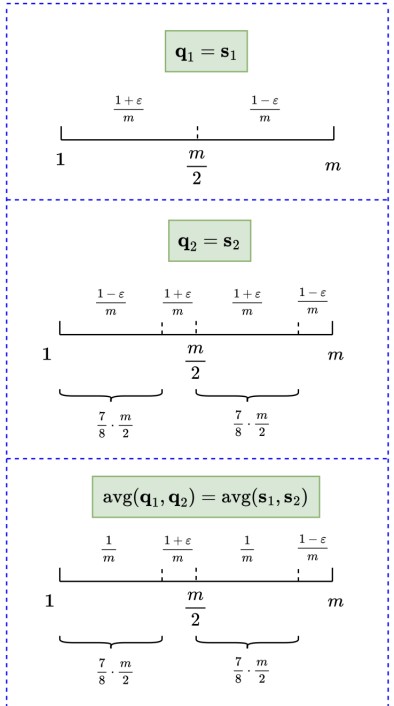

(a) Average is uniform

(b) Average far from uniform

Figure 1: Building block for the hard instance defined in Definition 8.

Now, set $\alpha = \frac{3}{4}$. Say we toss a coin with bias $\alpha$. If we observe heads, we will draw two samples each from $\mathbf{p}_1$ and $\mathbf{p}_2$. If we observe tails, we will draw two samples each from $\mathbf{r}_1$ and $\mathbf{r}_2$. We can calculate the expected number of 2/3/4-way collisions in the 4 samples obtained via this process.

$$\mathbb{E}[\alpha c_2^{\mathbf{p}} + (1-\alpha)c_2^{\mathbf{r}}] = \alpha \binom{4}{2}\frac{1}{m} + (1-\alpha)\cdot\left(\frac{2}{m} + \frac{2}{m}\right) = \frac{2\alpha + 4}{m} = \frac{11}{2m}.$$

$$\mathbb{E}[\alpha c_3^{\mathbf{p}} + (1-\alpha)c_3^{\mathbf{r}}] = \frac{4\alpha}{m^2} = \frac{3}{m^2}.$$

$$\mathbb{E}[\alpha c_4^{\mathbf{p}} + (1-\alpha)c_4^{\mathbf{r}}] = \frac{\alpha}{m^3} = \frac{3}{4m^3}.$$

These are precisely the expressions for the expectations in Equation (13). Finally, the collision counts are related linearly to the fingerprint (Lemma 14), and this completes the proof of the part (3). $\qquad\square$

**Remark 7.** *Note that as we draw only 4 samples from the building block given above, the number of elements that occur exactly zero times and exactly once are a deterministic function of the number of elements that occur exactly 2,3 and 4 times. Furthermore, since we will tile up the above instance over disjoint supports of $m$ elements, no element will ever occur more than 4 times. Thus, we restrict our attention only to the fingerprint of elements appearing 2,3 and 4 times.*

Now, we propose two sequences of distributions $\mathbf{a} = (\mathbf{a}_1, \cdots, \mathbf{a}_T)$ and $\mathbf{b} = (\mathbf{b}_1, \cdots, \mathbf{b}_T)$ such that $\mathrm{avg}(\mathbf{a})$ is uniform and $\mathrm{avg}(\mathbf{b})$ is far from uniform.

**Definition 8.** *Divide the support $[k]$ into $n$ disjoint blocks, each of size $m$, so that $k = mn$. Each block corresponds to the support of 2 distributions in a sequence, so that the total number of distributions $T = 2n$. Assume that $k$ is large enough and $m$ is a large constant independent of $k$. $k$ being large enough and $m$ being a constant ensures that $T = \Theta(k)$ is also large enough.*

*First, we specify each block in the first $\alpha n$ many blocks (where $\alpha = \frac{3}{4}$ and $n$ will be some multiple of 4) — fix the $i^{th}$ such block of size $m$, where $1 \leq i \leq \alpha n$. Let $\boldsymbol{p}_1^i, \boldsymbol{p}_2^i, \boldsymbol{q}_1^i, \boldsymbol{q}_2^i$ correspond to distributions $\boldsymbol{p}_1, \boldsymbol{p}_2, \boldsymbol{q}_1, \boldsymbol{q}_2$ respectively (as defined in Lemma 6), supported on this block. For the first distribution supported on this block, that is for $t = 2i - 1$, we will set $\boldsymbol{a}_{2i-1} = \boldsymbol{p}_1^i, \boldsymbol{b}_{2i-1} = \boldsymbol{q}_1^i$. For the second distribution supported on this block, that is for $t = 2i$, we will set $\boldsymbol{a}_{2i} = \boldsymbol{p}_2^i, \boldsymbol{b}_{2i} = \boldsymbol{q}_2^i$.*

*Now, we specify each block in the remaining $(1 - \alpha)n$ many blocks — fix the $i^{th}$ such block of size $m$, where $\alpha n + 1 \leq i \leq n$. Let $\boldsymbol{r}_1^i, \boldsymbol{r}_2^i, \boldsymbol{s}_1^i, \boldsymbol{s}_2^i$ correspond to distributions $\boldsymbol{r}_1, \boldsymbol{r}_2, \boldsymbol{s}_1, \boldsymbol{s}_2$ respectively (as defined in Lemma 6), supported on this block. For the first distribution supported on this block, that is for $t = 2i - 1$, we will set $\boldsymbol{a}_{2i-1} = \boldsymbol{r}_1^i, \boldsymbol{b}_{2i-1} = \boldsymbol{s}_1^i$. For the second distribution supported on this block, that is for $t = 2i$, we will set $\boldsymbol{a}_{2i} = \boldsymbol{r}_2^i, \boldsymbol{b}_{2i} = \boldsymbol{s}_2^i$.*

These sequences $\mathbf{a}$ and $\mathbf{b}$ are the entry-point to Step 1 in the proof steps described above.

**Claim 5.1.** *Let $\boldsymbol{a}$ and $\boldsymbol{b}$ be the sequences constructed in Definition 8. $\mathrm{avg}(\boldsymbol{a})$ is uniform on $[k] = [mn]$ and $\mathrm{avg}(\boldsymbol{b})$ is far from uniform with $\mathrm{d_{TV}}(\mathrm{Unif}(k), \mathrm{avg}(\boldsymbol{b})) = \frac{1}{16\sqrt{2}}$.*

*Proof.* Observe that from Lemma 6, in every block $1 \leq i \leq n$, $\mathrm{avg}(\mathbf{a}_{2i-1}, \mathbf{a}_{2i}) = \mathrm{Unif}(m)$. Furthermore, since all blocks have disjoint supports of size $m$, we conclude that $\mathrm{avg}(\mathbf{a}) = \mathrm{avg}(\mathbf{a}_1, \ldots, \mathbf{a}_{2n}) = \mathbf{u}_{mn} = \mathrm{Unif}(k)$.

Also, from Lemma 6, in every block $1 \leq i \leq n$, $\|\mathrm{avg}(\mathbf{a}_{2i-1}, \mathbf{a}_{2i}) - \mathrm{avg}(\mathbf{b}_{2i-1}, \mathbf{b}_{2i})\|_1 = \frac{1}{8\sqrt{2}}$. This immediately gives us $\|\mathrm{avg}(\mathbf{a}_1, \mathbf{a}_{2n}) - \mathrm{avg}(\mathbf{b}_1, \mathbf{b}_{2n})\|_1 = \frac{1}{8\sqrt{2}}$. Thus, $\mathrm{d_{TV}}(\mathrm{avg}(\mathbf{a}), \mathrm{avg}(\mathbf{b})) = \mathrm{d_{TV}}(\mathrm{Unif}(k), \mathrm{avg}(\mathbf{b})) = \frac{1}{16\sqrt{2}}$. $\square$

Next, we proceed to Step 2 in the outline above, which shows that the first two moments of the collision vectors are close.

**Claim 5.2** (Means match)**.** *Let $C_{\boldsymbol{a}}$ and $C_{\boldsymbol{b}}$ be the 3-dimensional vectors of 2-way, 3-way and 4-way collisions observed for the sequences $\boldsymbol{a}$ and $\boldsymbol{b}$ respectively over all the $n$ blocks. Then,*

$$\mathbb{E}\left[C_{\boldsymbol{a}}\right] = \mathbb{E}\left[C_{\boldsymbol{b}}\right].$$

*Proof.* Let $C_{\mathbf{a}_i}$ be the 3-dimensional vector of 2-way, 3-way and 4-way collisions observed for $\mathbf{a}$ in the $i^{\text{th}}$ block (i.e. 2 samples each from $\mathbf{a}_{2i-1}$ and $\mathbf{a}_{2i}$). Similarly, define $C_{\mathbf{b}_i}$. Then, the total number of 2-way, 3-way and 4-way collision across all $n$ blocks will be $\sum_{i \in [n]} C_{\mathbf{a}_i}$ and $\sum_{i \in [n]} C_{\mathbf{b}_i}$. Now, recall that exactly $\alpha n$ many blocks are of one kind, where each $C_{\mathbf{a}_i} = C_{\mathbf{p}}$ (respectively, $C_{\mathbf{b}_i} = C_{\mathbf{q}}$) independently, and the remaining $(1 - \alpha)n$ many blocks are of the second kind, where each $C_{\mathbf{a}_i} = C_{\mathbf{r}}$ (respectively, $C_{\mathbf{b}_i} = C_{\mathbf{s}}$) independently. Thus we have that

$$\mathbb{E}\left[\sum_{i \in [n]} C_{\mathbf{a}_i}\right] = \mathbb{E}\left[\sum_{i=1}^{\alpha n} C_{\mathbf{a}_i} + \sum_{i=\alpha n+1}^{n} C_{\mathbf{a}_i}\right] = n(\mathbb{E}[\alpha C_{\mathbf{p}} + (1 - \alpha)C_{\mathbf{r}}])$$
$$= n(\mathbb{E}[\alpha C_{\mathbf{q}} + (1 - \alpha)C_{\mathbf{s}}]) \qquad \text{(from Lemma 6)}$$
$$= \mathbb{E}\left[\sum_{i=1}^{\alpha n} C_{\mathbf{b}_i} + \sum_{i=\alpha n+1}^{n} C_{\mathbf{b}_i}\right] = \mathbb{E}\left[\sum_{i \in [n]} C_{\mathbf{b}_i}\right].$$

$\square$

**Claim 5.3** (Covariances match approximately)**.** *Let $C_{\boldsymbol{a}}$ and $C_{\boldsymbol{b}}$ be the 3-dimensional vectors of 2-way, 3-way and 4-way collisions observed for the sequences $\boldsymbol{a}$ and $\boldsymbol{b}$ respectively over all the $n$ blocks. Then the covariance matrices are as follows:*

$$\Sigma\left[C_{\boldsymbol{a}}\right] = n \begin{bmatrix} \gamma_{11}/m & \gamma_{12}/m^2 & \gamma_{13}/m^3 \\ \gamma_{12}/m^2 & \gamma_{22}/m^2 & \gamma_{23}/m^3 \\ \gamma_{13}/m^3 & \gamma_{23}/m^3 & \gamma_{33}/m^3 \end{bmatrix} + n \begin{bmatrix} \alpha_{11}/m^2 & \alpha_{12}/m^3 & \alpha_{13}/m^4 \\ \alpha_{12}/m^3 & \alpha_{22}/m^3 & \alpha_{23}/m^4 \\ \alpha_{13}/m^4 & \alpha_{23}/m^4 & \alpha_{33}/m^4 \end{bmatrix},$$

$$\Sigma\left[C_b\right] = n \begin{bmatrix} \gamma_{11}/m & \gamma_{12}/m^2 & \gamma_{13}/m^3 \\ \gamma_{12}/m^2 & \gamma_{22}/m^2 & \gamma_{23}/m^3 \\ \gamma_{13}/m^3 & \gamma_{23}/m^3 & \gamma_{33}/m^3 \end{bmatrix} + n \begin{bmatrix} \alpha'_{11}/m^2 & \alpha'_{12}/m^3 & \alpha'_{13}/m^4 \\ \alpha'_{12}/m^3 & \alpha'_{22}/m^3 & \alpha'_{23}/m^4 \\ \alpha'_{13}/m^4 & \alpha'_{23}/m^4 & \alpha'_{33}/m^4 \end{bmatrix},$$

*with $|\alpha_{ij}|, |\alpha'_{ij}|, |\gamma_{ij}|$ bounded by constants independent of $m$ (for $m$ large enough).*

*Proof.* As in Claim 5.2, we have that $C_{\mathbf{a}} = \sum_{i\in[n]} C_{\mathbf{a}_i}$ and $C_{\mathbf{b}} = \sum_{i\in[n]} C_{\mathbf{b}_i}$, where $C_{\mathbf{a}_i}$ is either $C_{\mathbf{p}}$ or $C_{\mathbf{r}}$, and $C_{\mathbf{b}_i}$ is either $C_{\mathbf{q}}$ or $C_{\mathbf{s}}$. Further, since the samples in each block are independent, the covariances add up. Thus, we only need to compute the covariance in one block, which we recall supports two distributions and contributes 4 samples. Suppressing subscripts for now, we have that

$$\Sigma[C] = \begin{bmatrix} \mathbb{E}[c_2^2] - \mathbb{E}[c_2]^2 & \mathbb{E}[c_2 c_3] - \mathbb{E}[c_2]\mathbb{E}[c_3] & \mathbb{E}[c_2 c_4] - \mathbb{E}[c_2]\mathbb{E}[c_4] \\ \mathbb{E}[c_2 c_3] - \mathbb{E}[c_2]\mathbb{E}[c_3] & \mathbb{E}[c_3^2] - \mathbb{E}[c_3]^2 & \mathbb{E}[c_3 c_4] - \mathbb{E}[c_3]\mathbb{E}[c_4] \\ \mathbb{E}[c_2 c_4] - \mathbb{E}[c_2]\mathbb{E}[c_4] & \mathbb{E}[c_3 c_4] - \mathbb{E}[c_3]\mathbb{E}[c_4] & \mathbb{E}[c_4^2] - \mathbb{E}[c_4]^2 \end{bmatrix}.$$

Let $Y_{ab}$ be the indicator that the elements at indices $a$ and $b$ collide. Note that $1 \le a, b \le 4$, since we have 4 samples in each block. Then,

$$\mathbb{E}[c_2^2] = \mathbb{E}\left[\left(\sum_{a<b} Y_{ab}\right)\left(\sum_{i<j} Y_{ij}\right)\right]$$

$$= \mathbb{E}\left[\sum_{a<b} Y_{ab}^2 + 6\sum_{a<b<c} Y_{abc} + 6\sum_{a<b<c<d} Y_{ab}Y_{cd}\right]$$

$$= \mathbb{E}\left[\sum_{a<b} Y_{ab}\right] + 6 \cdot \mathbb{E}\left[\sum_{a<b<c} Y_{abc}\right] + 6 \cdot \mathbb{E}[Y_{12}] \cdot \mathbb{E}[Y_{34}]$$

$$= \mathbb{E}[c_2] + 6 \cdot \mathbb{E}[c_3] + 6 \cdot \Pr[\text{first two samples collide}] \cdot \Pr[\text{second two samples collide}].$$

In a similar manner, we can obtain the following expressions for the remaining terms:

$$\mathbb{E}[c_2^3] = \mathbb{E}[c_3] + 12 \cdot \mathbb{E}[c_4]$$
$$\mathbb{E}[c_4^2] = \mathbb{E}[c_4]$$
$$\mathbb{E}[c_2 c_3] = 3 \cdot \mathbb{E}[c_3] + 12 \cdot \mathbb{E}[c_4]$$
$$\mathbb{E}[c_2 c_4] = 6 \cdot \mathbb{E}[c_4]$$
$$\mathbb{E}[c_3 c_4] = 4 \cdot \mathbb{E}[c_4].$$

Thus, we have expressed all the entries in the covariance matrices in terms of just the expected number of 2-way, 3-way and 4-way collisions. We have already computed explicit expressions for these for each of $\mathbf{p}$, $\mathbf{q}$, $\mathbf{r}$ and $\mathbf{s}$ in the proof of Lemma 6. Plugging these in, we obtain

$$\Sigma[C_{\mathbf{p}}] = \begin{bmatrix} \frac{6}{m} & \frac{12}{m^2} & \frac{6}{m^3} \\ \frac{12}{m^2} & \frac{4}{m^2} & \frac{4}{m^3} \\ \frac{6}{m^3} & \frac{4}{m^3} & \frac{1}{m^3} \end{bmatrix} + \begin{bmatrix} -\frac{6}{m^2} & -\frac{12}{m^3} & -\frac{6}{m^4} \\ -\frac{12}{m^3} & \frac{12}{m^3} - \frac{16}{m^4} & -\frac{4}{m^5} \\ -\frac{6}{m^4} & -\frac{4}{m^5} & -\frac{1}{m^6} \end{bmatrix}$$

$$\Sigma[C_{\mathbf{r}}] = \begin{bmatrix} \frac{4}{m} & 0 & 0 \\ 0 & 0 & 0 \\ 0 & 0 & 0 \end{bmatrix} + \begin{bmatrix} \frac{8}{m^2} & 0 & 0 \\ 0 & 0 & 0 \\ 0 & 0 & 0 \end{bmatrix}$$

$$\Sigma[C_{\mathbf{q}}] = \Sigma[C_{\mathbf{s}}] = \begin{bmatrix} \frac{11}{2m} & \frac{9}{m^2} & \frac{9}{2m^3} \\ \frac{9}{m^2} & \frac{3}{m^2} & \frac{3}{m^3} \\ \frac{9}{2m^3} & \frac{3}{m^3} & \frac{3}{4m^3} \end{bmatrix} + \begin{bmatrix} \frac{5}{4m^2} & -\frac{15}{2m^3} & -\frac{33}{8m^4} \\ -\frac{15}{2m^3} & \frac{9}{m^3} - \frac{9}{m^4} & -\frac{33}{4m^5} \\ -\frac{33}{8m^4} & -\frac{9}{4m^5} & -\frac{9}{16m^6} \end{bmatrix}.$$

Finally, recall that

$$\Sigma[C_{\mathbf{a}}] = \Sigma\left[\sum_{i=1}^{\alpha n} C_{\mathbf{a}_i}\right] + \Sigma\left[\sum_{i=\alpha n+1}^{n} C_{\mathbf{a}_i}\right] = \alpha n \Sigma[C_{\mathbf{p}}] + (1-\alpha)n\Sigma[C_{\mathbf{r}}]$$

$$\Sigma[C_{\mathbf{b}}] = \Sigma\left[\sum_{i=1}^{\alpha n} C_{\mathbf{b}_i}\right] + \Sigma\left[\sum_{i=\alpha n+1}^{n} C_{\mathbf{b}_i}\right] = \alpha n \Sigma[C_{\mathbf{q}}] + (1-\alpha)n\Sigma[C_{\mathbf{s}}]$$

for $\alpha = \frac{3}{4}$, and hence, substituting the values above, we get

$$\Sigma[C_{\mathbf{a}}] = n\begin{bmatrix} \frac{11}{2m} & \frac{9}{m^2} & \frac{9}{2m^3} \\ \frac{9}{m^2} & \frac{3}{m^2} & \frac{3}{m^3} \\ \frac{9}{2m^3} & \frac{3}{m^3} & \frac{3}{4m^3} \end{bmatrix} + n\begin{bmatrix} -\frac{5}{2m^2} & -\frac{9}{m^3} & -\frac{9}{2m^4} \\ -\frac{9}{m^3} & \frac{9}{m^3} - \frac{12}{m^4} & -\frac{3}{m^5} \\ -\frac{9}{2m^4} & -\frac{3}{m^5} & -\frac{3}{4m^6} \end{bmatrix}$$

$$\Sigma[C_{\mathbf{b}}] = n\begin{bmatrix} \frac{11}{2m} & \frac{9}{m^2} & \frac{9}{2m^3} \\ \frac{9}{m^2} & \frac{3}{m^2} & \frac{3}{m^3} \\ \frac{9}{2m^3} & \frac{3}{m^3} & \frac{3}{4m^3} \end{bmatrix} + n\begin{bmatrix} \frac{5}{4m^2} & -\frac{15}{2m^3} & -\frac{33}{8m^4} \\ -\frac{15}{2m^3} & \frac{9}{m^3} - \frac{9}{m^4} & -\frac{9}{4m^5} \\ -\frac{33}{8m^4} & -\frac{9}{4m^5} & -\frac{9}{16m^6} \end{bmatrix}.$$

For the terms $9/m^3 - 12/m^4$ and $9/m^3 - 9/m^4$, we can write them as $\frac{(9-12/m)}{m^3}$ and $\frac{9-9/m}{m^3}$ respectively. Thus for $m$ large enough, we can write all the terms in the matrices in the form specified in the claim, completing the proof. $\square$

We now prove a CLT (Step 3) which helps us approximate the distributions of the fingerprints by multivariate Gaussians having the same mean and covariance.

**Lemma 9.** *Consider a c-dimensional distribution (for $c \leq 10$) supported on $[0, 0, \ldots, 0]$, the c-dimensional basis vectors, and $[2, 0, \ldots, 0]$, such that the masses on the non-origin support points are fixed positive constants, and the mass on origin $[0, 0, \ldots, 0]$ is at least $1 - 10^{-16}$. Then for $\ell$ large enough, the distribution of the sum of $\ell$ i.i.d samples from this distribution has total variation distance at most $0.002$ from the Gaussian with corresponding mean and covariance, whose mass has been discretized to the nearest point in the integer lattice.*

*Proof.* Let $u_0 = \Pr([0, 0, \ldots, 0])$, and $u_1 = \Pr([1, 0, \ldots, 0]), u_2 = \Pr([0, 1, \ldots, 0]), \ldots, u_c = \Pr([0, 0, \ldots, 1])$ and $v = \Pr([2, 0, \ldots, 0])$. We draw a sample from this distribution as follows: first flip a fair coin, and if the coin lands heads, sample from the distribution over $[0, 0, \ldots, 0]$ and the basis vectors, given by respective probabilities $1 - 2\sum_{i=1}^{c} u_i, 2u_1, 2u_2, \ldots, 2u_c$. If the coin lands tails, we sample from the distribution over $[0, 0, \ldots, 0], [2, 0, \ldots, 0]$ with respective probability $1 - 2v, 2v$. Note that this sampling procedure yields a sample from the original distribution. Here, we are using that $v, \sum_{i=1}^{c} u_i$ are both at most $10^{-16}$ for the probabilities to be well-defined.

Given $\ell$ independent samples sampled from the distribution in this way, let us consider the distribution of the sum of the samples, conditioned on exactly $t$ of the $\ell$ fair coins having landed heads. Note that for large $\ell$, with probability at least $0.999$, $t \in [\ell/2 - 4\sqrt{\ell}, \ell/2 + 4\sqrt{\ell}]$.

Thus for all $t$ in this range, if we can show that the sum of $\ell$ i.i.d. samples has TV distance at most $\epsilon$ from the desired discretized Gaussian, this would imply a total variation distance of at most $0.001 + \epsilon$ without conditioning on $t$. Thus we would focus on the distribution conditioned on $t$ taking a value in $[\ell/2 - 4\sqrt{\ell}, \ell/2 + 4\sqrt{\ell}]$.

Conditioned on a value of $t$, the distribution corresponds to the convolution of $t$ independent draws from the multinomial distribution given by the distribution corresponding to the coin landing heads, and the distribution corresponding to the $\ell - t$ samples from the distribution supported on $[0, 0, \ldots, 0]$ and $[2, 0, \ldots, 0]$, which is a binomial random variable in the first coordinate, scaled by a factor of 2.

Let us first reason about the $t$ independent draws from the multinomial distribution. Leveraging a multivariate CLT, the distribution of the sum of the $t$ draws from the multinomial will have total variation distance at most $O\left(\frac{(\log t)^{2/3}}{\sigma^{1/3}}\right)$ from the multivariate Gaussian of corresponding mean and covariance, discretized to the nearest point in the integer lattice ( (Valiant & Valiant, 2011, Theorem 4); see Section F.2 for a restatement). Here, $\sigma^2$ denotes the smallest eigenvalue of the covariance of the sum vector of the $t$ draws which will be $\Omega(t)$.

To see this, observe that this sum vector has mean $t \cdot [2u_1, \dots, 2u_c]$, and covariance

$$t \cdot \begin{bmatrix} 2u_1(1-2u_1) & -4u_1u_2 & -4u_1u_3 & \dots & -4u_1u_c \\ -4u_1u_2 & 2u_2(1-2u_2) & -4u_2u_3 & \dots & -4u_2u_c \\ \vdots & & \dots & & \\ -4u_1u_c & -4u_2u_c & -4u_3u_c & \dots & 2u_c(1-2u_c) \end{bmatrix}$$

The determinant of the matrix above, without the $t$ scaling, is exactly

$$2u_1 \cdot 2u_2 \cdots \cdots 2u_c \cdot \left( 1 - 2\sum_{i=1}^{c} u_i \right).$$

For constant $c$, and by the assumption that $u_i > 0$ for all $i$ and $\sum_{i=1}^{c} u_i \le 10^{-16}$, this determinant is a positive constant independent of $\ell$. This implies that the smallest eigenvalue of this (positive semidefinite) matrix is at least (and at most) a positive constant. Together with the scaling of $t$, we get that $\sigma^2 = \Omega(t)$, meaning that $\sigma = \Omega(\sqrt{t})$. Thus, the total variation distance goes to 0 as $t$ gets large. Consequently, we take $\ell$ to be large enough so that for each $t$ under consideration, this distance is at most $10^{-5}$.

Similarly instantiating the CLT on the sum vector of the $\ell - t$ samples corresponding to the coin landing tails, which, recall is simply a scaled binomial random variable in the first coordinate (and zero elsewhere) yields that for large enough $\ell$, this vector has TV distance at most $10^{-5}$ close to a Gaussian having the corresponding mean and covariance (discretized to the closest integer) in the first coordinate, and zero elsewhere. Also, note that the variance of first coordinate is $O(\ell)$ (for $t$ in the assumed range) which is large for $\ell$ large enough.

Thus, the sum of $t$ vectors corresponding to coin landing heads and $\ell - t$ vectors corresponding to coin landing tails has distance at most $2 \cdot 10^{-5}$ from the convolution of the corresponding discretized Gaussians. Now we can apply Lemma 21 which gives us that the convolution of the discretized Gaussians here has distance at most $10^{-5}$ to the discretization of the convolution of two Gaussians (here we use the fact that the variance of the first coordinate of the vector corresponding to coin landing tails is large enough and all its other coordinates are zero). By a triangle inequality, we get that the sum of $t$ vectors corresponding to coin landing heads and $\ell - t$ vectors corresponding to coin landing tails has distance at most $3 \cdot 10^{-5}$ from the discretized Gaussian of corresponding mean and covariance.

So far, we have shown that for any $t \in [\ell/2 - 4\sqrt{\ell}, \ell/2 + 4\sqrt{\ell}]$, sum of $t$ vectors corresponding to coin landing heads and $\ell - t$ vectors corresponding to coin landing tails has distance at most $3 \cdot 10^{-5}$ from the discretized Gaussian of corresponding mean and covariance. In Lemma 23, we show that all these Gaussians have TV distance at most $10^{-5}$ from the desired Gaussian (the Gaussian having mean and covariance same as mean and covariance of sum of $\ell$ i.i.d. samples from our original distribution). Lemma 23 uses the fact that $u_0$ is close to 1 and $c$ is a small constant. This Lemma essentially involves showing that the difference between $i^{\text{th}}$ coordinate of the means of the Gaussians is roughly $u_i\sqrt{\ell}$ whereas the standard deviation is roughly $\sqrt{u_i}\sqrt{\ell}$. Thus for $u_0$ large enough, $u_i$ will be small, and the standard deviation will be much larger than the difference between the means.

Combining the above arguments, we get that sum of $\ell$ i.i.d. samples from our distribution has TV distance at most $0.001 + 3 \cdot 10^{-5} + 10^{-5} \le 0.002$ from the discretized Gaussian with corresponding mean and covariance.

$\square$

**Lemma 10.** *Consider two distributions supported on $[0,0,0]$, the basis vectors, and $[2,0,0]$ such that the proability of $[0,0,0]$ is at least $1 - 10^{-16}$. Let for the first distribution, mass on all these vectors is guaranteed to be non-zero, and for the second distribution, mass on $[0,0,0]$, $[1,0,0]$ and $[2,0,0]$ is guaranteed to be non-zero. Then for constant $\alpha \in (0,1)$, and $n$ large enough, the distribution of the sum of $\alpha n$ i.i.d samples from the first distribution and $(1-\alpha)n$ i.i.d samples from the second distribution has total variation distance at most $0.01$ from a Gaussian with corresponding mean and covariance, where the mass has been discretized to the nearest point in the integer lattice.*

*Proof.* Using Lemma 9, we know that the sum of $\alpha n$ i.i.d. samples from the first distribution and sum of $(1-\alpha)n$ i.i.d. samples from the second distribution has TV distance at most $0.002$ from the corresponding

discretized Gaussians. Here, for the first distribution we apply Lemma 9 with $c = 3$ and for second distribution, we use $c \leq 3$ depending on whether it has non-zero mass on $[0, 1, 0]$ and $[0, 0, 1]$. All that remains, is to prove that the convolution of two discretized Gaussians is close, in total variation distance, to the discretization of the convolution of the two Gaussians. This is true for any Gaussians in constant dimension such that the minimum eigenvalue of covariance of at least one of the Gaussians is super-constant (Lemma 18). In our case, the Gaussian corresponding to the sum of $\alpha n$ samples from the first distribution has minimum eigenvalue $\Theta(n)$. This is because the per-sample covariance for the first distribution $\Sigma_1$ has $\det(\Sigma_1) > 0$ and $\|\Sigma_1\| = O(1)$ (both independent of $n$), so with $\Sigma_{\alpha n} = \alpha n \Sigma_1$ we get $\lambda_{\min}(\Sigma_{\alpha n}) \geq \alpha n \det(\Sigma_1)/\|\Sigma_1\|^2 = \Theta(n)$ (see Lemma 22 for detailed proof). Taking $n$ to be large enough, we get that convolution of the two discretized Gaussians has distance at most $10^{-5}$, to the discretization of the convolution of the two Gaussians. Applying a triangle inequality, we get that the sum of $n$ i.i.d. samples produced as above has TV distance at most $0.004 + 10^{-5} \leq 0.01$ from the corresponding discretized Gaussian.

$\square$

Let $FP(\mathbf{a})$ and $FP(\mathbf{b})$ be 3 dimensional random variables containing the number of elements observed exactly 2, 3 and 4 times, for samples drawn from $\mathbf{a}$ and $\mathbf{b}$ respectively. Let $G_{\mathbf{a}}$ be a random variable corresponding to picking a sample from gaussian with same mean and covariance as $FP(\mathbf{a})$ and rounding each coordinate to the nearest integer. Similarly, $G_{\mathbf{b}}$ corresponds to $FP(\mathbf{b})$.

**Lemma 11** (Fingerprints close to discretized Gaussians). $d_{\mathrm{TV}}(FP(\boldsymbol{a}), G_{\boldsymbol{a}}) \leq 0.01$ and $d_{\mathrm{TV}}(FP(\boldsymbol{b}), G_{\boldsymbol{b}}) \leq 0.01$.

*Proof.* Recall that $FP(\mathbf{a})$ and $FP(\mathbf{b})$ are $3-$dimensional random variables containing number of elements appearing exactly 2, 3 and 4 times in samples drawn according to $\mathbf{a}$ and $\mathbf{b}$ respectively. Due to the structure of $\mathbf{a}$ and $\mathbf{b}$ (see Definition 8), we can decompose $FP(\mathbf{a})$ and $FP(\mathbf{b})$ as

$$FP(\mathbf{a}) = \sum_{i=1}^{\alpha n} FP(\mathbf{p}^i) + \sum_{i=\alpha n+1}^{n} FP(\mathbf{r}^i),$$

$$FP(\mathbf{b}) = \sum_{i=1}^{\alpha n} FP(\mathbf{q}^i) + \sum_{i=\alpha n+1}^{n} FP(\mathbf{s}^i).$$

Here, $FP(\mathbf{p}^i)$ denotes the fingerprint vector when we obtain 2 samples from $\mathbf{p}_1^i$ and 2 samples from $\mathbf{p}_2^i$. Note that as we vary $i$, $FP(\mathbf{p}^i)$ corresponds to independent samples from the same distribution. $FP(\mathbf{q}^i)$, $FP(\mathbf{r}^i)$ and $FP(\mathbf{s}^i)$ are defined analogously. Note that each of $FP(\mathbf{p}^i)$, $FP(\mathbf{q}^i)$, $FP(\mathbf{r}^i)$ and $FP(\mathbf{s}^i)$ are supported on $[0, 0, 0]$, the basis vectors and $[2, 0, 0]$. Here, $FP(\mathbf{p}^i)$, $FP(\mathbf{q}^i)$, $FP(\mathbf{s}^i)$ has non-zero mass on all of these vectors, and $FP(\mathbf{r}^i)$ has non-zero mass on $[0, 0, 0], [1, 0, 0], [2, 0, 0]$.

Also, note that $FP(\mathbf{p}^i) = [0, 0, 0]$ corresponds to the event that when we draw 2 samples each from $p_1^i$ and $p_2^i$, the 4 samples so obtained are distinct. The probability of this event is equal to $(1 - 1/m)(1 - 2/m)(1 - 3/m)$ which is at least $1 - 10^{-16}$ if we choose $m$ to be large enough. Similarly, one can verify that for $m$ large enough constant, for each of $FP(\mathbf{q}^i)$, $FP(\mathbf{r}^i)$ and $FP(\mathbf{s}^i)$, the probability of these random variables being $[0, 0, 0]$ is at least $1 - 10^{-16}$.

By applying Lemma 10, we get $d_{\mathrm{TV}}(FP(\mathbf{a}), G_{\mathbf{a}}) \leq 0.01$ and $d_{\mathrm{TV}}(FP(\mathbf{b}), G_{\mathbf{b}}) \leq 0.01$, which completes the proof. $\square$

Finally, we conclude with the calculations required in Step 4 of the outline.

**Lemma 12** (Discretized Gaussians are close). $d_{\mathrm{TV}}(G_{\boldsymbol{a}}, G_{\boldsymbol{b}}) \leq 0.02$.

*Proof.* Let $G'_{\mathbf{a}}$ be a random variable corresponding to picking a sample from gaussian with same mean and co-variance as $FP(\mathbf{a})$ (without any rounding). Similarly, $G'_{\mathbf{b}}$ corresponds to $FP(\mathbf{b})$. By data processing inequality, rounding each coordinate can only decrease the total variation distance, $d_{\mathrm{TV}}(G_{\mathbf{a}}, G_{\mathbf{b}}) \leq d_{\mathrm{TV}}(G'_{\mathbf{a}}, G'_{\mathbf{b}})$.

By Lemma 14, given 4 samples from a finite support, there exists an invertible linear map, say $A$, such that we can go from a vector of fingerprints to vector of collisions by applying $A$ to the vector of fingerprints.

Since we can decompose $\mathbf{a}$ and $\mathbf{b}$ into disjoint supports from each of which we are getting 4 samples, applying $A$ also lets us go from vector of fingerprints to vector of collisions for samples obtained from $\mathbf{a}$ and $\mathbf{b}$.

We apply $A$ to random variables $G'_{\mathbf{a}}$ and $G'_{\mathbf{b}}$. Since $A$ is invertible, $d_{\mathrm{TV}}(AG'_{\mathbf{a}}, AG'_{\mathbf{b}}) = d_{\mathrm{TV}}(G'_{\mathbf{a}}, G'_{\mathbf{b}})$. Here $AG'_{\mathbf{a}}$ is a 3 dimensional Gaussian random variable with mean and covariance same as the mean and covariance of the vector containing number of 2-way, 3-way and 4-way collisions in the samples produced by $\mathbf{a}$. $AG'_{\mathbf{b}}$ is related to $\mathbf{b}$ in the same fashion.

From Claim 5.2, we know that expectations of $AG'_{\mathbf{a}}$ and $AG'_{\mathbf{b}}$ match. Let us denote the expectation by $\mu$. From Claim 5.3, we know that covariances of $AG'_{\mathbf{a}}$ and $AG'_{\mathbf{b}}$ are as follows:

$$\Sigma(AG'_{\mathbf{a}}) = \Sigma_1 + n \begin{bmatrix} \alpha_{11}/m^2 & \alpha_{12}/m^3 & \alpha_{13}/m^4 \\ \alpha_{12}/m^3 & \alpha_{22}/m^3 & \alpha_{23}/m^4 \\ \alpha_{13}/m^4 & \alpha_{23}/m^4 & \alpha_{33}/m^4 \end{bmatrix}, \quad \Sigma(AG'_{\mathbf{b}}) = \Sigma_1 + n \begin{bmatrix} \alpha'_{11}/m^2 & \alpha'_{12}/m^3 & \alpha'_{13}/m^4 \\ \alpha'_{12}/m^3 & \alpha'_{22}/m^3 & \alpha'_{23}/m^4 \\ \alpha'_{13}/m^4 & \alpha'_{23}/m^4 & \alpha'_{33}/m^4 \end{bmatrix},$$

where

$$\Sigma_1 = n \begin{bmatrix} \gamma_{11}/m & \gamma_{12}/m^2 & \gamma_{13}/m^3 \\ \gamma_{12}/m^2 & \gamma_{22}/m^2 & \gamma_{23}/m^3 \\ \gamma_{13}/m^3 & \gamma_{23}/m^3 & \gamma_{33}/m^3 \end{bmatrix}.$$

Let $G_1$ be a random variable distributed as a Gaussian with mean $\mu$ and covariance $\Sigma_1$. From Lemma 17, which is a technical lemma that bounds the TV distance between Gaussians having the same mean and similar covariances, and whose proof is given in Appendix F, we have that $d_{\mathrm{TV}}(G_1, AG'_{\mathbf{a}}) \leq 0.01$ and $d_{\mathrm{TV}}(G_1, AG'_{\mathbf{b}}) \leq 0.01$ (compared to the lemma statement there, covariances here have an additional factor of $n$ but that does not change the TV distance). Using triangle inequality, we get $d_{\mathrm{TV}}(AG'_{\mathbf{a}}, AG'_{\mathbf{b}}) \leq 0.02$. Since $d_{\mathrm{TV}}(G_{\mathbf{a}}, G_{\mathbf{b}}) \leq d_{\mathrm{TV}}(G'_{\mathbf{a}}, G'_{\mathbf{b}}) = d_{\mathrm{TV}}(AG'_{\mathbf{a}}, AG'_{\mathbf{b}})$, this completes the proof. $\qquad\square$

**Lemma 13** (Fingerprints are close). $d_{\mathrm{TV}}(FP(\boldsymbol{a}), FP(\boldsymbol{b})) \leq 0.04$.

*Proof.* $d_{\mathrm{TV}}(FP(\mathbf{a}), FP(\mathbf{b})) \leq d_{\mathrm{TV}}(FP(\mathbf{a}), G_{\mathbf{a}}) + d_{\mathrm{TV}}(FP(\mathbf{b}), G_{\mathbf{b}}) + d_{\mathrm{TV}}(G_{\mathbf{a}}, G_{\mathbf{b}})$ by triangle inequality which is at most 0.04 due to Lemma 11 and Lemma 12. $\qquad\square$

With this, we have shown that the TV distance between the distribution of fingerprints of the two sequences is small, even when the total variation distance between the corresponding average distributions is at least some constant. We can then invoke the Neyman-Pearson lemma (Neyman & Pearson, 1933), (Canonne, 2022, Lemma 1.4) to say that no testing algorithm can distinguish between these two cases with high probability. This concludes the proof of Theorem 1.6.

# 6 Conclusion

We show that sublinear sample complexity property testing extends to the non-i.i.d. setting where the samples are drawn from different distributions, and we are interested about the average distribution. In particular, natural collision-based testers with just a constant number of samples from each distribution suffice to solve the identity and closeness testing problems. While our analysis is optimal with respect to the support size, it is still sub-optimal with respect to the distance parameter $\varepsilon$—resolving this is an interesting future direction. Another direction could be to chart the landscape of the problem when $T$ (number of distributions) is fixed, and $c$ (number of samples from each distribution) varies.

# Acknowledgments

S.G. did part of this work at Stanford where he was supported by a Stanford Interdisciplinary Graduate Fellowship, and part of it at Harvard, where he was supported by a postdoctoral fellowship from Harvard's Digital Data Design Institute. C.P. was supported by Moses Charikar's and Gregory Valiant's Simons Investigator Awards. K.S. was supported, in part, by funding from the Eric and Wendy Schmidt Center at the Broad Institute of MIT and Harvard. G.V. was supported, in part, by a Simons Investigator Award.

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

## A  Learning the average distribution

In this section, we show that the sample complexity of *learning* the average distribution in the setting of non-identical samples is asymptotically equal to that of the i.i.d. setting.

**Claim 1.1.** *Given access to $T$ distributions, $\boldsymbol{p}_1, \ldots, \boldsymbol{p}_T$, each supported on a common domain of size $\leq k$, for any $\varepsilon > 0$, given $c = 1$ samples drawn from each $\boldsymbol{p}_i$, one can output a distribution $\tilde{\boldsymbol{p}}_{\mathrm{avg}}$ such that with probability at least $2/3$, $\mathrm{d}_{\mathrm{TV}}(\boldsymbol{p}_{\mathrm{avg}}, \tilde{\boldsymbol{p}}_{\mathrm{avg}}) \leq \varepsilon$, provided $T = O(k/\varepsilon^2)$. Furthermore, $\Omega(k/\varepsilon^2)$ samples are necessary, even if $\boldsymbol{p}_1 = \ldots = \boldsymbol{p}_T = \boldsymbol{p}_{avg}$ for such a guarantee.*

*Proof.* The lower bound for the setting of non-identical samples follows directly from the lower bound of the standard i.i.d setting, that is if we let $\mathbf{p}_t = \mathbf{p}, \forall t \in [T]$. For the upper bound, we follow the proof in (Canonne, 2020, Theorem 1). Consider the empirical estimator given by,

$$\tilde{\mathbf{p}}_{\mathrm{avg}}(i) = \frac{1}{T} \sum_{t \in [T]} \mathbb{1}[X_t = i].$$

We have that

$$\mathbb{E}\left\|\tilde{\mathbf{p}}_{\mathrm{avg}} - \mathbf{p}_{\mathrm{avg}}\right\|_2^2 = \frac{1}{T^2} \sum_i \sum_{t \in [T]} \mathbf{p}_t(i)(1 - \mathbf{p}_t(i))$$

$$\leq \frac{1}{T^2} \sum_i \sum_{t \in [T]} \mathbf{p}_t(i)$$

$$= \frac{1}{T} \sum_i \mathbf{p}_{\mathrm{avg}}(i) = \frac{1}{T}.$$

Thus, by Markov's inequality, the squared $\ell_2$ distance of this estimator from $\mathbf{p}_{\mathrm{avg}}$ is smaller than $\frac{3}{T}$ with probability at least $\frac{2}{3}$. By Cauchy-Schwartz, the $\ell_1$ distance of this estimator from $\mathbf{p}_{\mathrm{avg}}$ is at most $\sqrt{\frac{3k}{T}}$ with probability at least $\frac{2}{3}$ and thus, this distance is smaller than $\varepsilon$ if $T = O(k/\varepsilon^2)$.

$\square$

## B   Impossibility Result with $c = 1$

**Claim 1.2.** *Given access to $T \leq k/2$ distributions, $\boldsymbol{p}_1, \ldots, \boldsymbol{p}_T$, and $c = 1$ sample drawn from each distribution, no tester can distinguish the case that the average distribution, $\boldsymbol{p}_{\mathrm{avg}}$, is the uniform distribution of support $k$, versus has total variation distance at least $1/4$ from $\mathrm{Unif}(k)$, with probability of success greater than $2/3$.*

We prove the following equivalent claim.

**Claim B.1.** *For all $c_1 > 0$, there is a $k \geq c_1$ such that for any $T \leq k/2$, given $c = 1$ sample drawn from each of $\boldsymbol{p}_1, \ldots, \boldsymbol{p}_T$, there exists no testing algorithm that succeeds with probability greater than $1/2$ and tests whether*

$$\boldsymbol{p}_{\mathrm{avg}} = \mathrm{Unif}(k) \quad versus \quad \mathrm{d}_{\mathrm{TV}}(\boldsymbol{p}_{\mathrm{avg}}, \mathrm{Unif}(k)) \geq 1/4 .$$

*Proof.* All the logarithms in the proof below are base 2. Let $k$ be the smallest power of 2 greater than or equal to $c_1$,

$$k = 2^{\lceil \log c_1 \rceil}.$$

For $T \leq k/2$, let $r = k/T$, so that $r \geq 2$. We set $r'$ to be the smallest power of 2 greater than or equal to $r$,

$$r' = 2^{\lceil \log r \rceil}.$$

Let $T' = k/r'$. By construction $T' \leq T \leq k/2$, $T'$ is integral, and $T'/T \geq 1/2$.

We now consider two distributions $D_1$ and $D_2$ over sequences of distributions $(\mathbf{p}_1 \ldots \mathbf{p}_T)$ such that for all sequences $(\mathbf{p}_1 \ldots \mathbf{p}_T)$ in the support of $D_1$, $\mathrm{avg}(\mathbf{p}_1 \ldots \mathbf{p}_T) = \mathrm{Unif}(k)$ and for all sequences in the support of $D_2$, $\mathrm{d}_{\mathrm{TV}}(\mathrm{avg}(\mathbf{p}_1 \ldots \mathbf{p}_T), \mathrm{Unif}(k)) \geq 1/4$. Further, we will show that the samples drawn from distribution sequences drawn from $D_1$ are indistinguishable from samples drawn from distribution sequences drawn from $D_2$ (that is, corresponding distributions have total variation distance 0). This implies that there exists no testing algorithm that succeeds with probability greater than $1/2$ and tests whether average distribution is the uniform distribution versus average distribution has total variation distance at least $1/4$ from the uniform distribution.

Here is the procedure to draw a sequence of distribution $(\mathbf{p}_1 \ldots \mathbf{p}_T)$ from $D_1$:

1. Draw a permutation $\pi : [k] \to [k]$ uniformly at random.

2. Use the permutation $\pi$ to partition the domain into $T'$ sets of size $r'$ each where for $i \leq T'$, each $\mathbf{p}_i$ corresponds to a uniform distribution over a distinct set. That is, for $1 \leq i \leq T'$, $\mathbf{p}_i$ corresponds to a uniform distribution over $\{\pi((i-1) * r' + 1), \pi((i-1) * r' + 2), \cdots, \pi(i * r')\}$.

3. For $T' + 1 \leq i \leq T$, $\mathbf{p}_i$ corresponds to a uniform distribution over the whole support $[k]$, that is $\mathbf{p}_i = \mathrm{Unif}(k)$.

Here is the procedure to draw a sequence of distributions $(\mathbf{p}_1 \ldots \mathbf{p}_T)$ from $D_2$:

1. Draw a permutation $\pi : [k] \to [k]$ uniformly at random.

2. Use the permutation $\pi$ to partition the domain into $T'$ sets of size $r'$ each where for $i \le T'$, each $\mathbf{p}_i$ corresponds to a point-mass distribution supported over an element of a distinct set: For $1 \le i \le T'$, $\mathbf{p}_i$ corresponds to a point-mass distribution with probability mass 1 on element $\pi\left((i-1) * r' + 1\right)$.

3. For $T' + 1 \le i \le T$, $\mathbf{p}_i$ corresponds to a uniform distribution over the whole support $[k]$, that is $\mathbf{p}_i = \mathrm{Unif}(k)$.

Note that all the distributions sequences in the support of $D_1$ satisfy $\mathrm{avg}(\mathbf{p}_1 \ldots \mathbf{p}_T) = u_k$. Next, we show that all the distribution sequences in the support of $D_2$ satisfy $\mathrm{d_{TV}}(\mathrm{avg}(\mathbf{p}_1 \ldots \mathbf{p}_T), \mathrm{Unif}(k)) \ge 1/4$. For such distribution sequences, $\mathrm{avg}(\mathbf{p}_1 \ldots \mathbf{p}'_T)$ has mass $\frac{1}{k} - \frac{T'}{Tk} + \frac{1}{T}$ on $T'$ domain elements and mass $\frac{1}{k} - \frac{T'}{Tk}$ on $k - T'$ domain elements. From this, we get

$$\mathrm{d_{TV}}(\mathrm{avg}(\mathbf{p}_1 \ldots \mathbf{p}_T), \mathrm{Unif}(k)) = \frac{(k - T')T'}{Tk}$$
$$\ge \frac{1}{4}$$

where for the last inequality, we used $T' \le T \le k/2$ and $T'/T \ge 1/2$.

Now, we will show that the samples drawn from distribution sequences drawn from $D_1$ are indistinguishable from samples drawn from distribution sequences drawn from $D_2$ which will complete the proof. Let $D_1^X$ be the distribution corresponding to $c = 1$ sample drawn from each of $\mathbf{p}_1, \ldots, \mathbf{p}_T$, where $(\mathbf{p}_1 \ldots \mathbf{p}_T)$ is drawn from $D_1$ (with $c = 1$). We define $D_2^X$ analogously. We want to show that $\mathrm{d_{TV}}(D_1^X, D_2^X) = 0$. Note that in both $D_1^X$ and $D_2^X$, each of last $(T - T')$ samples $\{X_t\}_{t \in [T - T']}$ are drawn from $\mathrm{Unif}(k)$ independent of the first $T'$ samples $\{X_t\}_{t \in [T']}$. And the first $T'$ samples $\{X_t\}_{t \in [T']}$ correspond to choosing $T'$ distinct domain elements uniformly at random. Thus $D_1^X$ and $D_2^X$ are the same distributions.

$\square$

## C   Poissonized setting

In this section, we state and prove some results for the non-i.i.d. setting, where we are also allowed to use Poissonization. We will use the following standard facts about Poisson distributions.

Let $\mathrm{Poi}(\lambda)$ denote a Poisson random variable with mean parameter $\lambda \ge 0$.

**Fact C.1.** *For any $\lambda_1, \lambda_2 \ge 0$, if $X_1$ and $X_2$ are independent random variables distributed as $X_1 \sim \mathrm{Poi}(\lambda_1)$ and $X_2 \sim \mathrm{Poi}(\lambda_2)$, then $X_1 + X_2 \sim \mathrm{Poi}(\lambda_1 + \lambda_2)$ .*

**Fact C.2.** *Given $\mathrm{Poi}(c)$ samples from a distribution $\boldsymbol{p} \in \Delta_k$, the frequency of any element $j \in [k]$ follows the distribution $\mathrm{Poi}(c \cdot \boldsymbol{p}(j))$.*

We are now ready to prove the following theorem.

**Claim C.3** (Poisson identity testing)**.** *For any $\varepsilon > 0$ and reference distribution $\boldsymbol{q} \in \Delta_k$, with $\mathrm{Poi}(c)$ samples from each of $\boldsymbol{p}_1, \ldots, \boldsymbol{p}_T$ and $T = O(\sqrt{k}/\varepsilon^2)$, there exists a testing algorithm that succeeds with probability at least $2/3$ and tests whether,*

$$\boldsymbol{p}_{\mathrm{avg}} = \boldsymbol{q} \quad \textit{versus} \quad \mathrm{d_{TV}}(\boldsymbol{p}_{\mathrm{avg}}, \boldsymbol{q}) \ge \varepsilon \;,$$

*for any $c \ge 1$.*

*Proof.* Note that we are given $\mathrm{Poi}(c)$ samples from each distribution $\mathbf{p}_t$ for all $t \in [T]$. For each $t \in [T]$, let $N_t$ denote the number of samples drawn from $\mathbf{p}_t$. Note that $N_t \sim \mathrm{Poi}(c)$. For $i \in [k]$, let $N_t(i)$ be the random variable that denotes the frequency of element $i$ in samples $X_t$, that is, $N_t(i) = |\{j \in [N_t] \; : \; X_t(j) = i\}|$. By Fact C.2, we know that

$$N_t(i) \sim \mathrm{Poi}(c \cdot \mathbf{p}_t(i)) \;.$$

Furthermore, if we let $N(i) = \sum_{t \in [T]} N_t(i)$, then by Fact C.1, we get that,

$$N(i) \sim \mathrm{Poi}\Big(c \cdot \sum_{t \in [T]} \mathbf{p}_t(i)\Big) = \mathrm{Poi}(cT \cdot \mathbf{p}_{\mathrm{avg}}(i)) \ .$$

Therefore, $N(i)$ has the same distribution as the frequency of element $i$ when we are given $\mathrm{Poi}(cT)$ i.i.d samples from distribution $\mathbf{p}_{\mathrm{avg}}$. As $cT \in O(\sqrt{k}/\varepsilon^2)$, and since we just argued that we are under the setting of having drawn $\mathrm{Poi}(cT)$ samples from $\mathbf{p}_{\mathrm{avg}}$, we can invoke existing identity testing algorithms that use Poissonization for the i.i.d setting to solve the problem with success probability $2/3$, even with $c = 1$. $\qquad\square$

## D  Collisions and Fingerprints

We relate the collision statistics and the fingerprint of a sample by an invertible linear map. To recall, the number of $m$-way collisions in a sample $S = \{X_1, \ldots, X_n\}$ is equal to $\sum_{i_1 < \cdots < i_m}^{n} 1[X_{i_1} = \cdots = X_{i_m}]$. The fingerprint of the sample is a count array, where for each $i$, we list the count of the number of support elements that appeared exactly $i$ times in $S$.

**Lemma 14** (Collisions to fingerprint). *Let $x_1, x_2, x_3, x_4 \in [m]^4$ for $m > 4$. Let $c_2, c_3, c_4$ denote the number of 2-way, 3-way and 4-way collisions amongst $x_1, x_2, x_3, x_4$. Further, let $n_2, n_3, n_4$ denote the number of elements in $[m]$ occurring exactly 2 times, 3 times and 4 times respectively in $x_1, x_2, x_3, x_4$. Then, the vectors $\begin{bmatrix} c_2 \\ c_3 \\ c_4 \end{bmatrix}$ and $\begin{bmatrix} n_2 \\ n_3 \\ n_4 \end{bmatrix}$ are related by an invertible linear map. In particular,*

$$\begin{bmatrix} n_2 \\ n_3 \\ n_4 \end{bmatrix} = \begin{bmatrix} 1 & -3 & 6 \\ 0 & 1 & -4 \\ 0 & 0 & 1 \end{bmatrix} \begin{bmatrix} c_2 \\ c_3 \\ c_4 \end{bmatrix}.$$

*Proof.* Observe that

$$n_4 = c_4, \quad n_3 = c_3 - \binom{4}{3} n_4 = c_3 - 4c_4$$

$$n_2 = c_2 - \binom{3}{2} n_3 - \binom{4}{2} n_4 = c_2 - 3(c_3 - 4c_4) - 6c_4 = c_2 - 3c_3 + 6c_4.$$

The matrix associated with the linear map has determinant 1, therefore the map is invertible. $\qquad\square$

## E  Sufficiency of fingerprint for pooling-based testing algorithms

We first formally state the definition of a pooling-based testing algorithm.

**Definition E.1** (Pooling-based testing algorithm). Fix $c$ and let $\pi \sim \mathrm{Perm}(cT)$ denote drawing a uniformly random permutation $\pi$ over $cT$ elements. A pooling-based testing algorithm $\mathcal{A}$ for property $\mathcal{P} \subseteq \Delta_k$ with sample complexity $cT$ is a (possibly randomized) algorithm which, when given a randomly shuffled pooled multiset $S$ of $c$ samples drawn independently from each of $\mathbf{p}_1, \ldots, \mathbf{p}_T$, i.e., $S = \bigcup_{t=1}^{T} \{X_1^{(t)}, \ldots, X_c^{(t)} \sim \mathbf{p}_t\}$, $\pi \sim \mathrm{Perm}(cT)$, $S \leftarrow \pi(S)$, produces output $\mathcal{A}(S) \in \{0, 1\}$ satisfying the following

- If $\mathbf{p}_{\mathrm{avg}} \in \mathcal{P}$, then $\Pr_{S, \pi, \mathcal{A}}[\mathcal{A}[S] = 1] \geq 2/3$.

- If $\mathrm{d}_{\mathrm{TV}}(\mathbf{p}_{\mathrm{avg}}, \mathcal{P}) \geq \varepsilon$, then $\Pr_{S, \pi, \mathcal{A}}[\mathcal{A}[S] = 0] \geq 2/3$.

The following lemma states that the fingerprint of the combined sample captures all the information required for a pooling-based testing algorithm to test a symmetric property of the average distribution.

**Lemma 15.** *Let $\mathcal{A}$ be a pooling-based testing algorithm having sample complexity $cT$ that tests a symmetric property $\mathcal{P} \subseteq \Delta_k$ of the average distribution $\boldsymbol{p}_{\mathrm{avg}} = \mathrm{avg}(\boldsymbol{p}_1, \ldots, \boldsymbol{p}_T)$. Then, there exists a pooling-based algorithm $\mathcal{A}'$ for the same task having the same sample complexity which takes as input only the fingerprint of the combined sample.*

*Proof.* Let the fingerprint of the combined sample $S$ be denoted by $f = [f_0, f_1, \ldots, f_{cT}]$. Here, $f_i$ is the number of elements in $[k]$ that occur exactly $i$ times in the combined sample. Given as input $f$, the algorithm $\mathcal{A}'$ constructs a sequence $S$ in the following way:

1. Initially, $S = \phi$, $e = 1$.

2. For $i = 0, \ldots, cT$ :

   > While $f_i > 0$ :
   >> Append $i$ copies of $e$ to $S$.
   >> $e \leftarrow e + 1$, $f_i \leftarrow f_i - 1$.

3. Draw $\pi_k \sim \mathrm{Perm}(k)$.

4. For every $s \in S$:

   > Set $s \leftarrow \pi_k(s)$.

5. Draw $\pi_S \sim \mathrm{Perm}(cT)$.

6. Set $S \leftarrow \pi_S(S)$.

The algorithm $\mathcal{A}'$ then feeds $S$ constructed as above to $\mathcal{A}$, and returns $\mathcal{A}$'s output.

Observe that the distribution of $S$ constructed as above is identical to the distribution of $S$ obtained in the following way:

1. Draw $\pi_k \sim \mathrm{Perm}(k)$.

2. Draw $S = \bigcup_{t=1}^{T} \{X_1^{(t)}, \ldots, X_c^{(t)} \sim \pi_k(\mathbf{p}_t)\}$.

3. Draw $\pi_S \sim \mathrm{Perm}(cT)$.

4. Set $S \leftarrow \pi_S(S)$.

Here, $\pi(\mathbf{p})$ denotes permuting the probability distribution $\mathbf{p}$ according to $\pi$ i.e. $\pi(\mathbf{p})_i = \mathbf{p}_{\pi(i)}$. Now, for a fixed permutation $\pi_k$, observe that $\mathrm{avg}(\pi_k(\mathbf{p}_1), \ldots, \pi_k(\mathbf{p}_T)) = \pi_k(\mathrm{avg}(\mathbf{p}_1, \ldots, \mathbf{p}_T)) = \pi_k(\mathbf{p}_{\mathrm{avg}})$. Further, recall that the property $\mathcal{P}$ we are thinking about is symmetric. Thus, if $\mathbf{p}_{\mathrm{avg}} \in \mathcal{P}$, then $\pi_k(\mathbf{p}_{\mathrm{avg}}) \in \mathcal{P}$ for any $\pi_k \in \mathrm{Perm}(k)$. Similarly, if $\mathbf{p}_{\mathrm{avg}} \notin \mathcal{P}$, then $\pi_k(\mathbf{p}_{\mathrm{avg}}) \notin \mathcal{P}$ for any $\pi_k \in \mathrm{Perm}(k)$. Thus, for any fixed $\pi_k$, $\mathcal{A}$ correctly tests if $\mathrm{avg}(\pi_k(\mathbf{p}_1), \ldots, \pi_k(\mathbf{p}_T))$ has the property $\mathcal{P}$ ( $\iff$ $\mathrm{avg}(\mathbf{p}_1, \ldots, \mathbf{p}_T)$ has the property $\mathcal{P}$). Consequently, $\mathcal{A}$ (and hence $\mathcal{A}'$) correctly tests for a randomly chosen $\pi_k$ as well.

$\square$

# F   Technical Results for Section 5

We prove a technical lemma that bounds the TV distance between Gaussians having the same mean and similar covariances.

**Lemma 16.** *Let A and D be defined as follows:*

$$A = \begin{bmatrix} \alpha_{11}/m^2 & \alpha_{12}/m^3 & \alpha_{13}/m^4 \\ \alpha_{12}/m^3 & \alpha_{22}/m^3 & \alpha_{23}/m^4 \\ \alpha_{13}/m^4 & \alpha_{23}/m^4 & \alpha_{33}/m^4 \end{bmatrix} \text{ and matrix } D = \begin{bmatrix} 1/m^2 & 0 & 0 \\ 0 & 1/m^3 & 0 \\ 0 & 0 & 1/m^4 \end{bmatrix},$$

*where $\alpha_{ij}$ are bounded by constants independent of m. For large enough m, there exists a positive constant $\alpha$ such that,*

$$-\alpha D \preceq A \preceq \alpha D.$$

*Proof.* Let $v_1 = [1, 1, 0]$, $v_2 = [1, 0, 1]$ and $v_3 = [0, 1, 1]$. Note that,

$$A = \frac{\alpha_{12}}{m^3} v_1 v_1^T + \frac{\alpha_{13}}{m^4} v_2 v_2^T + \frac{\alpha_{23}}{m^4} v_3 v_3^T + D',$$

where $D'$ is a diagonal matrix, whose entries are as follows:

$$D' = \begin{bmatrix} \frac{\alpha_{11}}{m^2} - \frac{\alpha_{12}}{m^3} - \frac{\alpha_{13}}{m^4} & 0 & 0 \\ 0 & \frac{\alpha_{22}}{m^3} - \frac{\alpha_{12}}{m^3} - \frac{\alpha_{23}}{m^4} & 0 \\ 0 & 0 & \frac{\alpha_{33}}{m^4} - \frac{\alpha_{13}}{m^4} - \frac{\alpha_{23}}{m^4} \end{bmatrix} \tag{14}$$

Note that, it is immediate that, $v_1 v_1^T \preceq 2D_1$, $v_2 v_2^T \preceq 2D_2$ and $v_3 v_3^T \preceq 2D_3$, where matrices $D_1, D_2$ and $D_3$ are defined as follows,

$$D_1 = \begin{bmatrix} 1 & 0 & 0 \\ 0 & 1 & 0 \\ 0 & 0 & 0 \end{bmatrix}, \quad D_2 = \begin{bmatrix} 1 & 0 & 0 \\ 0 & 0 & 0 \\ 0 & 0 & 1 \end{bmatrix} \text{ and } D_3 = \begin{bmatrix} 0 & 0 & 0 \\ 0 & 1 & 0 \\ 0 & 0 & 1 \end{bmatrix}.$$

This is because for any $x \in \mathbb{R}^3$, $x^\top v_1 v_1^\top x = (x_1 + x_2)^2 \le 2(x_1^2 + x_2^2) = 2\, x^\top D_1 x$, and similarly $(x_1 + x_3)^2 \le 2(x_1^2 + x_3^2)$ and $(x_2 + x_3)^2 \le 2(x_2^2 + x_3^2)$.

Furthermore, as all $\alpha_{ij}$s are bounded independent of $m$, we have that the matrix $D'$ satisfies, $D' \preceq c \cdot D$ for some positive constant $c$.

Combining all the above inequalities we get that,

$$A = \frac{\alpha_{12}}{m^3} v_1 v_1^T + \frac{\alpha_{13}}{m^4} v_2 v_2^T + \frac{\alpha_{23}}{m^4} v_3 v_3^T + D' \preceq \frac{2|\alpha_{12}|}{m^3} D_1 + \frac{2|\alpha_{13}|}{m^4} D_2 + \frac{2|\alpha_{23}|}{m^4} D_3 + c \cdot D .$$

The matrix in the final expression simplifies to,

$$\begin{bmatrix} \frac{c}{m^2} + \frac{2|\alpha_{12}|}{m^3} + \frac{2|\alpha_{13}|}{m^4} & 0 & 0 \\ 0 & \frac{c}{m^3} + \frac{2|\alpha_{12}|}{m^3} + \frac{2|\alpha_{23}|}{m^4} & 0 \\ 0 & 0 & \frac{c}{m^4} + \frac{2|\alpha_{13}|}{m^4} + \frac{2|\alpha_{23}|}{m^4} \end{bmatrix}$$

As all $\alpha_{ij}$s are bounded by constants, we have that the above matrix is upper bounded by $\alpha \cdot D$ for some large constant $\alpha > 0$. Therefore $A \preceq \alpha \cdot D$.

The proof for the other side, that is $-\alpha \cdot D \preceq A$ follows exactly the same argument when applied to $-A$ and we conclude the proof. □

**Lemma 17.** *Let $G_1$ be distributed as $N(\mu, \Sigma_1)$ and $G_2$ be distributed as $N(\mu, \Sigma_2)$ where*

$$\Sigma_1 = \begin{bmatrix} \gamma_{11}/m & \gamma_{12}/m^2 & \gamma_{13}/m^3 \\ \gamma_{12}/m^2 & \gamma_{22}/m^2 & \gamma_{23}/m^3 \\ \gamma_{13}/m^3 & \gamma_{23}/m^3 & \gamma_{33}/m^3 \end{bmatrix} \text{ and } \Sigma_2 = \Sigma_1 + \begin{bmatrix} \alpha_{11}/m^2 & \alpha_{12}/m^3 & \alpha_{13}/m^4 \\ \alpha_{12}/m^3 & \alpha_{22}/m^3 & \alpha_{23}/m^4 \\ \alpha_{13}/m^4 & \alpha_{23}/m^4 & \alpha_{33}/m^4 \end{bmatrix},$$

*for $\gamma_{ij}$ and $\alpha_{ij}$ such that $|\gamma_{ij}|$ and $|\alpha_{ij}|$ are bounded by constants independent of m. Then $d_{\mathrm{TV}}(G_1, G_2) \le 0.01$, for m large enough.*

*Proof.* In our proof, we show that, for a sufficiently large $m$, the covariance matrices $\Sigma_1$ and $\Sigma_2$ satisfy: $(1 - \alpha/m)\Sigma_1 \preceq \Sigma_2 \preceq (1 + \alpha/m)\Sigma_1$ for some constant $\alpha > 0$. Equivalently, all the eigenvalues of the matrix $L^{-1}\Sigma_2(L^T)^{-1}$ lie in the range $[1 - \alpha/m, 1 + \alpha/m]$, where $\Sigma_1 = LL^T$ is the Cholesky decomposition of matrix $\Sigma_1$. Note that, by the TV distance bound in Fact F.3 this result immediately implies the following upper bound on $d_{\mathrm{TV}}(G_1, G_2)$ between the two Gaussian distributions ,

$$d_{\mathrm{TV}}(G_1, G_2) \leq \sum_{i=1}^{3} \frac{\max(\lambda_i, \lambda_i^{-1}) - 1}{\sqrt{2\pi e}} \leq \sum_{i=1}^{3} \frac{O(1/m)}{\sqrt{2\pi e}} \leq O(1/m) \ .$$

Here, while applying Fact F.3, we used that $G_1$ and $G_2$ have the same mean. Now note that, for a sufficiently large $m$, we get the desired upper bound on the $d_{\mathrm{TV}}(G_1, G_2)$.

Therefore to prove our lemma, all that remains is to show that, $(1 - \alpha/m)\Sigma_1 \preceq \Sigma_2 \preceq (1 + \alpha/m)\Sigma_1$ for some constant $\alpha > 0$. In the remainder of the proof, we prove this claim. First we show invertibility of $\Sigma_1$. Since $\Sigma_1$ is a covariance matrix in our construction, the diagonal coefficients $\gamma_{11}, \gamma_{22}, \gamma_{33}$ are fixed positive constants (independent of $m$). Thus the diagonal permutation contributes $\gamma_{11}\gamma_{22}\gamma_{33}/m^6$ to $\det(\Sigma_1)$, while any other permutation includes at least one off-diagonal of order $1/m^2$ or $1/m^3$ and hence has magnitude $O(1/m^7)$. Therefore, for $m$ large enough,

$$\det(\Sigma_1) \ \geq \ \frac{\gamma_{11}\gamma_{22}\gamma_{33}}{2\, m^6} \ > \ 0,$$

so $\Sigma_1$ is invertible.

Now let

$$E = \begin{bmatrix} \alpha_{11}/m^2 & \alpha_{12}/m^3 & \alpha_{13}/m^4 \\ \alpha_{12}/m^3 & \alpha_{22}/m^3 & \alpha_{23}/m^4 \\ \alpha_{13}/m^4 & \alpha_{23}/m^4 & \alpha_{33}/m^4 \end{bmatrix} \tag{15}$$

To prove the claim, we in turn show that, $-\frac{\alpha}{m}\Sigma_1 \preceq E \preceq \frac{\alpha}{m}\Sigma_1$.

By Lemma 16, we know that there exist constants $\alpha_1$ and $\alpha_2$ such that, $-\alpha_1 D \preceq E \preceq \alpha_1 D$ and $0 \preceq \Sigma_1 \preceq \alpha_2 m D$, where

$$D = \begin{bmatrix} 1/m^2 & 0 & 0 \\ 0 & 1/m^3 & 0 \\ 0 & 0 & 1/m^4 \end{bmatrix}$$

Here the bound $0 \preceq \Sigma_1 \preceq \alpha_2 m D$ follows by applying Lemma 16 to $m^{-1}\Sigma_1$, which has the same $(1/m^2, 1/m^3, 1/m^4)$ structure with coefficients $\alpha_{ij} = \gamma_{ij}$.

In the following, we prove $E \preceq \frac{\alpha}{m}\Sigma_1$ for some constant $\alpha > 0$. We first show that $D \preceq \frac{\gamma}{m}\Sigma_1$ for some constant $\gamma$, as $E \preceq \alpha_1 D$, we conclude that $E \preceq \frac{\alpha}{m}\Sigma_1$. Towards this end, we now prove that $D \preceq \frac{\gamma}{m}\Sigma_1$ for some constant $\gamma$. Equivalently, we wish to show that the smallest eigenvalue of the matrix $F = D^{-1/2}\frac{1}{m}\Sigma_1 D^{-1/2}$ is lower bounded by some constant. Note that $F$ is PSD and is equal to,

$$F = \begin{bmatrix} \gamma_{11} & \gamma_{12}/m^{1/2} & \gamma_{13}/m \\ \gamma_{12}/m^{1/2} & \gamma_{22} & \gamma_{23}/m^{1/2} \\ \gamma_{13}/m & \gamma_{23}/m^{1/2} & \gamma_{33} \end{bmatrix} \ .$$

Its trace is $\mathrm{Tr}(F) = \gamma_{11} + \gamma_{22} + \gamma_{33} = \Theta(1)$. Moreover, the determinant satisfies

$$\det(F) \ \geq \ \frac{\gamma_{11}\gamma_{22}\gamma_{33}}{2} \ =: \ c_0 \ > 0$$

for all sufficiently large $m$ (the diagonal term dominates, while all off-diagonal permutations are $o(1)$). By Gershgorin circle theorem (Fact F.1), the spectral norm is bounded:

$$\lambda_{\max}(F) \ \leq \ \max_i \left( \gamma_{ii} + \sum_{j \neq i} \frac{|\gamma_{ij}|}{m^{1/2}} \right) \ \leq \ C_0,$$

for some constant $C_0$ and all large $m$. Therefore, with eigenvalues $\lambda_1 \geq \lambda_2 \geq \lambda_3 \geq 0$,

$$\lambda_{\min}(F) = \lambda_3 \;\geq\; \frac{\det(F)}{\lambda_1 \lambda_2} \;\geq\; \frac{c_0}{C_0^2} \;=:\; c_1 \;>\; 0.$$

Equivalently, $D \;\preceq\; \frac{1}{c_1}\frac{1}{m}\Sigma_1$, i.e. $D \preceq \frac{\gamma}{m}\Sigma_1$ with $\gamma = 1/c_1$. Hence $E \preceq \alpha_1 D \preceq \frac{\alpha_1 \gamma}{m}\Sigma_1$, i.e. $E \preceq \frac{\alpha}{m}\Sigma_1$ for $\alpha := \alpha_1 \gamma$.

The proof for the other side, that is $-\frac{\alpha}{m} \cdot \Sigma_1 \preceq E$ follows exactly the same argument when applied to $-E$ and we conclude the proof. $\qquad\square$

**Lemma 18** (Discretization–convolution almost commute for Gaussians)**.** *Let $X \sim \mathcal{N}(\mu_X, \Sigma_X)$ and $Y \sim \mathcal{N}(\mu_Y, \Sigma_Y)$ be independent in $\mathbb{R}^d$ (with d constant), and assume at least one of $\Sigma_X, \Sigma_Y$ is positive definite.*[2] *Let $\mathrm{Disc}(\cdot)$ be coordinate-wise rounding to the nearest integer (ties are immaterial since Gaussians put zero mass on boundaries). Then*

$$d_{\mathrm{TV}}\big(\mathrm{Disc}(X) + \mathrm{Disc}(Y),\, \mathrm{Disc}(X+Y)\big) \;\leq\; \frac{\sqrt{d}}{4}\,\min\Big\{\sqrt{\mathrm{Tr}(\Sigma_X^{-1})},\ \sqrt{\mathrm{Tr}(\Sigma_Y^{-1})}\Big\}.$$

*In particular, if $\lambda_{\min}(\Sigma_X) \to \infty$ or $\lambda_{\min}(\Sigma_Y) \to \infty$, the left-hand side is $O(1/\sqrt{\lambda_{\min}})$ and tends to 0.*

*Proof. Without loss of generality assume $\Sigma_Y \succ 0$; otherwise swap $X$ and $Y$. Let $f$ and $g$ be the densities of $X$ and $Y$. For $m \in \mathbb{Z}^d$, write $C_m := m + [-\frac{1}{2}, \frac{1}{2})^d$. Define*

$$P(m) := \Pr(\mathrm{Disc}(X) + \mathrm{Disc}(Y) = m) = \sum_{a \in \mathbb{Z}^d} \Big( \int_{C_a} f(x)\,dx \Big)\Big( \int_{C_{m-a}} g(y)\,dy \Big),$$

$$Q(m) := \Pr(\mathrm{Disc}(X+Y) = m) = \int_{C_m} \int_{\mathbb{R}^d} f(x)\,g(z-x)\,dx\,dz.$$

For $x \in C_a$, set $\delta(x) := a - x \in [-\frac{1}{2}, \frac{1}{2}]^d$. Changing variables $z = y + a$ in $Q(m)$ yields

$$Q(m) = \sum_{a \in \mathbb{Z}^d} \int_{C_a} \Big( \int_{C_{m-a}} g\big(y + \delta(x)\big)\,dy \Big) f(x)\,dx,$$

and hence

$$P(m) - Q(m) = \sum_{a \in \mathbb{Z}^d} \int_{C_a} \Big( \int_{C_{m-a}} \big[g(y) - g\big(y + \delta(x)\big)\big]\,dy \Big) f(x)\,dx. \tag{16}$$

**Bounding $\sum_m |P(m) - Q(m)|$ by an expectation times a norm.** Summing equation 16 in absolute value over $m$ and using that the cubes $\{C_{m-a}\}_m$ are disjoint and partition $\mathbb{R}^d$,

$$\sum_{m \in \mathbb{Z}^d} |P(m) - Q(m)| \leq \sum_a \int_{C_a} f(x)\, \Big( \int_{\mathbb{R}^d} \big|g(y) - g(y + \delta(x))\big|\,dy \Big)\,dx$$

$$= \int_{\mathbb{R}^d} f(x)\,\big\|g(\cdot) - g(\cdot + \delta(x))\big\|_{L^1}\,dx. \tag{17}$$

(*We used $\sum_m \big| \int_{C_{m-a}} \cdot \big| \leq \sum_m \int_{C_{m-a}} |\cdot| = \int_{\mathbb{R}^d} |\cdot|$, since $\{C_{m-a}\}_m$ partition $\mathbb{R}^d$.*)

Apply Lemma 19 with $h = g$ and $u = \delta(x)$ pointwise in $x$ to get

$$\big\|g(\cdot) - g(\cdot + \delta(x))\big\|_{L^1} \;\leq\; \|\delta(x)\|_2\,\|\nabla g\|_{L^1}. \tag{18}$$

---

[2]If one covariance is singular, the bound below with that matrix should be read as $+\infty$; the inequality is then applied with the other (positive definite) covariance.

Plugging equation 18 into equation 17 and factoring out the constant $\|\nabla g\|_{L^1}$ yields

$$\sum_{m\in\mathbb{Z}^d} |P(m) - Q(m)| \leq \left(\mathbb{E}\big[\|\delta(X)\|_2\big]\right) \cdot \|\nabla g\|_{L^1}. \tag{19}$$

Since $X \in C_{\mathrm{Disc}(X)}$ almost surely, $\delta(X) = \mathrm{Disc}(X) - X \in [-\tfrac{1}{2}, \tfrac{1}{2}]^d$, and thus

$$\mathbb{E}\big[\|\delta(X)\|_2\big] \leq \sup_{\|\delta\|_\infty \leq 1/2} \|\delta\|_2 = \frac{\sqrt{d}}{2}. \tag{20}$$

**Computing $\|\nabla g\|_{L^1}$.** For $Y \sim \mathcal{N}(\mu_Y, \Sigma_Y)$,

$$\nabla g(y) = -\Sigma_Y^{-1}(y - \mu_Y)\,g(y), \qquad \|\nabla g\|_{L^1} = \int \|\Sigma_Y^{-1}(y - \mu_Y)\|_2\,g(y)\,dy = \mathbb{E}\big[\|\Sigma_Y^{-1/2}Z\|_2\big],$$

where $Z \sim \mathcal{N}(0, I_d)$. By Cauchy–Schwarz,

$$\|\nabla g\|_{L^1} \leq \sqrt{\mathbb{E}\big[Z^\top \Sigma_Y^{-1} Z\big]} = \sqrt{\mathrm{Tr}(\Sigma_Y^{-1})}. \tag{21}$$

Combining equation 19, equation 20, and equation 21,

$$\sum_{m\in\mathbb{Z}^d} |P(m) - Q(m)| \leq \frac{\sqrt{d}}{2}\sqrt{\mathrm{Tr}(\Sigma_Y^{-1})},$$

hence

$$\mathrm{d}_{\mathrm{TV}}\big(\mathrm{Disc}(X) + \mathrm{Disc}(Y),\ \mathrm{Disc}(X + Y)\big) = \tfrac{1}{2}\sum_m |P(m) - Q(m)| \leq \frac{\sqrt{d}}{4}\sqrt{\mathrm{Tr}(\Sigma_Y^{-1})}.$$

By symmetry, the same bound holds with $\Sigma_X$; taking the minimum proves the claim. $\qquad\square$

**Lemma 19.** *Let $h : \mathbb{R}^d \to \mathbb{R}_+$ be locally absolutely continuous with $\nabla h \in L^1(\mathbb{R}^d)$. Then for every $u \in \mathbb{R}^d$,*

$$\|h(\cdot + u) - h(\cdot)\|_{L^1} \leq \|u\|_2\,\|\nabla h\|_{L^1}.$$

*Proof.* Fix $y \in \mathbb{R}^d$ and consider the path $t \mapsto y + t\,u$. By the fundamental theorem of calculus,

$$h(y + u) - h(y) = \int_0^1 \langle \nabla h(y + t\,u), u\rangle\,dt,$$

so $|h(y + u) - h(y)| \leq \|u\|_2 \int_0^1 \|\nabla h(y + t\,u)\|_2\,dt$ by Cauchy–Schwarz. Integrating in $y$ and using Fubini with the change of variables $y \mapsto y + t\,u$ (unit Jacobian) gives

$$\int_{\mathbb{R}^d} |h(y + u) - h(y)|\,dy \leq \|u\|_2 \int_0^1 \int_{\mathbb{R}^d} \|\nabla h(y + t\,u)\|_2\,dy\,dt = \|u\|_2\,\|\nabla h\|_{L^1}.$$

$\qquad\square$

**Lemma 20.** *Let $X_1$ and $X_2$ be independent with distributions $\mathcal{N}(\mu_1, \sigma_1^2)$ and $\mathcal{N}(\mu_2, \sigma_2^2)$, respectively. Let $t := \max(\sigma_1^2, \sigma_2^2)$. Then, for $t$ large enough,*

$$\mathrm{d}_{\mathrm{TV}}\big(\mathrm{Disc}(X_1) + \mathrm{Disc}(X_2),\ \mathrm{Disc}(X_1 + X_2)\big) \leq 0.00001.$$

*Proof.* Apply Lemma 18 with $d = 1$:

$$\mathrm{d}_{\mathrm{TV}}\big(\mathrm{Disc}(X_1) + \mathrm{Disc}(X_2),\ \mathrm{Disc}(X_1 + X_2)\big) \leq \frac{1}{4}\min\Big\{\tfrac{1}{\sigma_1}, \tfrac{1}{\sigma_2}\Big\} = \frac{1}{4\sqrt{t}}.$$

Thus the bound is $\leq 10^{-5}$ as soon as $\sqrt{t} \geq 25{,}000$, i.e. $t \geq 6.25 \times 10^8$. $\qquad\square$

**Lemma 21.** *Let $X$ and $Y$ be $n$-dimensional independent random variables, where $X$ is multivariate Gaussian and $Y$ has first coordinate distributed as $\mathcal{N}(0, t)$ and all other coordinates identically $0$. Then, for $t$ large enough,*

$$d_{\mathrm{TV}}\big(\mathrm{Disc}(X) + \mathrm{Disc}(Y), \ \mathrm{Disc}(X + Y)\big) \leq 0.00001,$$

*where $\mathrm{Disc}(\cdot)$ denotes coordinate-wise discretization to the nearest integer.*

*Proof.* The idea is that $Y$ perturbs only the first coordinate. Let $X_{-1}$ denote the last $n-1$ coordinates of $X$. After discretization, the last $(n-1)$ coordinates of $\mathrm{Disc}(X) + \mathrm{Disc}(Y)$ equal $\mathrm{Disc}(X_{-1})$, and those of $\mathrm{Disc}(X+Y)$ equal $\mathrm{Disc}(X_{-1})$ as well. Thus, conditioned on $X_{-1}$, the event $\{\mathrm{Disc}(X) + \mathrm{Disc}(Y) \in A\}$ depends only on the first coordinate, and likewise for $\{\mathrm{Disc}(X + Y) \in A\}$. We therefore reduce to the one-dimensional case of Lemma 18 on the first coordinate and then average over $X_{-1}$. We formalize this below.

Write $X = (X_1, X_{-1})$ and $Y = (Y_1, 0, \dots, 0)$ with $Y_1 \sim \mathcal{N}(0, t)$ independent of $X$. For any $A \subseteq \mathbb{Z}^n$ and $u \in \mathbb{R}^{n-1}$, define

$$A_u := \{m_1 \in \mathbb{Z} : \ (m_1, \mathrm{Disc}(u)) \in A\}.$$

Condition on $X_{-1}$ and apply Lemma 18 in dimension 1 (to the pair $(X_1, Y_1)$) to get

$$\left| \Pr\big(\mathrm{Disc}(X_1) + \mathrm{Disc}(Y_1) \in A_{X_{-1}} \mid X_{-1}\big) - \Pr\big(\mathrm{Disc}(X_1 + Y_1) \in A_{X_{-1}} \mid X_{-1}\big) \right| \leq \frac{1}{4\sqrt{t}}.$$

In the above, we conditioned on $X_{-1} = u$, which freezes the last $n-1$ coordinates; thus $(X_1, Y_1)$ is one–dimensional with $Y_1 \sim \mathcal{N}(0, t)$ independent of $X_1$. We then applied Lemma 18 in $d = 1$, yielding a bound $\leq 1/(4\sqrt{t})$ uniformly in $u$ since the right-hand side depends only on $t$.

Averaging over $X_{-1}$ and taking the supremum over $A$ therefore gives

$$d_{\mathrm{TV}}\big(\mathrm{Disc}(X) + \mathrm{Disc}(Y), \ \mathrm{Disc}(X + Y)\big) \ \leq \ \frac{1}{4\sqrt{t}} \ \leq \ 10^{-5} \quad \text{for } t \geq 6.25 \times 10^8.$$

$\square$

**Lemma 22.** *Let $X$ be a sample from a distribution supported on $\{[0, 0, 0], e_1, e_2, e_3, 2e_1\}$ with probabilities $u_0, u_1, u_2, u_3, v$ such that $u_1, u_2, u_3, v > 0$ and $u_0 \geq 1 - 10^{-16}$ (all parameters independent of $n$). Let $\Sigma_1 = \mathrm{Cov}(X)$ and let $\Sigma_{\alpha n}$ be the covariance of the sum of $\alpha n$ i.i.d. copies of $X$. Then there exist constants $c_\star, C_\star > 0$ depending only on $(u_1, u_2, u_3, v)$ (hence independent of $n$) such that*

$$c_\star \alpha n \ \leq \ \lambda_{\min}(\Sigma_{\alpha n}) \ \leq \ \lambda_{\max}(\Sigma_{\alpha n}) \ \leq \ C_\star \alpha n.$$

*In particular, $\lambda_{\min}(\Sigma_{\alpha n}) = \Theta(n)$.*

*Proof.* Write $\mu = \mathbb{E}[X] = (u_1 + 2v, \ u_2, \ u_3)$ and $D = \mathbb{E}[XX^\top] = \mathrm{diag}(u_1 + 4v, \ u_2, \ u_3)$, so $\Sigma_1 = D - \mu\mu^\top$. By the matrix determinant lemma,

$$\det(\Sigma_1) = \det(D)\big(1 - \mu^\top D^{-1}\mu\big) = (u_1 + 4v)u_2 u_3 \left(1 - \frac{(u_1 + 2v)^2}{u_1 + 4v} - u_2 - u_3\right) \geq (u_1 + 4v)u_2 u_3\,(u_0 - v) =: c_{\det} > 0,$$

using $\frac{(u_1 + 2v)^2}{u_1 + 4v} \leq u_1 + 2v$ and $u_0 = 1 - (u_1 + u_2 + u_3 + v)$. Also $\Sigma_1 \preceq D$, hence $\lambda_{\max}(\Sigma_1) \leq \lambda_{\max}(D) = \max\{u_1 + 4v, u_2, u_3\} =: c_D < \infty$. If the eigenvalues of $\Sigma_1$ are $\lambda_1 \geq \lambda_2 \geq \lambda_3 > 0$, then

$$\lambda_{\min}(\Sigma_1) = \lambda_3 \ \geq \ \frac{\det(\Sigma_1)}{\lambda_1 \lambda_2} \ \geq \ \frac{c_{\det}}{c_D^2} =: c_0 > 0.$$

For the sum of $\alpha n$ i.i.d. draws, $\Sigma_{\alpha n} = \alpha n \Sigma_1$, so $\lambda_{\min}(\Sigma_{\alpha n}) = \alpha n \lambda_{\min}(\Sigma_1) \geq \alpha c_0 n$ and $\lambda_{\max}(\Sigma_{\alpha n}) \leq \alpha n \lambda_{\max}(\Sigma_1) \leq \alpha c_D n$. Take $c_\star := c_0$ and $C_\star := c_D$ to complete the proof. $\square$

**Lemma 23.** *Let $X_1 \sim \mathcal{N}(\overline{\mu}_1, \overline{\Sigma}_1)$, $X_2 \sim \mathcal{N}(\overline{\mu}_2, \overline{\Sigma}_2)$ and $X_3 \sim \mathcal{N}(\overline{\mu}_3, \overline{\Sigma}_3)$ be c-dimensional Gaussians, for $c \leq 10$, where*

$$\overline{\mu}_1 = \begin{bmatrix} 2u_1 \\ 2u_2 \\ \vdots \\ 2u_c \end{bmatrix}, \qquad \overline{\Sigma}_1 = \begin{bmatrix} 2u_1(1 - 2u_1) & -4u_1u_2 & \dots & -4u_1u_c \\ -4u_1u_2 & 2u_2(1 - 2u_2) & \dots & -4u_2u_c \\ \vdots & & \dots & \\ -4u_1u_c & -4u_2u_c & \dots & 2u_c(1 - 2u_c) \end{bmatrix}$$

$$\overline{\mu}_2 = \begin{bmatrix} 4v \\ 0 \\ \vdots \\ 0 \end{bmatrix}, \qquad \overline{\Sigma}_2 = \begin{bmatrix} 8v(1 - 2v) & 0 & \dots & 0 \\ 0 & 0 & \dots & 0 \\ \vdots & & \dots & \\ 0 & 0 & \dots & 0 \end{bmatrix}$$

$$\overline{\mu}_3 = \begin{bmatrix} u_1 + 2v \\ u_2 \\ \vdots \\ u_c \end{bmatrix}, \qquad \overline{\Sigma}_3 = \begin{bmatrix} u_1 + 4v - (u_1 + 2v)^2 & -(u_1 + 2v)u_2 & \dots & -(u_1 + 2v)u_c \\ -(u_1 + 2v)u_2 & u_2(1 - u_2) & \dots & -u_2u_c \\ \vdots & & \dots & \\ -(u_1 + 2v)u_c & -u_2u_c & \dots & u_c(1 - u_c) \end{bmatrix},$$

*and furthermore, 1) $v + \sum_i u_i \leq 10^{-16}$, and 2) $u_i, v$ are positive constants. Now, let $Z_1$ be the sum of $l$ independent draws of $X_3$. Let $Z_2$ be the sum of $\frac{l}{2} + n$ independent draws of $X_1$ and $\frac{l}{2} - n$ independent draws of $X_2$. As $l$ gets large, for any $n \in [-4\sqrt{l}, 4\sqrt{l}]$,*

$$\mathrm{d}_{\mathrm{TV}}(Z_1, Z_2) \leq 10^{-4}.$$

*Proof.* First, note that both $Z_1$ and $Z_2$ are also Gaussians, because a linear combination of independent Gaussians is a Gaussian. Let $\mu_1, \Sigma_1$ and $\mu_2, \Sigma_2$ respectively be the mean and covariance of $Z_1$ and $Z_2$. Then,

$$\mu_1 = l\,\overline{\mu}_3,$$
$$\Sigma_1 = l\,\overline{\Sigma}_3$$
$$\mu_2 = (l/2 + n)\overline{\mu}_1 + (l/2 - n)\overline{\mu}_2$$
$$\Sigma_2 = (l/2 + n)\overline{\Sigma}_1 + (l/2 - n)\overline{\Sigma}_2$$

Let $\overline{Z}_1$ be the Gaussian with mean $\mu_1$ and covariance $\mathrm{diag}(\Sigma_1)$, where $\mathrm{diag}(\Sigma_1)$ is simply the matrix $\Sigma_1$ with all non-diagonal entries zeroed out. We will first show that $\mathrm{d}_{\mathrm{TV}}(Z_1, \overline{Z}_1) \leq 10^{-13}$.

Consider the matrix $\mathrm{diag}(\Sigma_1)^{-1/2} \cdot \Sigma_1 \cdot \mathrm{diag}(\Sigma_1)^{-1/2}$. Let us denote this matrix by $M_1$ and it is equal to

$$\begin{bmatrix} 1 & \frac{-(u_1 + 2v)u_2}{\sqrt{v_1 v_2}} & \dots & \frac{-(u_1 + 2v)u_c}{\sqrt{v_1 v_c}} \\ \frac{-(u_1 + 2v)u_2}{\sqrt{v_1 v_2}} & 1 & \dots & \frac{-u_2u_c}{\sqrt{v_2 v_c}} \\ \vdots & & \dots & \\ \frac{-(u_1 + 2v)u_c}{\sqrt{v_1 v_c}} & \frac{-u_2u_c}{\sqrt{v_2 v_c}} & \dots & 1 \end{bmatrix},$$

where $v_1 = u_1 + 4v - (u_1 + 2v)^2$ and $v_i = u_i(1 - u_i)$ for $i \geq 2$. Note that because each $u_i \leq 10^{-16}$ and $v \leq 10^{-16}$,

$$\frac{u_1 + 2v}{\sqrt{v_1}} = \frac{u_1 + 2v}{\sqrt{u_1(1 - u_1) + 4v(1 - v) - 4u_1v}} \leq \frac{u_1}{\sqrt{u_1(1 - u_1)}} + \frac{2v}{\sqrt{4v(1 - v)}} \leq 2(\sqrt{u_1} + \sqrt{v})$$

and also, for $i = 2, \dots, c$,

$$\frac{u_i}{\sqrt{v_i}} = \frac{u_i}{\sqrt{u_i(1 - u_i)}} \leq 2\sqrt{u_i}.$$

Thus, the sum of absolute values of off-diagonal elements in any row $i$ of $M_1$ is at most

$$4\sum_{j\neq i}\sqrt{u_iu_j}+4\sum_j\sqrt{vu_j}\leq 8\cdot c\cdot 10^{-16}\leq 10^{-14}.$$

Then, by the Gershgorin circle theorem (Fact F.1), all the eigenvalues of this matrix are contained in $[1-10^{-14},1+10^{-14}]$. Using corollary 24 that bounds the TV distance between multivariate Gaussian distributions, we get

$$d_{\mathrm{TV}}(Z_1,\overline{Z_1})\leq \frac{10}{\sqrt{2\pi e}}\cdot\left(\max\left(\frac{1}{1-10^{-14}},1+10^{-14}\right)-1\right)\leq 10^{-13}.$$

Now, let $\overline{Z}_2$ be the Gaussian with mean $\mu_2$ and covariance $\mathrm{diag}(\Sigma_1)$. We will next show that $d_{\mathrm{TV}}(Z_2,\overline{Z}_2)\leq 10^{-12}$.

Again, consider the matrix $\mathrm{diag}(\Sigma_1)^{-1/2}\cdot\Sigma_2\cdot\mathrm{diag}(\Sigma_1)^{-1/2}$. Let us denote this matrix by $M_2$ and this is equal to

$$\begin{bmatrix} \frac{\left(1+\frac{2n}{l}\right)\cdot 2u_1(1-2u_1)+\left(1-\frac{2n}{l}\right)\cdot 8v(1-2v)}{2u_1+8v-2(u_1+2v)^2} & -\left(1+\frac{2n}{l}\right)\frac{2u_1u_2}{\sqrt{v_1v_2}} & \cdots & -\left(1+\frac{2n}{l}\right)\frac{2u_1u_c}{\sqrt{v_1v_c}} \\ -\left(1+\frac{2n}{l}\right)\frac{2u_1u_2}{\sqrt{v_1v_2}} & \left(1+\frac{2n}{l}\right)\left(\frac{1-2u_2}{1-u_2}\right) & \cdots & -\left(1+\frac{2n}{l}\right)\frac{2u_2u_c}{\sqrt{v_2v_c}} \\ \vdots & & \cdots & \\ -\left(1+\frac{2n}{l}\right)\frac{2u_1u_c}{\sqrt{v_1v_c}} & -\left(1+\frac{2n}{l}\right)\frac{2u_2u_c}{\sqrt{v_2v_c}} & \cdots & \left(1+\frac{2n}{l}\right)\left(\frac{1-2u_c}{1-u_c}\right) \end{bmatrix},$$

where $v_1,\ldots,v_c$ are the same as above. Again, using a similar calculation as above, the sum of absolute values of off-diagonal entries in any row $i$ is at most

$$2\left|1+\frac{2n}{l}\right|\left(4\sum_{j\neq i}\sqrt{u_iu_j}+4\sum_j\sqrt{vu_j}\right)\leq 2\left|1+\frac{2n}{l}\right|\cdot 10^{-14}\leq 10^{-13},$$

where the last inequality holds for large enough $l$.

Diagonals of $M_2$: For $i=1$, write

$$(M_2)_{11}=\frac{A+\frac{2n}{\ell}B}{D},\quad A:=2u_1(1-2u_1)+8v(1-2v),\ B:=2u_1(1-2u_1)-8v(1-2v),\ D:=2u_1+8v-2(u_1+2v)^2.$$

When $n=0$, a direct expansion gives $A=D-2(u_1-2v)^2$, hence

$$\left|(M_2)_{11}-1\right|=\frac{2(u_1-2v)^2}{D}\leq\frac{2(u_1+2v)^2}{2u_1+8v-2(u_1+2v)^2}\leq 12\cdot 10^{-16}\ll 10^{-13},$$

using $u_1,v\leq 10^{-16}$. For the $n$–dependent term,

$$\left|\frac{2n}{\ell}\frac{B}{D}\right|\leq\frac{2|n|}{\ell}\cdot\frac{2u_1+8v}{2u_1+8v-2(u_1+2v)^2}\leq\frac{16}{\sqrt{\ell}},$$

since $|n|\leq 4\sqrt{\ell}$ and the ratio is $\leq 2$ for these parameters; thus $(M_2)_{11}\in[1-10^{-13},\,1+10^{-13}]$ for $\ell$ large enough. For $i\geq 2$,

$$(M_2)_{ii}=\left(1+\frac{2n}{\ell}\right)\frac{1-2u_i}{1-u_i},\quad\text{so}\quad\left|(M_2)_{ii}-1\right|\leq\frac{8}{\sqrt{\ell}}+\frac{u_i}{1-u_i}\leq\frac{8}{\sqrt{\ell}}+2\cdot 10^{-16}.$$

Hence, for sufficiently large $\ell$, all diagonal entries of $M_2$ lie in $[1-10^{-13},\,1+10^{-13}]$.

Thus by the Gershgorin circle theorem (Fact F.1), all the eigenvalues of this matrix $M_2$ are contained in $[1-2\cdot 10^{-13},1+2\cdot 10^{-13}]$. Again using corollary 24 that bounds the TV distance between multivariate Gaussian distributions, we get

$$d_{\mathrm{TV}}(Z_2,\overline{Z_2})\leq\frac{10}{\sqrt{2\pi e}}\cdot\left(\max\left(\frac{1}{1-2\cdot 10^{-13}},1+2\cdot 10^{-13}\right)-1\right)\leq 10^{-12}.$$

Finally, we bound $\mathrm{d}_{\mathrm{TV}}(\overline{Z}_1, \overline{Z}_2)$. Note that both $\overline{Z}_1$ and $\overline{Z}_2$ are Gaussians with the same diagonal covariance $\mathrm{diag}(\Sigma_1)$, but their means are $\mu_1$ and $\mu_2$ respectively. Since the covariance is diagonal, we can upper-bound the total variation distance between the product distributions by the sum of the total variation distances between the coordinate-wise one-dimensional Gaussians. Namely, denoting by $\sigma_1^2, \ldots, \sigma_c^2$ the $c$ diagonal entries in $\mathrm{diag}(\Sigma_1)$,

$$\mathrm{d}_{\mathrm{TV}}(\overline{Z}_1, \overline{Z}_2) \leq \mathrm{d}_{\mathrm{TV}}(\mathcal{N}(\mu_{11}, \sigma_1^2), \mathcal{N}(\mu_{21}, \sigma_1^2)) + \cdots + \mathrm{d}_{\mathrm{TV}}(\mathcal{N}(\mu_{1c}, \sigma_c^2), \mathcal{N}(\mu_{2c}, \sigma_c^2))$$

$$= \mathrm{d}_{\mathrm{TV}}\left(\mathcal{N}\left(\frac{\mu_{11}}{\sigma_1}, 1\right), \mathcal{N}\left(\frac{\mu_{21}}{\sigma_1}, 1\right)\right) + \cdots + \mathrm{d}_{\mathrm{TV}}\left(\mathcal{N}\left(\frac{\mu_{1c}}{\sigma_c}, 1\right), \mathcal{N}\left(\frac{\mu_{2c}}{\sigma_c}, 1\right)\right)$$

where in the last equality, we used the property that total variation distance is invariant to affine transformations. For each of these, we can use the formula for bounding the total variation distance between one-dimensional Gaussians given in Fact F.2 to get

$$\mathrm{d}_{\mathrm{TV}}(\overline{Z}_1, \overline{Z}_2) \leq \frac{1}{\sqrt{2\pi}} \left(\frac{|\mu_{11} - \mu_{21}|}{\sigma_1} + \cdots + \frac{|\mu_{1c} - \mu_{2c}|}{\sigma_c}\right)$$

$$= \frac{|n|}{\sqrt{l\pi}} \left(\frac{|2u_1 - 4v|}{\sqrt{u_1(1 - 2u_1) + 4v(1 - v) - 4u_1 v}} + \cdots + \frac{2u_c}{\sqrt{u_c(1 - u_c)}}\right)$$

$$\leq \frac{4|n|}{\sqrt{l\pi}} \left(\sqrt{v} + \sqrt{u_1} + \cdots + \sqrt{u_c}\right) \qquad \text{(using } u_i \text{ and } v \text{ at most } 10^{-16})$$

$$\leq \frac{16}{\sqrt{\pi}} \left(\sqrt{v} + \sqrt{u_1} + \cdots + \sqrt{u_c}\right)$$

$$\leq \frac{16}{\sqrt{\pi}} \cdot 11 \cdot 10^{-8}$$

$$\leq 10^{-5}.$$

Putting everything together, we get using the triangle inequality that

$$\mathrm{d}_{\mathrm{TV}}(Z_1, Z_2) \leq \mathrm{d}_{\mathrm{TV}}(Z_1, \overline{Z}_1) + \mathrm{d}_{\mathrm{TV}}(Z_2, \overline{Z}_2) + \mathrm{d}_{\mathrm{TV}}(\overline{Z}_1, \overline{Z}_2)$$

$$\leq 10^{-13} + 10^{-12} + 10^{-5}$$

$$\leq 10^{-4}.$$

$\square$

### F.1 Other useful facts

**Fact F.1** (Gershgorin circle theorem). *(Horn & Johnson, 2012, Theorem 6.1.1) Let $A = (a_{ij}) \in \mathbb{C}^{d \times d}$ and set $R_i := \sum_{j \neq i} |a_{ij}|$. Then all eigenvalues of $A$ lie in*

$$\bigcup_{i=1}^{d} \left\{ z \in \mathbb{C} : |z - a_{ii}| \leq R_i \right\}.$$

*In particular, if $A \in \mathbb{R}^{d \times d}$ is symmetric, all its eigenvalues are real and lie in*

$$\bigcup_{i=1}^{d} [a_{ii} - R_i, \ a_{ii} + R_i].$$

**Fact F.2** ((Valiant & Valiant, 2011, Fact 29)). *Letting $\mathcal{N}(\mu, \sigma^2)$ denote the univariate Gaussian distribution,*

$$\mathrm{d}_{\mathrm{TV}}\big(\mathcal{N}(\mu, 1), \ \mathcal{N}(\mu + \alpha, 1)\big) \leq \frac{|\alpha|}{\sqrt{2\pi}}.$$

**Fact F.3** ((Valiant & Valiant, 2011, Fact 31)). *Let $G_1 = \mathcal{N}(\mu_1, \Sigma_1)$ and $G_2 = \mathcal{N}(\mu_2, \Sigma_2)$ in $\mathbb{R}^m$, and let $\Sigma_1 = TT^\top$ be the Cholesky decomposition of $\Sigma_1$. Let $\lambda_i$ denote the $i^{th}$ the eigenvalue of $T^{-1}\Sigma_2 T^{-\top}$, then*

$$d_{\mathrm{TV}}(G_1, G_2) \leq \frac{1}{\sqrt{2\pi e}} \sum_{i=1}^m \left( \max\{\lambda_i, \lambda_i^{-1}\} - 1 \right) + \frac{\left\| T^{-1}(\mu_1 - \mu_2) \right\|}{\sqrt{2\pi}}.$$

We use the following corollary of this fact:

**Corollary 24.** *Let $G_1 = \mathcal{N}(\mu, \Sigma_1)$ and $G_2 = \mathcal{N}(\mu, \Sigma_2)$ in $\mathbb{R}^m$, and suppose $\Sigma_1$ is diagonal with positive entries. Then, with $T = \Sigma_1^{1/2}$ (so $T^{-1}\Sigma_2 T^{-\top} = \Sigma_1^{-1/2}\Sigma_2\Sigma_1^{-1/2}$), we get*

$$d_{\mathrm{TV}}(G_1, G_2) \leq \frac{1}{\sqrt{2\pi e}} \sum_{i=1}^m \left( \max\{\lambda_i, \lambda_i^{-1}\} - 1 \right),$$

*where $\lambda_1, \ldots, \lambda_m$ are the eigenvalues of $\Sigma_1^{-1/2}\Sigma_2\Sigma_1^{-1/2}$.*

### F.2 Restating CLT from Valiant & Valiant (2011)

**Definition 25** (Generalized Multinomial). *Let $n \geq 1$ and $k \geq 1$. For each $r \in \{1, \ldots, n\}$ let $\rho_{r,1}, \ldots, \rho_{r,k} \in [0, 1]$ with $\sum_{i=1}^k \rho_{r,i} \leq 1$. Let $X^{(r)} \in \{0, e_1, \ldots, e_k\} \subset \mathbb{R}^k$ be independent with*

$$\Pr\big[X^{(r)} = e_i\big] = \rho_{r,i}, \qquad \Pr\big[X^{(r)} = 0\big] = 1 - \sum_{i=1}^k \rho_{r,i}.$$

*The sum $S := \sum_{r=1}^n X^{(r)} \in \mathbb{Z}_{\geq 0}^k$ is said to be distributed as a generalized multinomial.*

**Definition 26** (Discretized Gaussian). *For $\mu \in \mathbb{R}^k$ and positive semidefinite $\Sigma \in \mathbb{R}^{k \times k}$, let $\mathcal{N}(\mu, \Sigma)$ denote the $k$-dimensional Gaussian. $\mathcal{N}_{\mathrm{disc}}(\mu, \Sigma)$ denotes the distribution corresponding to obtaining a draw from $\mathcal{N}(\mu, \Sigma)$ and rounding each coordinate to the nearest integer.*

**Theorem 27** (Valiant & Valiant (2011, Thm. 4)). *Let $S$ be a $k$-dimensional Generalized Multinomial with parameters as above, mean $\mu$, covariance $\Sigma$, and $\sigma^2 = \lambda_{\min}(\Sigma)$. Then,*

$$d_{\mathrm{TV}}\big(S, \mathcal{N}_{\mathrm{disc}}(\mu, \Sigma)\big) \leq \frac{k^{4/3}}{\sigma^{1/3}} \cdot 2.2 \cdot \big(3.1 + 0.83 \log n\big)^{2/3}.$$

**Corollary 28.** *For constant $k$,*

$$d_{\mathrm{TV}}\big(S, \mathcal{N}_{\mathrm{disc}}(\mu, \Sigma)\big) = O\left( \frac{(\log n)^{2/3}}{\sigma^{1/3}} \right).$$

## G Closeness Testing

Our tester for testing closeness follows the proof of Batu et al. (2001). Specifically, we divide the analysis into two parts: heavy elements $B$ and light elements $S$. We show that we can estimate the distance $\sum_{i \in B} |\mathbf{p}_{\mathrm{avg}}(i) - \mathbf{q}_{\mathrm{avg}}(i)|$ up to accuracy $\epsilon$ using the provided samples. On light elements, since the norms corresponding to these elements is bounded, we can apply the $\ell_2$ tester to get the desired closeness tester. We divide the analysis into four sections. In the first section, we provide the $\ell_2$ tester, whose sample complexity depends on the norm of the underlying distributions. Therefore invoking $\ell_2$ tester only on light elements reduces our sample complexity as their corresponding norm is low. Further, in the second and third section we show several properties of light and heavy elements. In particular, for the heavy elements we show that the we can estimate the $\ell_1$ distance for these elements quite well. Furthermore, in the same section, we also show that the norm of the light elements is low, setting ourselves to use $\ell_2$ tester on these elements. Finally in the last section, we provide the final theorem, which invokes the results from sections one and two to get the desired bounds on the closeness testing.

### G.1 $\ell_2$ tester

To test closeness in $\ell_2$, we build an unbiased estimator for the terms $\|\mathbf{p}_{\text{avg}}\|_2^2$, $\|\mathbf{q}_{\text{avg}}\|_2^2$ and $\langle \mathbf{p}_{\text{avg}}, \mathbf{q}_{\text{avg}} \rangle$. In the following, we provide a description of these estimator. For each $t \in [T]$, let $(X_t(2), X_t(3))$ and $(X'_t(2), X'_t(3))$ denote the 2nd and 3rd i.i.d samples drawn from the distribution $\mathbf{p}_t$ and $\mathbf{q}_t$ respectively. For each $t \in [T]$, we define,

$$Y_t = \mathbb{1}[X_t(2) = X_t(3)] \text{ and } Y_{st} = \mathbb{1}[X_t(2) = X_s(3)] .$$

$$Y'_t = \mathbb{1}[X'_t(2) = X'_t(3)] \text{ and } Y'_{st} = \mathbb{1}[X'_t(2) = X'_s(3)] .$$

Note that $\mathbb{E}[Y_t] = \|\mathbf{p}_t\|_2^2$, $\mathbb{E}[Y'_t] = \|\mathbf{q}_t\|_2^2$ and $\mathbb{E}[Y_{st}] = \langle \mathbf{p}_t, \mathbf{p}_s \rangle$, $\mathbb{E}[Y'_{st}] = \langle \mathbf{q}_t, \mathbf{q}_s \rangle$ for all $s, t \in [T]$. Using these random variables, we define an estimator $Z$ and $Z'$ as follows,

$$Z = \frac{1}{T^2} \Big( \sum_{t \in [T]} Y_t + 2 \sum_{s < t} Y_{st} \Big).$$

$$Z' = \frac{1}{T^2} \Big( \sum_{t \in [T]} Y'_t + 2 \sum_{s < t} Y'_{st} \Big).$$

**Lemma 29.** *The estimator $Z$ and $Z'$ defined in Equation* (1) *satisfy the following,*

$$\mathbb{E}[Z] = \|\boldsymbol{p}_{\text{avg}}\|_2^2 \text{ and } \operatorname{Var}[Z] \leq \frac{4}{T^2} \Big\| \boldsymbol{p}_{\text{avg}} \Big\|_2^2 + \frac{48}{T} \Big\| \boldsymbol{p}_{\text{avg}} \Big\|_3^3.$$

$$\mathbb{E}[Z'] = \|\boldsymbol{q}_{\text{avg}}\|_2^2 \text{ and } \operatorname{Var}[Z'] \leq \frac{4}{T^2} \Big\| \boldsymbol{q}_{\text{avg}} \Big\|_2^2 + \frac{48}{T} \Big\| \boldsymbol{q}_{\text{avg}} \Big\|_3^3.$$

The proof of the above lemma follows immediately from Lemma 4.

Next, define

$$C_{st} = \mathbb{1}[X_s(2) = X'_t(2)].$$

$$Q = \frac{1}{T^2} \Big( \sum_{s,t \in [T]} C_{s,t} \Big).$$

**Lemma 30.** *The estimator $Q$ defined above satisfies the following,*

$$\mathbb{E}[Q] = \langle \boldsymbol{p}_{\text{avg}}, \boldsymbol{q}_{\text{avg}} \rangle.$$

$$\operatorname{Var}[Q] = \frac{1}{T^2} \langle \boldsymbol{p}_{\text{avg}}, \boldsymbol{q}_{\text{avg}} \rangle + \frac{1}{T} (\langle \boldsymbol{p}_{\text{avg}}, \boldsymbol{q}_{\text{avg}}, \boldsymbol{q}_{\text{avg}} \rangle + \langle \boldsymbol{p}_{\text{avg}}, \boldsymbol{p}_{\text{avg}}, \boldsymbol{q}_{\text{avg}} \rangle).$$

*Proof.* The expectation of the estimator $Q$ is as follows,

$$\mathbb{E}[Q] = \frac{1}{T^2} \sum_{s,t \in [T]} \langle \mathbf{p}_s, \mathbf{q}_t \rangle = \Big\langle \frac{1}{T} \sum_{s \in [T]} \mathbf{p}_s, \frac{1}{T} \sum_{s \in [T]} \mathbf{q}_t \Big\rangle = \langle \mathbf{p}_{\text{avg}}, \mathbf{q}_{\text{avg}} \rangle.$$

In the following we bound the variance, define $\hat{C}_{st} = C_{st} - \mathbb{E}[C_{st}]$.

$$\mathbb{E}\Big[ \Big( \sum_{s,t \in [T]} \hat{C}_{st} \Big)^2 \Big]$$

$$= \mathbb{E}\Big[\Big(\sum_{s,t} C_{st}\Big)^2\Big] - \Big(\mathbb{E}\Big[\sum_{s,t} C_{st}\Big]\Big)^2$$

$$= \underbrace{\sum_{s,t} \mathbb{E}\left[C_{st}^2\right]}_{T^2 \text{ terms}} + \underbrace{\sum_{\substack{s,t \\ a,b \\ |\{s,t,a,b\}|=3}} \mathbb{E}[C_{st}C_{ab}]}_{O(\binom{T}{3}) \text{ terms}} + \underbrace{\sum_{\substack{s,t \\ a,b \\ |\{s,t,a,b\}|=4}} \mathbb{E}[C_{st}C_{ab}]}_{O(\binom{T}{4}) \text{ terms}} - \underbrace{\Big(\mathbb{E}\Big[\sum_{s,t} C_{st}\Big]\Big)^2}_{\binom{T}{2}^2 \text{ terms}}$$

$$= \sum_{s,t} \Big(\mathbb{E}\left[C_{st}^2\right] - \mathbb{E}\left[C_{st}\right]^2\Big) + \sum_{\substack{s,t \\ a,b \\ |\{s,t,a,b\}|=3}} \big(\mathbb{E}[C_{st}C_{ab}] - \mathbb{E}[C_{st}]\mathbb{E}[C_{ab}]\big) + \sum_{\substack{s,t \\ a,b \\ |\{s,t,a,b\}|=4}} \big(\mathbb{E}[C_{st}C_{ab}] - \mathbb{E}[C_{st}]\mathbb{E}[C_{ab}]\big)$$

$$\leq \sum_{s,t} \big(\mathbb{E}\left[C_{st}^2\right]\big) + \sum_{\substack{s,t \\ a,b \\ |\{s,t,a,b\}|=3}} \big(\mathbb{E}[C_{st}C_{ab}] - \mathbb{E}[C_{st}]\mathbb{E}[C_{ab}]\big) + \sum_{\substack{s,t \\ a,b \\ |\{s,t,a,b\}|=4}} \big(\mathbb{E}[C_{st}C_{ab}] - \mathbb{E}[C_{st}]\mathbb{E}[C_{ab}]\big)$$

$$= \sum_{s,t} \big(\mathbb{E}\left[C_{st}^2\right]\big) + \sum_{\substack{s,c \\ a,c \\ s\neq a}} \big(\mathbb{E}[C_{sc}C_{ac}] - \mathbb{E}[C_{sc}]\mathbb{E}[C_{ac}]\big) + \sum_{\substack{a,t \\ a,b \\ t\neq b}} \big(\mathbb{E}[C_{at}C_{ab}] - \mathbb{E}[C_{at}]\mathbb{E}[C_{ab}]\big).$$

In the last step, we used the fact that $\mathbb{E}[C_{st}C_{ab}] - \mathbb{E}[C_{st}]\mathbb{E}[C_{ab}] = 0$ when $s \neq a$ and $t \neq b$. Further, since $\mathbb{E}[C_{st}] \geq 0$ for all $s, t$, we get

$$\mathbb{E}\Big[\Big(\sum_{s,t\in[T]} \hat{C}_{st}\Big)^2\Big] \leq \sum_{s,t} \big(\mathbb{E}\left[C_{st}^2\right]\big) + \sum_{\substack{s,c \\ a,c \\ s\neq a}} \big(\mathbb{E}[C_{sc}C_{ac}]\big) + \sum_{\substack{a,t \\ a,b \\ t\neq b}} \big(\mathbb{E}[C_{at}C_{ab}]\big) \tag{22}$$

Now, we look at individual terms in the above expression. The last two terms can be bounded as follows:

$$\sum_{\substack{s,c \\ a,c \\ |s\neq a}} \mathbb{E}[C_{sc}C_{ac}] = \sum_{s\neq a} \sum_c \sum_{\ell=1}^k \mathbf{p}_s(\ell)\mathbf{p}_a(\ell)\mathbf{q}_c(\ell) := \sum_{s\neq a} \sum_c \langle \mathbf{p}_s, \mathbf{p}_a, \mathbf{q}_c \rangle, \tag{23}$$

$$\leq T^3 \cdot \langle \mathbf{p}_{\mathrm{avg}}, \mathbf{p}_{\mathrm{avg}}, \mathbf{q}_{\mathrm{avg}} \rangle \tag{24}$$

$$\sum_{\substack{a,t \\ a,b \\ |t\neq b}} \mathbb{E}[C_{at}C_{ab}] = \sum_{t\neq b} \sum_a \sum_{\ell=1}^k \mathbf{p}_a(\ell)\mathbf{q}_t(\ell)\mathbf{q}_b(\ell) := \sum_{t\neq b} \sum_a \langle \mathbf{p}_a, \mathbf{q}_t, \mathbf{q}_b \rangle, \tag{25}$$

$$\leq T^3 \cdot \langle \mathbf{p}_{\mathrm{avg}}, \mathbf{q}_{\mathrm{avg}}, \mathbf{q}_{\mathrm{avg}} \rangle \tag{26}$$

The first term can be bounded as follows:

$$\sum_{s,t} \big(\mathbb{E}\left[C_{st}^2\right]\big) = T^2 \langle \mathbf{p}_{\mathrm{avg}}, \mathbf{q}_{\mathrm{avg}} \rangle \tag{27}$$

Substituting these in Equation 22, and using $\mathrm{Var}[Q] = \frac{1}{T^4}\mathbb{E}\Big[\Big(\sum_{s,t\in[T]} \hat{C}_{st}\Big)^2\Big]$, we get the desired variance bound. $\qquad\square$

**Lemma 31.** *Let $F = Z + Z' - 2Q$, then note that,*

$$\mathbb{E}[F] = \|\boldsymbol{p}_{\mathrm{avg}} - \boldsymbol{q}_{\mathrm{avg}}\|_2^2 .$$

$$\mathrm{Var}[F] \leq \frac{O(1)}{T^2}(\|\boldsymbol{q}_{\mathrm{avg}}\|_2^2 + \boldsymbol{p}_{\mathrm{avg}}\|_2^2 + \langle \boldsymbol{p}_{\mathrm{avg}}, \boldsymbol{q}_{\mathrm{avg}} \rangle) + \frac{O(1)}{T}(\langle \boldsymbol{p}_{\mathrm{avg}}, \boldsymbol{q}_{\mathrm{avg}}, \boldsymbol{q}_{\mathrm{avg}} \rangle + \langle \boldsymbol{p}_{\mathrm{avg}}, \boldsymbol{p}_{\mathrm{avg}}, \boldsymbol{q}_{\mathrm{avg}} \rangle) .$$

*Proof.* The expression for $E[F]$ follows immediately by combining bias terms in Lemma 29 and Lemma 30.

The variance bound follows by using the fact that $\text{Var}[F] \leq 8(\text{Var}(Z) + \text{Var}(Z') + \text{Var}(Q))$ and further using variance bounds on $Z, Z', Q$ from Lemma 29 and Lemma 30. $\square$

**Lemma 32.** *We can test if* $\|\boldsymbol{p}_{\text{avg}} - \boldsymbol{q}_{\text{avg}}\|_2 \leq \epsilon/3$ *vs* $\|\boldsymbol{p}_{\text{avg}} - \boldsymbol{q}_{\text{avg}}\|_2 > \epsilon$ *using* $T = O(\frac{\sqrt{\|\boldsymbol{q}_{\text{avg}}\|_2^2 + \boldsymbol{p}_{\text{avg}}\|_2^2 + \langle \boldsymbol{p}_{\text{avg}}, \boldsymbol{q}_{\text{avg}} \rangle}}{\epsilon^2}) + \frac{(\langle \boldsymbol{p}_{\text{avg}}, \boldsymbol{q}_{\text{avg}}, \boldsymbol{q}_{\text{avg}} \rangle + \langle \boldsymbol{p}_{\text{avg}}, \boldsymbol{p}_{\text{avg}}, \boldsymbol{q}_{\text{avg}} \rangle)}{\epsilon^4}$ *samples.*

*Proof.* It is sufficient to bound the probability of $P(|F - \mathbb{E}[F]| > \epsilon/3)$. By Chebyshev's inequality, we get that $P(|F - \mathbb{E}[F]| > \epsilon/3) \leq 9\,\text{Var}(F)/\epsilon^2$. Substituting the upper bound on the variance of $F$, we get that,

$$P(|F - \mathbb{E}[F]| > \epsilon^2/3) \leq \frac{O(1)}{T^2 \epsilon^4}(\|\mathbf{q}_{\text{avg}}\|_2^2 + \mathbf{p}_{\text{avg}}\|_2^2 + \langle \mathbf{p}_{\text{avg}}, \mathbf{q}_{\text{avg}} \rangle)$$
$$+ \frac{O(1)}{T \epsilon^4}(\langle \mathbf{p}_{\text{avg}}, \mathbf{q}_{\text{avg}}, \mathbf{q}_{\text{avg}} \rangle + \langle \mathbf{p}_{\text{avg}}, \mathbf{p}_{\text{avg}}, \mathbf{q}_{\text{avg}} \rangle)$$

The above inequality is further upper bounded by a small constant for the value of $T$ specified in the conditions of the lemma. $\square$

### G.2 Heavy elements

Let $X_t(1)$ and $X_t'(1)$ be the 1st i.i.d. sample from distribution $\mathbf{p}_t$ and $\mathbf{q}_t$ respectively. Let $\widehat{p}(i) = \frac{1}{T} \sum_{t \in [T]} \mathbb{1}[X_t(1) = i]$ and $\widehat{q}(i) = \frac{1}{T} \sum_{t \in [T]} \mathbb{1}[X_t'(1) = i]$. We further define big elements and small elements as follows,

$$B_p = \{i \in [k] \mid \widehat{p}(i) \geq b\} \text{ and } S_p = \{i \in [k] \mid \widehat{p}(i) < b\} \ ,$$
$$B_q = \{i \in [k] \mid \widehat{q}(i) \geq b\} \text{ and } S_q = \{i \in [k] \mid \widehat{q}(i) < b\} \ ,$$

where $b = (\epsilon/k)^{2/3}$.

**Lemma 33.** *Let* $B_p$ *and* $B_q$ *be random sets as defined above and let* $B = B_p \cup B_q$. *If* $n = O(\frac{1}{b\epsilon^2})$, *then the following conditions holds,*

$$P(\sum_{i \in B} |\widehat{p}(i) - \boldsymbol{p}_{\text{avg}}(i)| > \epsilon/12) \leq 1/100.$$

*and*

$$P(\sum_{i \in B} |\widehat{q}(i) - \boldsymbol{q}_{\text{avg}}(i)| > \epsilon/12) \leq 1/100.$$

*Proof.* Note that $B$ is a random set that includes at most $2/b$ elements. Given any fixed subset $A \subseteq [n]$, note that,

$$P(\sum_{i \in A} |\widehat{p}(i) - \mathbf{p}_{\text{avg}}(i)| > \epsilon/12) \leq P(\exists\ A' \subseteq A \text{ such that } \widehat{p}(A') - \mathbf{p}_{\text{avg}}(A') > \epsilon/12)$$

$$\leq 2^{|A|} \exp(-O(n\epsilon^2)) \ .$$

Recall in the above expression $\widehat{p}(A') = \sum_{i \in A'} \widehat{p}(i)$ and $\mathbf{p}_{\text{avg}}(A') = \sum_{i \in A'} \mathbf{p}_{\text{avg}}(i)$.

In the last inequality, we used the fact that, $P(\widehat{p}(A') - \mathbf{p}_{\text{avg}}(A') > \epsilon/12) \leq \exp(-n\epsilon^2/144)$ and did a union bound over all subsets of $A$. Therefore for any fixed $A$ such that, $|A| \leq 2/b$, we have that,

$$P(|\widehat{p}(A) - \mathbf{p}_{\text{avg}}(A)| > \epsilon/12) \leq \frac{1}{100} \ .$$

$$P(|\widehat{p}(B) - \mathbf{p}_{\text{avg}}(B)| > \epsilon/12) = \sum_{A \subseteq [n] || A| \leq 2/b} P(|\widehat{p}(B) - \mathbf{p}_{\text{avg}}(B)| > \epsilon/12 | B = A) P(B = A) \quad (28)$$

$$\leq \sum_{A \subseteq [n] || A| \leq 2/b} P(|\widehat{p}(A) - \mathbf{p}_{\text{avg}}(A)| > \epsilon/12) P(B = A) \quad (29)$$

$$\leq \sum_{A \subseteq [n] | |A| \leq 2/b} \frac{1}{100} P(B = A) \leq 1/100. \tag{30}$$

The analogous proof also holds for $\mathbf{q}_{\text{avg}}$ and we conclude the proof. □

**Lemma 34.** *Let $B_p$ and $B_q$ be random sets as defined above and let $B = B_p \cup B_q$. If $n = O(\frac{1}{b\epsilon^2})$, then the following conditions holds,*

$$P(|\sum_{i \in B} |\hat{p}(i) - \hat{q}(i)| - \sum_{i \in B} |\boldsymbol{p}_{\text{avg}}(i) - \boldsymbol{q}_{\text{avg}}(i)|| \geq \epsilon/6) \leq 1/50 .$$

*Proof.* Let $E = \sum_{i \in B} |\hat{p}(i) - \hat{q}(i)|$, $F = \sum_{i \in B} |\mathbf{p}_{\text{avg}}(i) - \mathbf{q}_{\text{avg}}(i)|$, $G = \sum_{i \in B} |\hat{p}(i) - \mathbf{p}_{\text{avg}}(i)|$ and $H = \sum_{i \in B} |\hat{q}(i) - \mathbf{q}_{\text{avg}}(i)|$. Note that, $E \leq F + G + H$ and $F \leq E + G + H$. Combining these two inequalities we get that,

$$|E - F| \leq G + H. \tag{31}$$

Using the above inequality we get that,

$$P(|E - F| \geq \epsilon/6) \leq P(G + H \geq \epsilon/6) \leq P(G \geq \epsilon/12) + P(H \geq \epsilon/12) \leq 1/50. \tag{32}$$

The last expression follows from Lemma 33. □

### G.3 Light elements

**Lemma 35** (Light elements)**.** *Let $S$ denote the complement of $B$, corresponding to the random set containing elements with empirical probability less than $b$ in both $\hat{p}$ and $\hat{q}$. The following norm bounds hold:*

$$P(\sum_{i \in S} \boldsymbol{p}_{\text{avg}}(i)^2 > 4b/\delta) \leq \delta \text{ and } P(\sum_{i \in S} \boldsymbol{p}_{\text{avg}}(i)^3 > 8b^2/\delta) \leq \delta ,$$

$$P(\sum_{i \in S} \boldsymbol{q}_{\text{avg}}(i)^2 > 4b/\delta) \leq \delta \text{ and } P(\sum_{i \in S} \boldsymbol{q}_{\text{avg}}(i)^3 > 8b^2/\delta) \leq \delta ,$$

$$P(\sum_{i \in S} \boldsymbol{p}_{\text{avg}}(i)\boldsymbol{q}_{\text{avg}}(i) > 4b/\delta) \leq \delta, P(\sum_{i \in S} \boldsymbol{p}_{\text{avg}}^2(i)\boldsymbol{q}_{\text{avg}}(i) > 8b^2/\delta) \leq \delta \text{ and } P(\sum_{i \in S} \boldsymbol{p}_{\text{avg}}(i)\boldsymbol{q}_{\text{avg}}^2(i) > 8b^2/\delta) \leq \delta .$$

*Proof.* This argument is essentially identical to the proof of Theorem 20 in Chan et al. (2014).

Let $H = \{i \in [k] | \mathbf{p}_{\text{avg}}(i) > 2b\}$ and recall that $S = \{i \in [k] | \hat{p}(i) < b \text{ and } \hat{q}(i) < b\}$. Here we provide the proof for $P(\sum_{i \in S} \mathbf{p}_{\text{avg}}(i)^t > 2^t b/\delta) \leq \delta$ for $t \in \{2, 3\}$ and the proof for all the other probability statements follow the same argument.

To prove $P(\sum_{i \in S} \mathbf{p}_{\text{avg}}(i)^t > 2^t b^{t-1}/\delta) \leq \delta$ for $t \in \{2, 3\}$ we first show that,

$$E[\sum_{i \in S} \mathbf{p}_{\text{avg}}(i)^t] \leq 2^t b^{t-1},$$

and the lemma statements follow immediately by applying Markov Inequality. Therefore, in the remainder of the proof, we focus on proving the upper bound on the expectation.

Note that, $E[\sum_{i \in S} \mathbf{p}_{\text{avg}}(i)^t] = E[\sum_{i \in S \cap H} \mathbf{p}_{\text{avg}}(i)^t + \sum_{i \in S \cap H^c} \mathbf{p}_{\text{avg}}(i)^t] \leq 2^{t-1} b^{t-1} + \sum_{i \in S \cap H} \mathbf{p}_{\text{avg}}(i)^t$. In the last inequality, we used $\mathbf{p}_{\text{avg}}(i) < 2b$ for all $i \in H^c$. Therefore all that remains is to show that, $E[\sum_{i \in S \cap H} \mathbf{p}_{\text{avg}}(i)^t] \leq 2^{t-1} b^{t-1}$.

For $i \in S \cap H$, let $\mathbf{p}_{\text{avg}}(i) = x_i b$ and note that $x_i \geq 2$. Then we have that,

$$E[\sum_{i \in S \cap H} \mathbf{p}_{\text{avg}}(i)^t] = \sum_{i \in H} \mathbf{p}_{\text{avg}}(i)^t P(i \in S),$$

$$= \sum_{i \in H} \mathbf{p}_{\text{avg}}(i) b^{t-1} x_i^{t-1} P(\mathbf{p}_{\text{avg}}(i) = x_i b \text{ and } \hat{p}(i) < b).$$

In the last inequality, we used $\mathbf{p}_{\text{avg}}(i) = x_i b$ and $P(i \in S) \le P(\mathbf{p}_{\text{avg}}(i) = x_i b \text{ and } \hat{p}(i) < b)$.

Note that, by Chernoff bound it is immediate that, $P(i \in S) \le \exp(-Cbn) \le \exp(-Cx_i/\epsilon^2)$, for some constant $C$ just dependent on $t$. Substituting this probability upper bound back in the expectation we get that,

$$E[\sum_{i \in S \cap H} \mathbf{p}_{\text{avg}}(i)^t] \le cb^{t-1},$$

for some constant $c$ dependent on $t$ and we conclude the proof.

$\square$

### G.4 $\ell_1$ tester

Here we finally present our $\ell_1$ tester.

**Theorem 36.** *For any $\epsilon > k^{-1/3}$, there exists an algorithm $\ell_1$-Distance-Test that, for $O(\frac{k^{2/3}}{\epsilon^{8/3}})$ samples from $\boldsymbol{p}_1, \ldots, \boldsymbol{p}_T$ and $\boldsymbol{q}_1, \ldots, \boldsymbol{q}_T$ has the following behavior: it rejects with probability $2/3$ when $\|\boldsymbol{p}_{\text{avg}} - \boldsymbol{q}_{\text{avg}}\|_1 \ge \epsilon$, and accepts with probability $2/3$ when $p = q$.*

*Proof.* Let $b = (\epsilon/k)^{2/3}$. Note that $n = O(1/(b\epsilon^2))$. Our $\ell_1$ tester works as follows. 1) It uses the first set of samples from $\mathbf{p}_1 \ldots \mathbf{p}_T$ and $\mathbf{q}_1 \ldots \mathbf{q}_T$ to estimate the heavy elements, which are defined as follows.

$$B_p = \{i \in [k] \mid \hat{p}(i) \ge b\} \text{ and } B_q = \{i \in [k] \mid \hat{q}(i) \ge b\} .$$

Define $B = B_p \cup B_q$. By Lemma 34, we know that with a good probability, $|\sum_{i \in B} |\hat{p}(i) - \hat{q}(i)|$ estimates the quantity $|\sum_{i \in B} |\mathbf{p}_{\text{avg}}(i) - \mathbf{q}_{\text{avg}}(i)|$ upto $\epsilon/6$ accuracy. Therefore, if $\|\mathbf{p}_{\text{avg}} - \mathbf{q}_{\text{avg}}\|_1 \ge \epsilon/3$, then this implies that $|\sum_{i \in B} |\hat{p}(i) - \hat{q}(i)| > \epsilon/6$. In this particular case, we know that we are in the case of $\|\mathbf{p}_{\text{avg}} - \mathbf{q}_{\text{avg}}\|_1 \ge \epsilon$ and we reject the instance. Therefore in the remainder of the proof we focus on the case, where $|\sum_{i \in B} |\mathbf{p}_{\text{avg}}(i) - \mathbf{q}_{\text{avg}}(i)| \le \epsilon/3$.

Here we define new distributions $\mathbf{p}'_1 \ldots \mathbf{p}'_T$ and $\mathbf{q}'_1 \ldots \mathbf{q}'_T$ focused on light elements as follows: Sample an element from $\mathbf{p}_t$. If this sample is in $S$ output it; otherwise, output uniformly random element from $[k]$. Define $\mathbf{q}'_t$ similarly. We generate two samples from $\mathbf{p}'_1 \ldots \mathbf{p}'_T$ and $\mathbf{q}'_1 \ldots \mathbf{q}'_T$ using this procedure. Let $\mathbf{p}'_{\text{avg}}$ and $\mathbf{q}'_{\text{avg}}$ be respective averages.

Note that, for $i \in S$, $\mathbf{p}'_{\text{avg}}(i) = \mathbf{p}_{\text{avg}}(i) + \mathbf{p}_{\text{avg}}(B)/k$ and $\mathbf{q}'_{\text{avg}}(i) = \mathbf{q}_{\text{avg}}(i) + \mathbf{q}_{\text{avg}}(B)/k$. For $i \in B$, we have that, $\mathbf{p}'_{\text{avg}}(i) = \mathbf{q}_{\text{avg}}(B)/k$ and $\mathbf{q}'_{\text{avg}}(i) = \mathbf{q}_{\text{avg}}(B)/k$.

Note that, when $\mathbf{p}_{\text{avg}} = \mathbf{q}_{\text{avg}}$, we have that, $\mathbf{p}'_{\text{avg}} = \mathbf{q}'_{\text{avg}}$. Furthermore in the other case, when $\|\mathbf{p}_{\text{avg}} - \mathbf{q}_{\text{avg}}\|_1 \ge \epsilon$ and $\sum_{i \in B} |\mathbf{p}_{\text{avg}}(i) - \mathbf{q}_{\text{avg}}(i)| \le \epsilon/3$, we get that, $\|\mathbf{p}'_{\text{avg}} - \mathbf{q}'_{\text{avg}}\|_1 \ge \epsilon/3$.

Note that by Lemma 35, we get that, the 2nd and 3rd order norms for $\mathbf{p}'_{\text{avg}}$ and $\mathbf{q}'_{\text{avg}}$ are bounded by $O(b)$ and $O(b^2)$ respectively with very tiny constant probability. Therefore we get that, $\|\mathbf{p}'_{\text{avg}}\|_2^2 \le \sum_{i \in S} \mathbf{p}_{\text{avg}}(i)^2 + O(1/k) \le O(b + 1/k)$, $\|\mathbf{q}'_{\text{avg}}\|_2^2 \le \sum_{i \in S} \mathbf{q}_{\text{avg}}(i)^2 + O(1/k) \le O(b + 1/k)$ and $\langle \mathbf{p}_{\text{avg}}, \mathbf{q}_{\text{avg}} \rangle \le O(b + 1/k)$. Similarly, the third order norms are also bounded, $\|\mathbf{p}'_{\text{avg}}\|_3^3 \le \sum_{i \in S} \mathbf{p}_{\text{avg}}(i)^3 + 3/k * \sum_{i \in S} \mathbf{p}_{\text{avg}}(i)^2 + O(1/k^2) \le O(b^2 + b/k + 1/k^2)$ and $\|\mathbf{q}'_{\text{avg}}\|_3^3 \le \sum_{i \in S} \mathbf{q}_{\text{avg}}(i)^3 + +3/k * \sum_{i \in S} \mathbf{q}_{\text{avg}}(i)^2 + O(1/k) \le O(b^2 + b/k + 1/k^2)$, $\langle \mathbf{p}_{\text{avg}}, \mathbf{q}_{\text{avg}}, \mathbf{q}_{\text{avg}} \rangle \le O(b^2 + b/k + 1/k^2)$ and $\langle \mathbf{p}_{\text{avg}}, \mathbf{p}_{\text{avg}}, \mathbf{q}_{\text{avg}} \rangle \le O(b^2 + b/k + 1/k^2)$. We then invoke the $\ell_2$ tester with $\epsilon' = \epsilon/10\sqrt{k}$. Further using the sample complexity bounds from Lemma 32, we get the following upper bound,

$$O(\sqrt{b + 1/k}/\epsilon'^2 + (b^2 + b/k + 1/k^2)/\epsilon'^4) \in O(k^{2/3}/\epsilon^{8/3}) ,$$

which is the required sample complexity.

$\square$

