# OpenReview forum: "Testing with Non-identically Distributed Samples"
_TMLR — Accepted by TMLR_

### Review · Reviewer_incz · 2025-07-31

**Summary Of Contributions:**

The paper initiates a systematic study of learning and testing the average $p_{\text{avg}}$ of $T$ distributions $p_1, \dots, p_T$, each supported over a discrete set $[k]$, when given access to samples from the individual distributions. This is a generalization of the standard distribution learning/testing framework where $T = 1$. The paper presents several interesting results in this setting. The questions answered in the paper have the following general flavor: When one receives exactly $c$ samples from each distribution, what are the lower and upper bounds on the number $T$ of distributions whose average can be learned/tested? The results in the paper affirms the intuition that when $c$ is smaller, $T$ needs to be larger.

The specific results are the following:

(1) If $c = 1$, then $T = \Theta(k/\epsilon^2)$ for the problem of learning $p_{\text{avg}}$ up to a total variation (TV) distance of $\epsilon$.

(2) If $c = 1$, then $T = \Omega(k)$ for testing properties of $p_{\text{avg}}$.

(3) If $c = 2$, then $T = O(\sqrt{k})$ for uniformity (and identity) testing of $p_{\text{avg}}$ for constant $\epsilon$.

(4) If $c = 3$, then for $T = O(k^{2/3})$, one can test closeness of $p_{\text{avg}}$ and $q_{\text{avg}}$ for two sequences $p_1, \dots, p_T$ and $q_1, \dots, q_T$ of distributions supported over $[k]$.

(5) If one does not have the information regarding which sample is drawn from which distribution, then testing properties of $p_{\text{avg}}$, in general, requires $T = \Omega(k)$ even for $c = 2$.

**Audience:**

Yes

**Broader Impact Concerns:**

None.

**Claims And Evidence:**

Yes

**Requested Changes:**

I think that the paper is quite well-written and have no specific comments here.

**Strengths And Weaknesses:**

The main strength of the paper is indeed the new model regarding testing properties of averages of distributions. The authors have motivated it well and it is also a natural generalization of the standard distribution testing. Many of the techniques used are standard. Personally, I think that the lower bound regarding the so-called "pooled estimators", i.e., testing in the absence of information about what sample was drawn from what distribution, is quite interesting.

The following is not really a weakness of the paper. I am curious if the sample complexity of uniformity testing or related problems could be improved if one were to increase $c$. In general, it is worth exploring tradeoffs between $c$ and $T$ for general values of these parameters for various problems.

---

> ### Author Response · Authors · 2025-08-14
>
> We thank the reviewer for their time spent in reading our paper, and their positive comments.
>
> **Regarding tradeoffs between $c$ and $T$**:
>
> This is a great question, ripe for future inquiry! In particular, a generic lower bound of $cT=\Omega(\sqrt{k})$ follows from the lower bound in the standard i.i.d case where all the $p_i$’s are identical, for any constant $c$. For $c=1$, we are indeed able to show (Claim 1.1) that $T=\Omega(k)$ may sometimes be required, and for $c=2$, we are able to attain $T=O(\sqrt{k})$. On the other extreme, when $c$ is taken to be as large as $\Omega(\sqrt{k})$,  $T=1$ suffices. It would be very interesting to chart out the landscape for intermediate values of $c$ and $T$!

---

### Review · Reviewer_iUE2 · 2025-08-03

**Summary Of Contributions:**

The authors initiate the study of uniformity testing (and its variants) over a finite universe [k] with respect to independent but *not identically* distributions $p_1,\ldots,p_T$. Roughly speaking, the authors show that when only 1 sample is available from each distribution, sublinear property testing is impossible, and testing requires $\Omega(k)$ samples. On the other hand, as soon as one as at least 2 samples from each $p_i$, it is possible to test in only $O(\sqrt{k})$ samples, recovering (up to error dependence) classical results from uniformity testing.

In slightly more detail, the authors are interested in testing whether the *average distribution* $p_{avg}= \sum\limits_{I \in T} p_i$ is uniform or $\varepsilon$-far from uniform when a fixed number $c$ samples are drawn from each distribution and we think of $T$ (number of distributions) and $k$ (universe size) as growing.

The authors prove three main results in this setting: first, any uniformity testing algorithm with only $c=1$ samples per distribution requires $T \geq \Omega(k)$. Second, there exists an algorithm using only $c=2$ samples per distribution in the sublinear regime, namely whenever $T \geq \frac{\sqrt{k}}{\varepsilon^2}+\frac{1}{\varepsilon^4}$. Finally, any sublinear uniformity testing algorithm must use information about the "position" of the samples, i.e. which p_i they came from. In other words, any algorithm which only uses the `fingerprint’ (set of overall counts of each element) and not their positions requires $T \geq \Omega(k)$.

The authors also adapt the standard reduction from identity testing to uniformity testing to show a similar upper bound for the setting of testing whether $p_{avg}$ is close to some fixed distribution $q$ (rather than uniform). Finally, they prove a variant of their upper bound for *closeness testing*, where one is given $c=3$ samples from distributions $p_1,\ldots,p_T$ and $p’_1,\ldots,p’_T$ and wishes to test if their averages are $\varepsilon$-far. In this case they give an algorithm in the sublinear regime whenever $T \geq k^{2/3}/\varepsilon^{8/3}$

At a technical level, the authors adapt the classical method of using collision statistics to measure distance from uniformity. Namely they introduce a basic unbiased estimator for $p_{avg}$ using collisions within each $p_i$ and between $(p_i,p_j)$ pairs and prove the variance is low to show the estimator concentrates sufficiently in sublinear samples. Roughly speaking, this also explains why $c=1$ samples per distribution is insufficient (cannot measure intradistribution collisions), while $c=2$ is sufficient for sublinear testing. A similar argument is used in the closeness testing result, but the authors require an extra sample ($c=3$) in order to separately handle with `heavy’ elements with a standard estimator.

Most of the technical work in the paper actually lies in proving the lower bound against algorithms that just use count statistics. Here the authors construct two sequences of distributions which are far in total variation, but whose count statistics over $c=2$ samples from each distribution are close in total variation. The construction itself is elementary (simply distributing weight among a few different partitions of the universe), but showing the total variation bound requires some work relating the count statistics to discretized gaussians (then comparing the discretized gaussians themselves).

**Audience:**

Yes

**Claims And Evidence:**

Yes

**Requested Changes:**

**Revision requests:**

There are several places throughout the work where inconsistent notation is used and/or not defined. For instance, p_a^l is used e.g. in Eq (5) and not defined (this is already bad notation since it could be a power). You later use similar notation to mean something else to denote different distributions, then later in Eq (30), you use p_a(ell) instead to mean the same thing. Please go over your work more carefully and unify the notation used in different sections.


**Minor comments:**

When you use a period and it is not the end of a sentence, like in i.i.d., you need to “escape” the period in latex by writing ./ . Otherwise you get a double space as is seen in your pdf.

In Example 1, it’s not really clear what privacy has to do with collecting few samples per user. In fact to ensure privacy you need to do the opposite — collect many samples then add noise.

Typos: “are are that”, “smaller than greater than”, “ours map”, “uniformity random element”, “and and whose”

Pg 8: one place bold u_k is used, then just u? Similar, sometimes p(i) is used and sometimes p^I?

Below Eq (12), dropped subscript on $\Phi$

**Strengths And Weaknesses:**

Uniformity/Identity/Closeness testing are classical problems in statistical learning. Early work in the area assumed samples are drawn i.i.d from an unknown distribution $p$. Many recent works (both in testing and learning) have looked into relaxing this assumption, typically allowing some sort of light dependence between samples. The authors of this work introduce a new natural model relaxing the identically distributed assumption. They give reasonable motivation for this setting (e.g. testing distributions across spatially separated populations, or chronologically separated…) and essentially characterize when sublinear testing is possible in the model, up to the dependence on the testing error $\varepsilon$ where they leave the correct polynomial dependence open. Broadly speaking, the paper is also well written and fairly easy to follow (with a few exceptions noted in the requested changes below).

As the authors say, the main contribution of the work is really the introduction and basic analysis of a new model. At a technical level, most of the results are a straightforward adaptation of the standard collision based methods in the i.i.d uniformity testing setting. The lower bound against algorithms using count statistics does require some new insight and technical work, but the ideas therein are still fairly standard. In and of itself this is not a big weakness (since I do think the model is interesting and worth studying), but its disappointing the authors only study these basic adaptations *and* fail to achieve the right dependence in their bounds, e.g. the dependence 1/eps^4 vs 1/eps^2 (also missing epsilon dependence in some of the lower bounds). The work would be much stronger if the authors were able to give some new ideas beyond those used in the standard i.i.d setting to resolve the barriers they hit in the error dependence using standard techniques. The work could also be made stronger if the authors studied properties beyond testing $p_{avg}$ (which, while natural, does seem like a pretty limited setting), and seems like it may require more new ideas beyond standard techniques from the i.i.d setting.

Overall, the work introduces a nice model relaxing the classical i.i.d assumption and makes progress explaining when sublinear testing is possible largely using known techniques. I think the model is interesting and may inspire followup research, but the work itself would feel `more complete’ if they achieved tighter bounds or introduced more substantially new techniques to handle their introduced relaxation.

---

> ### Author Response · Authors · 2025-08-14
>
> We thank the reviewer for their time spent in reading our paper, and their positive comments. We will make sure to do the suggested cosmetic changes, and make notation consistent where it is not.
>
> As the reviewer pointed out, indeed our main contribution lies in the problem formulation and the conceptual takeaways. For completeness, we briefly reiterate the motivation and context for our work. Heterogeneity naturally arises in data collection, especially when data is gathered across time, geography, or users, as discussed in the paper. Existing sublinear testers are often applied without accounting for such variation. This raises a natural question: how robust are these testers to heterogeneity? Our main contribution is in concretely formulating this question with regards to testing the average distribution— an arguably natural and well-motivated starting point. We are able to establish the following conceptual takeaway: while existing testers are not directly applicable, they can be effectively adapted---provided one collects at least two samples per distribution and leverages source info. We also show the inapplicability of existing testers (which do not use source information) by proving a lower bound for pooled samples. Establishing this lower bound involves an independently interesting new construction, together with several non-trivial technical arguments. Thus, the lower bound can be seen as a technically novel contribution.
>
> It is true that the analysis of our proposed testers (i.e., the upper bounds) builds on existing techniques and doesn’t need fundamentally new ideas. However, technical novelty may not be the best metric to evaluate these bounds. They are valuable because they convey the main takeaway: that existing testers can be adapted to heterogeneous settings by effectively leveraging source information. In this sense, the simplicity of the testers and their analysis can be viewed as a strength---making them easier to apply and generalize.
>
> **Regarding tightness of the bounds $1/\varepsilon^4$ vs $1/\varepsilon^2$**:
>
> It is quite interesting to us that despite our numerous attempts to get the optimal dependence, the analysis of the collision-based tester does not yield an optimal bound in the non-i.i.d setting. Furthermore, it is also not at all clear how other testers from the standard i.i.d. setting may be adapted to the non-i.i.id setting. We describe the technical issue in detail in Remark 6, and the issue might even suggest something fundamental. For example, it may be the case that there is a $1/\varepsilon^4$ lower bound after all for the collision-based tester in the non-i.i.d setting! This is indeed a very concrete open question for future work.

---

### Review · Reviewer_7HxZ · 2025-08-05

**Summary Of Contributions:**

This paper addresses the problem of \emph{sublinear-sample distribution property testing} under a relaxation of the usual i.i.d. assumption. Specifically, the authors consider $T$ distinct discrete distributions $p_1,\dots,p_T$ (each supported on at most $k$ elements) and assume we can draw $c$ independent samples from each $p_i$. The goal is to test properties of the  average distribution $p_{avg} = \frac{1}{T}\sum_{i=1}^T p_i$, such as uniformity, identity to a given reference, or the closeness of two sequences of distributions. This setting models practical scenarios with heterogeneous data sources (different users, time periods, etc.) where data are independent but not identically distributed. The paper’s primary contributions are theoretical results characterizing how the sample complexity of property testing depends on the number of samples $c$ per distribution. The theoretical results are stated clearly with conditions (e.g. explicit constants $\alpha$, regimes for $\varepsilon$) and seem to be mathematically rigorous and to what I could verify to the best of my ability are correct. The submission provides detailed proofs, adapting techniques from sources like Diakonikolas et al. (collision-based testing) and Goldreich’s reduction for identity testing. The authors acknowledge where their analyses mirror prior proofs (for instance, variance bounds adapted from the i.i.d. case) and where new ideas are needed (such as the pooled-sample lower bound). There is no empirical evaluation which is fine for a theoretical paper.

**Audience:**

Yes

**Claims And Evidence:**

Yes

**Requested Changes:**

Please fix the (minor) typos, add more clarity so the proofs are easier to read and understand, and if possible clearly state future open directions; more details in the strenghts in weaknesses.

**Strengths And Weaknesses:**

Strenghts:
The problem setting of testing properties with non-i.i.d. samples across distributions is well-motivated, but the technical contributions of the paper seem to be as incremental The positive results (Theorem1.3 and Corollary1.4) essentially show that known distribution testing algorithms (collision-based $\ell_2$-statistics) can be adapted to the heterogeneous scenario as long as $c\ge 2$. The sample complexity achieved matches the classical results in the regime of interest. This is certainly a nice observation, but it might not be viewed as a deep new technique — it is leveraging existing tools (with some variance analysis tweaks). The most novel contribution is the negative result (Theorem 1.5) demonstrating the necessity of using label information (or multiple samples per distribution). Please correct me if I am misunderstanding. That lower bound is quite technically involved and appears to be the main theoretical innovation.

Weaknesses
The paper has a number of  minor typographical and presentation issues ( listed below) that detract from its clarity. Clarity issues + suggestions/questions:

The proof of Theorem1.5 (lower bound for “pooled” estimators) is very intricate, spanning multiple appendices (Appendix D, E, F). While I did not find a specific fatal flaw in the reasoning, the construction and analysis are difficult to parse and verify. For example, the authors construct pairs of distributions with carefully chosen “collisions” properties and then use the Gershgorin circle theorem and “a fact” from Valiant & Valiant to bound the total variation between outcome distributions. I would suggest creating an appendix with such useful “facts” to be included (with refernce ofcourse) within the paper itself to make it self-sufficient. Similarly in Appendix E and F, the authors claim the two constructed sample-output distributions $G_1$ and $G_2$ (corresponding to the two scenarios) have very close fingerprint distributions (frequency count profiles), ultimately within $0.01$ in total variation. The argument relies on showing certain moment or covariance matrices are almost identical (all eigenvalues in $[1-10^{-14},,1+10^{-14}]$ for a matrix difference, etc.). These steps invoke advanced tools (e.g. linear algebraic facts) without much explanation, which should/could be included in the “useful facts” in the additional appendix. It is hard to verify at a glance that all conditions for applying those tools are satisfied. In particular, ensuring that the perturbation bounds (like those from Gershgorin’s theorem) are tight enough to imply the desired $d_TV \le 0.01$ which requires careful error accounting. the lower bound proof contains no clear mathematical \emph{error} that I could pinpoint, but its presentation lacks sufficient detail to be easily checkable. This is a concern in terms of completeness and rigor: the authors should consider elaborating the proof (or at least providing a high-level outline in the main text) to convince the reader that each step is valid.

it would be helpful if the authors explicitly clarified in the main text why the Poisson model avoids the impossibility result of Claim 1.2. In particular, the statement “receiving $ Poi(c)$ samples from every distribution is equivalent to receiving $ Poi(cT)$ samples from $p_{avg}$” should be qualified: this equivalence holds in distribution (the pooled sample count from each element of the domain is Poisson with mean $cT p_{avg}}(x)$, by the Poisson superposition property). Thus all tests that depend only on the overall counts cannot distinguish between getting samples in one batch vs. across $T$ groups. This subtle point is essentially left to the reader. While not a flaw, it is a mathematical detail that should be stated for completeness.

I would suggest the author’s claim of O(k^{2/3}/ \varepsilon^{8/3}) should hold for c=2 should be explicitly marked as a conjecture in the future work. Is there a reason the authors believe this to be true ? Is it because of prior works of Batu et al for two distribution testing?  What are the main obstacles preventing the extension of Theorem1.6 to the case $c=2$? Do the authors think that $\varepsilon^{-8/3}$ is tight in the non-identical sample setting, or is it an artifact of the proof techniques? In particular, which part of the analysis causes the deterioration from $4/3$ to $8/3$ (for example, is it a union bound over many distributions, a higher moment bound, or something about needing a small success probability to “amplify”)? If one were to attempt to improve the $\varepsilon$-dependence to match the i.i.d. optimal rate, what modifications to the tester or its analysis might be required? I realize this may not be a direct/easy question to answer and does not reflect upon the paper’s evaluation, I am just curious about this.

The paper makes it clear that without multiple samples per distribution, one cannot test properties of the average distribution in general. Prior work (Aliakbarpour & Silwal 2019, etc.) got around this by assuming some structure or limited variation in the distributions. Do the authors have insight into what kinds of structural assumptions on $p_i$ could potentially substitute for having $c>1$? For example, if we assume that all $p_i$ are themselves close to $p_{avg}$ (say each $p_i$ is within $\delta$ in total variation of $p_ {avg}$), could one design a tester for $p_{avg}}$ with $c=1$? Or if all $p_i$ come from a parametric family (like all are Binomial or Zipf distributions with different parameters), would that make uniformity testing feasible with one sample each… do the authors think the lower bound of Claim 1.2 is completely general or if it breaks under natural restrictions … in other words, aside from collecting more samples per source, are there any other reasonable ways to overcome the $c=1$ barrier? Any comments on this would help place the results in context and possibly suggest directions for future work.

Minor:
Abstract: applies --> apply
Pg 4: “extends” --> extend
“are are”
“and and” – appendix
“for for” –appendix
Theres two Lemma 4 which seem to be hard-coded since these do not follow the paper’s numbering scheme (Theorem1.3, Corollary1.4, Theorem1.5 … etc)

---

> ### Author Response · Authors · 2025-08-14
>
> We thank the reviewer for their thorough review and positive comments. We are pleasantly surprised by the thoroughness of the reviews :) and we sincerely appreciate their time and effort spent in reading our paper. We will fix the typos pointed out.
>
> We first briefly reiterate the motivation and context for our work. We agree that our major contribution, indeed, lies in the problem formulation and the conceptual takeaways. Heterogeneity naturally arises in data collection, especially when data is gathered across time, geography, or users, as discussed in the paper. Existing sublinear testers are often applied without accounting for such variation. This raises a natural question: how robust are these testers to heterogeneity? Our main contribution is in formulating this question and in the following conceptual takeaway: while existing testers are not directly applicable, they can be effectively adapted---provided one collects at least two samples per distribution and leverages source info. We also show the inapplicability of existing testers (which do not use source information) by proving a lower bound for pooled samples. Establishing this lower bound involves an independently interesting new construction, together with several non-trivial technical arguments. Thus, the lower bound can be seen as a technically novel contribution.
>
> It is true that the analysis of our proposed testers (i.e., the upper bounds) builds on existing techniques and doesn’t need fundamentally new ideas. However, technical novelty may not be the best metric to evaluate these bounds. They are valuable because they convey the main takeaway: that existing testers can be adapted to heterogeneous settings by effectively leveraging source information. In this sense, the simplicity of the testers and their analysis can be viewed as a strength---making them easier to apply and generalize.
>
> **Regarding the presentation of Theorem 1.5**:
>
> We thank the reviewer for their careful reading—the suggestions are indeed very helpful and will help in improving the presentation of the proof. In the revised version, we will restructure the technical details in the appendix, and also elaborate more on the proof outline in the main body.
>
> **Regarding Poissonization and Claim 1.2**:
>
> We wanted to mention that in Appendix C, we do indeed spell out why Poissonization formally circumvents the impossibility result of Claim 1.2, as the reviewer suggests. Nevertheless, we will add a line detailing the required property about Poisson random variables in the main body.
>
> **Regarding closeness testing with $c=2$**:
>
> Thanks for the suggestion about stating the $c=2$ question formally as a conjecture. For closeness testing, in addition to keeping track of collisions for estimating the $l_2$ norm, we also require estimating “heavy” and “light” support elements in $p_{avg}$ and $q_{avg}$. We used $c=3$ as this ensures statistical independence between the samples used to estimate collisions and those used to estimate heavy/light elements, which makes the technical arguments cleaner. Whether this independence is really required is an interesting question: it is plausible that with a more involved analysis which reuses the same samples, and carefully keeps track of dependencies, the result might still hold. With regards to $\varepsilon^{8/3}$ vs $\varepsilon^{4/3}$: again, since we adapt Batu et al.’s collision based tester for closeness testing in the i.i.d setting, we directly borrow this dependence on $\varepsilon$ from their bound. Whether this is optimal in the non-i.i.d setting is not clear, but it is likely not. We will mention, however, that the optimal tester from Chan et al. in the i.i.d setting that gets the $\varepsilon^{4/3}$ dependence is based on a different statistic, namely count of each support element seen in the sample, as opposed to collisions. It is not clear to us how one would write an estimator based on counts in the non-iid setting for small values of $c$ (or in fact, any other estimator other than the collision-based one), and relate its expectation to the $l_2/l_1$ norm of the average distributions. For a start here, it would even be worth deriving different estimators that get sublinear sample complexities for the simpler problem of uniformity testing.

---

> > ### Author Response · Authors · 2025-08-14
> >
> > **Regarding structural assumptions on $p_i$ to substitute for $c>1$**:
> >
> > This is a great question, and indeed ripe for future inquiry! As initial thoughts, consider the case where all the $p_i$’s are $\delta/T$ away in TV from $p_{avg}$. In this case, even with $c=1$ we can pretend that all our $T$ samples are from $p_{avg}$ itself with probability $1-\delta$. So, we can apply any optimal tester from the i.i.d setting, and get a result for the non-i.i.d setting, with the same sample complexity $T=O(\sqrt{k}/\varepsilon^2)$, at the expense of an additional $\delta$ failure probability. But this analysis would necessitate each $p_i$ to be as much as $\varepsilon^2/\sqrt{k}$ close to $p_{avg}$ in TV; it would be very interesting to see if it is possible to tolerate larger gaps!
> >
> > With regards to what happens when the $p_i$’s come from a structured family: in the lower bound in Claim 1.1, we heavily use that the $p_i$s have disjoint supports. So, to the extent that members from families like zipfian distributions can be made to imitate such disjoint supports/”point mass behavior”, the lower bound for $c=1$ should work out, even if all the $p_i$’s belong to the family. It is plausible that the lower bound argument should stop holding when the $p_i$’s are restricted to have considerably overlapping supports, in which case arguments similar to those in the previous paragraph might make $c=1$ work.

---

### Comment · Action_Editor_CU61 · 2025-08-20

Hi authors,

The reviewers have a scheduled deadline (moveable) two weeks from now, at which point they have to make their final recommendation. It'd be great if you could upload a revision (especially with the revised proof presentation as you promised), color-coded to highlight changes, so the reviewers can see a final version before making that final recommendation.

Best,

Jasper

---

> ### Author Response · Authors · 2025-08-30
> **Manuscript updated with feedback incorporated**
>
> Dear Jasper and reviewers,
> Thank you again for all the feedback. We have updated our manuscript to reflect the suggested changes, which are color-coded in red. In particular, we have included a section (1.2) describing all concrete open directions, significantly simplified the technical proofs in Appendix F for the lower bound on pooled estimators, and also collected facts used in the proof in a dedicated section F.1 We have also fixed typos and made other cosmetic changes that were suggested.
> Best,
> Authors

---

### Decision · Action_Editor_CU61 · 2025-10-03

**Recommendation:** Accept as is

**Audience:**

Yes

**Audience Explanation:**

The problem setting is clearly motivated, and falls in the well-studied field of distribution testing. The paper presents interesting results that the distribution testing community would care about.

**Claims And Evidence:**

Yes

**Claims Explanation:**

The problem setting of testing the average distribution of independent non-identical samples is clearly stated and motivated. All mathematical claims are stated clearly and supported by proofs.